# Conflicting Biases at the Edge of Stability:
# Norm versus Sharpness Regularization

**Maria Matveev** [* 1 2]   **Vit Fojtik** [* 3]   **Hung-Hsu Chou** [4]   **Gitta Kutyniok** [1 2 5 6]   **Johannes Maly** [1 2]

## Abstract

The remarkable generalization properties of over-parameterized networks are often attributed to implicit biases, such as norm minimization at small learning rates and low sharpness in the Edge-of-Stability regime. In this work, we argue that a comprehensive understanding of the generalization performance of gradient descent requires analyzing the interaction between these various forms of implicit regularization. We empirically demonstrate that the learning rate interpolates between low parameter norm and low sharpness of the trained model. We furthermore prove that neither implicit bias alone minimizes the generalization error for diagonal linear networks trained on a simple regression task. These findings demonstrate that focusing on a single implicit bias is insufficient to explain good generalization, and they motivate a broader view of implicit regularization that captures the dynamic trade-off between norm and sharpness induced by non-negligible learning rates.

## 1. Introduction

First-order methods such as *gradient descent (GD)* are at the core of optimization in deep learning, used to train models which generalize remarkably well to unseen data while being able to interpolate random noise (Zhang et al., 2021). A widely believed explanation for this impressive generalization ability on meaningful data is that GD and its variants exhibit an implicit bias — a tendency of the optimization algorithm to favor well-structured solutions.

When rigorously characterizing this implicit bias for full-batch GD, recent works often consider small learning rates or even the corresponding *gradient flow (GF)*, which is GD's continuous time limit under infinitely small learning rates. For classification tasks, GF has been shown to favor max-margin solutions (Soudry et al., 2018). In regression tasks using diagonal linear networks initialized near the origin, GF induces an implicit bias toward parameters of minimal norm (Woodworth et al., 2020). In practice, however, optimization relies on finite learning rates that are bounded away from zero, raising the question of whether these explanations remain valid also in such scenarios.

At the same time, it was observed for standard architectures that full-batch GD can minimize the training loss even with learning rates that are larger than what classical optimization theory would require (Jastrzebski et al., 2019; Cohen et al., 2021). To be more precise, when optimizing a (locally) $L$-smooth[1] loss function $\mathcal{L} \colon \mathbb{R}^p \to \mathbb{R}$ via full-batch GD,

$$\boldsymbol{\theta}_{k+1} = \boldsymbol{\theta}_k - \eta \nabla \mathcal{L}(\boldsymbol{\theta}_k) \qquad (1)$$

with fixed learning rate $\eta > 0$, it is well-known that

$$\mathcal{L}(\boldsymbol{\theta}_{k+1}) \leq \mathcal{L}(\boldsymbol{\theta}_k) - \eta \left( 1 - \frac{L\eta}{2} \right) \|\nabla \mathcal{L}(\boldsymbol{\theta}_k)\|_2^2, \quad (2)$$

which means that monotonic decrease of GD is only ensured for $\eta < 2/L$ (Bubeck et al., 2015). This suggests for general twice differentiable $\mathcal{L}$ that GD with learning rate $\eta$ becomes unstable if $\|\nabla^2 \mathcal{L}(\boldsymbol{\theta}_k)\| > 2/\eta$. As a result, the training loss $\mathcal{L}$ is not to be expected to decrease in these sharp regions of the loss landscape.

When training neural networks via GD with fixed $\eta > 0$, it was however confirmed in extensive simulations (Cohen et al., 2021) that the *sharpness* $S_{\mathcal{L}}(\boldsymbol{\theta}_k) = \|\nabla^2 \mathcal{L}(\boldsymbol{\theta}_k)\|$ of the training loss $\mathcal{L}$ at iterate $\boldsymbol{\theta}_k$ increases along the GD trajectory until it exceeds the critical value $2/\eta$ at some $\boldsymbol{\theta}_{k_0}$. For $k > k_0$, the sharpness of the iterates starts hovering

---

[*]Equal contribution [1]Department of Mathematics, LMU Munich, Munich, Germany [2]Munich Center for Machine Learning (MCML), Munich, Germany [3]Prusa Research, Prague, Czech Republic (Work done while at LMU Munich & MCML.) [4]Department of Mathematics, University of Pittsburgh, Pittsburgh, PA, US [5]Institute for Robotics and Mechatronics, DLR-German Aerospace Center, Oberpfaffenhofen, Germany [6]Department of Physics and Technology, University of Tromsø, Tromsø, Norway. Correspondence to: Maria Matveev <matveev@math.lmu.de>.

*Proceedings of the 43rd International Conference on Machine Learning*, Seoul, South Korea. PMLR 306, 2026. Copyright 2026 by the author(s).

---

[1]A differentiable function $\mathcal{L} \colon \mathbb{R}^p \to \mathbb{R}$ is called $L$-smooth if $\nabla \mathcal{L}$ is $L$-Lipschitz. If $\mathcal{L}$ is twice differentiable, this is equivalent to the Hessian having operator norm $\|\nabla^2 \mathcal{L}\|$ bounded by $L$.

around and slightly above this value (see Figure 14 for illustration). In this phase, the loss decreases non-monotonically and faster than when using adaptive learning rates that stay in the stable regime $\eta_k < 2/S_{\mathcal{L}}(\boldsymbol{\theta}_k)$. Accordingly, the authors dubbed the phases $k < k_0$ *"Progressive Sharpening"* and the phase $k > k_0$ *"Edge of Stability (EoS)"*. In practice, convergence in the EoS regime is attractive due to the fast average loss decay. It was even suggested that large learning rates and thus EoS might be necessary to learn certain functions (Ahn et al., 2023). More importantly, recent works on EoS showed that large learning rates induce an implicit bias of GD towards minimizers with low sharpness (Ahn et al., 2022). Indeed, for fixed $\eta > 0$ and twice differentiable $\mathcal{L}$, GD can only converge towards stationary points $\boldsymbol{\theta}_\star$ with $S_{\mathcal{L}}(\boldsymbol{\theta}_\star) < 2/\eta$.

In summary, these different lines of works suggest that GD in (1) exhibits at least two distinct but entangled forms of implicit bias; one stemming from the underlying GF $\boldsymbol{\theta}' = -\nabla\mathcal{L}(\boldsymbol{\theta})$ and one induced by its learning rate $\eta$, both visualized in Figure 1. To fully understand the success of GD-based training via implicit bias, it is therefore insufficient to analyze each bias in isolation. Instead, it is essential to understand the trade-off between various biases and answer the central question: How do different implicit biases interact when GD is used for training neural networks? A better understanding of this interaction may ultimately lead to more principled choices in the design of training algorithms and hyperparameters.

### 1.1. Contribution

Our work focuses on the two previously mentioned biases: the sharpness regularization induced by large learning rates (Ahn et al., 2022) and the norm-regularization induced by vanishing learning rates due to the compositional structure of feedforward networks (FFNs) (Chou et al., 2023).

Our contribution consists of three major points:

(i) **Implicit bias trade-off in training:** Across a wide range of settings, we empirically demonstrate that at the end of training there is a trade-off between small norm of the parameters and small sharpness of the training loss (see Figure 1 for an illustration, and Figure 2 for a prototypical experiment). This trade-off is controlled by the learning rate. When comparing the final solutions across a range of learning rates (see Section 2), we observe a sharp phase transition at a data- and model-dependent critical learning rate $\eta_c$. Below $\eta_c$, both the norm and sharpness remain nearly constant. Above $\eta_c$, increasing the learning rate leads to an overall trend of increasing norm and decreasing sharpness. *We emphasize that this phase transition occurs when*

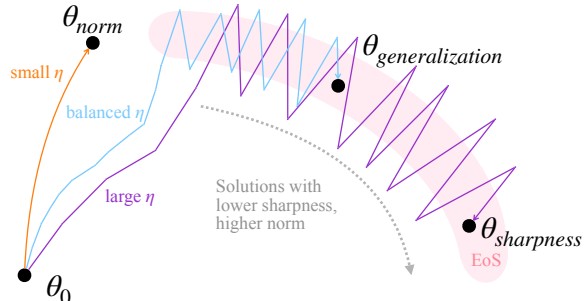

*Figure 1.* Starting at an initial parameter $\theta_0$, GF and GD with a small learning rate $\eta$ implicitly regularize the parameter norm ($\theta_{norm}$ has low norm). For GD with intermediate or large learning rates, the trajectories exhibit oscillatory Edge of Stability (EoS) behavior, implicitly regularizing the sharpness ($\theta_{sharpness}$ has low sharpness). Best generalization often is attained by carefully balancing the learning rate. In the EoS regime, we find a trade-off between low sharpness of the training loss and low norm of the final model parameter.

*comparing final GD iterates over the choice of learning rate, and does not correspond to the transition from Progressive Sharpening to EoS observed for fixed learning rate $\eta$ over the iterates $\boldsymbol{\theta}_k$ of GD (Cohen et al., 2021). To highlight that our observations do not depend on the specific choice of norm, we present different norms in Figures 2 – 4, and compare different norm choices in Appendix I.9.*

(ii) **Impact on generalization:** Remarkably, low generalization error often does not align with either extreme of the learning rate spectrum and never aligns with minimal norm. In some settings, the test error follows a U-shaped curve, with the best generalization occurring at intermediate learning rates where norm and sharpness biases are balanced, see Section 2.4. The learning rate can be interpreted as a regularization hyperparameter that controls generalization capacity of the resulting model, cf. Andriushchenko et al. (2023a).

(iii) **Theoretical analysis of a simple model:** Restricting ourselves to the strongly simplified setting of training a shallow diagonal linear network with shared weights for regression on a single data point with square loss, in Section 3 we analyze how the norm- and sharpness minimizers on the solution manifold $\mathcal{L} = 0$ are related and how they compare in terms of generalization. In fact, we provide scenarios where the lowest expected generalization error is attained by neither of them and the learning rate controls the generalization of the GD solution. Serving as a basic counterexample in which single biases do not generalize optimally, this supports our conjecture that the generalization behavior of neural networks cannot be explained by a

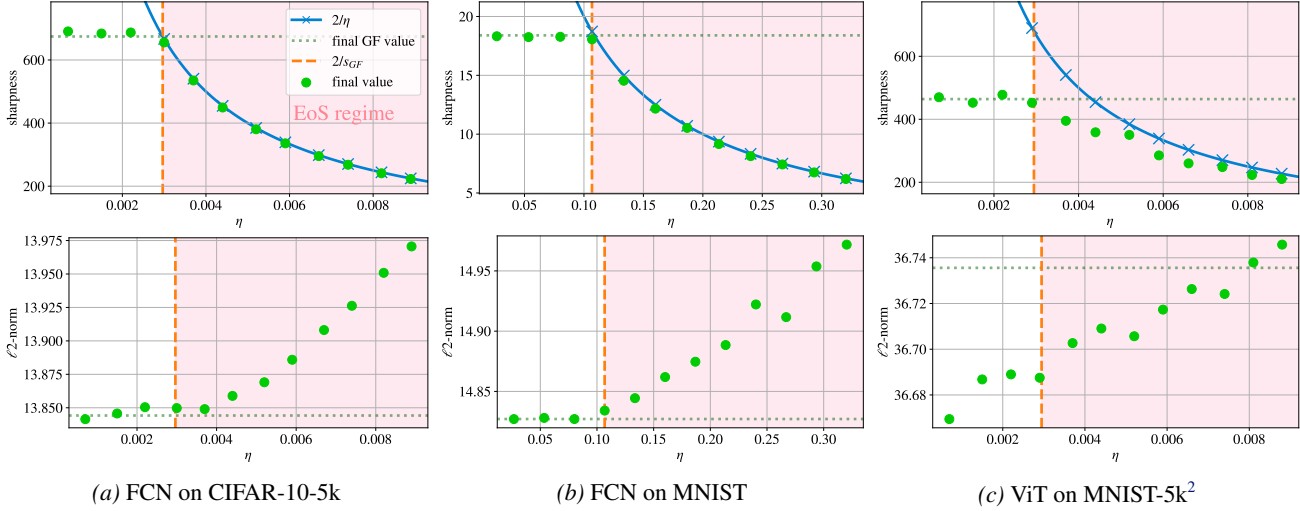

*(a)* FCN on CIFAR-10-5k     *(b)* FCN on MNIST     *(c)* ViT on MNIST-5k[2]

*Figure 2.* A critical learning rate $\eta_c = 2/s_{\text{GF}}$ marks a sharp phase transition between two regimes, a flow-aligned regime, where solutions match gradient flow in sharpness and norm, and an Edge-of-Stability (EoS) regime, where sharpness decreases while the $\ell_2$-norm increases, indicating a trade-off between low sharpness and small norm. Here, three models are trained with full-batch gradient descent with varying learning rates. This behavior is observed consistently across a wide range of experiments, see Section 2.

single implicit bias of GD. We analyze a comparably simple classification setting in Appendix G and extend the generalization analysis to multiple data points in Appendix F.2.

## 1.2. Notation and Outline

In the remainder of the paper, we denote vectors $\mathbf{x} \in \mathbb{R}^d$ and matrices $\mathbf{X} \in \mathbb{R}^{n \times d}$ by bold lower and upper case letters, and abbreviate $[n] := \{1, \ldots, n\}$. For vectors/matrices of ones and zeros we write $\mathbf{1}$ and $\mathbf{0}$, where the respective dimensions are clear from the context. The sharpness of a twice differentiable function $f \colon \mathbb{R}^d \to \mathbb{R}$ at a point $\boldsymbol{\theta}$ is defined as

$$S_f(\boldsymbol{\theta}) := \|\nabla^2 f(\boldsymbol{\theta})\| = \max_{\lambda \in \sigma(\nabla^2 f(\boldsymbol{\theta}))} |\lambda|,$$

where $\|\cdot\|$ denotes the operator norm and $\sigma(\mathbf{M})$ the spectrum of a matrix $\mathbf{M} \in \mathbb{R}^{d \times d}$. By $\odot$ we denote the (entry-wise) Hadamard product between two vectors/matrices and write $\mathbf{z}^{\odot k} = \mathbf{z} \odot \cdots \odot \mathbf{z}$ for the $k$-th Hadamard power. The support of a vector $\mathbf{z} \in \mathbb{R}^d$ is denoted by $\operatorname{supp}(\mathbf{z}) = \{i \in [d] \colon z_i \neq 0\}$ and the diagonal matrix with diagonal $\mathbf{z}$ by $\mathbf{D_z} \in \mathbb{R}^{d \times d}$. For any index set $I \subset [d]$ and $\mathbf{z} \in \mathbb{R}^d$, we furthermore write $\mathbf{z}|_I \in \mathbb{R}^d$ for the vector that is zero on $I^c$ and $\mathbf{z}$ on $I$.

Our numerical results are presented in Section 2. To shed

some light on the observed phenomena, we analyze a simple regression model in Section 3. Finally, we conclude in Section 4 with a discussion of our results. All proofs and further insights are deferred to the appendix.

## 1.3. Related Works

Before presenting our results in detail, let us review the current state of the art on analyzing the implicit bias of GF and GD, on EoS, which represent the two forms of regularization we study. Thereafter we discuss the question how generalization relates to each implicit bias. This section serves as a synopsis of Appendix A.

**Implicit Bias of GF.** To understand the remarkable generalization properties of unregularized gradient-based learning procedures for deep neural networks (Zhang et al., 2021; Belkin et al., 2019), a recent line of works has been analyzing the implicit bias of GD towards parsimoniously structured solutions in simplified settings such as linear classification (Soudry et al., 2018), matrix factorization (Gunasekar et al., 2017), training linear networks (Geyer et al., 2020), training two-layer networks for classification (Chizat & Bach, 2020), and training linear diagonal networks for regression (Vaskevicius et al., 2019). All of these results analyze GD with small or vanishing learning rate, i.e., the implicit biases identified therein can be ascribed to the underlying GF dynamics. Building on these results, parameter norm is widely used as a proxy for model complexity and structural simplicity in deep networks (Neyshabur et al., 2015). It is worth noting that there are other mechanisms inducing algorithmic regularization such as label noise (Pesme et al., 2021) or weight normalization (Chou et al., 2024b).

---

[2]The properties shown in the two left columns correspond to fully-connected FFNs (FCNs) trained with mean squared error (MSE), while the Vision Transformer (ViT) in the right column uses cross-entropy loss. We discuss the resulting qualitative differences between both loss functions in Appendix I.4.

**Edge of Stability.** Whereas most of the above studies rely on vanishing learning rates, results by Cohen et al. (2021) on EoS suggest that GD under finite, realistic learning rates behaves notably differently from its infinitesimal limit. Recently, a thorough analysis of EoS has been provided for training linear classifiers (Wu et al., 2024) and shallow near-homogeneous networks (Cai et al., 2024) on the logistic loss via GD. In particular, GD with fixed learning rate $\eta > 0$ can only converge to sufficiently flat minima (Ahn et al., 2022), i.e., stationary points $\boldsymbol{\theta}_\star$ of a loss $\mathcal{L}$ with bounded sharpness $S_{\mathcal{L}}(\boldsymbol{\theta}_\star) < 2/\eta$. Note that EoS was first observed for *stochastic gradient descent (SGD)* (Wu et al., 2018), for which the analogous sharpness bounds also depend on the batch size (Wu et al., 2022). Ghosh et al. (2025) show that large learning rates in deep linear networks induce a so-called beyond–EoS regime in which GD oscillates stably around the minimal sharpness solution.

**Generalization and Sharpness.** In the past, various notions of sharpness have been studied in connection to generalization. The idea that flat minima benefit generalization dates back to Wolpert (1993). Since then, many authors have conjectured that flatter solutions should generalize better. Nevertheless, the relationship between flatness and generalization remains disputed. Studies have found little correlation between sharpness and generalization performance (Kaur et al., 2023), even when using scaling invariant sharpness measures like *adaptive sharpness* (Kwon et al., 2021). On the contrary, in various cases the correlation is negative, i.e., sharper minima generalize better. Notably, one of these works by Andriushchenko et al. (2023a) observe correlation of generalization with parameters such as the learning rate, which agrees with the herein presented idea of an implicit bias trade-off that is governed by hyperparameters of GD. *We emphasize that with the present work we do not contribute to resolving the question of which notion of sharpness (Tahmasebi et al., 2024) might be most accurate as a measure of generalization. We restrict ourselves to the so-called worst-case sharpness $S_{\mathcal{L}}$ defined as the operator norm of the loss Hessian since this version of sharpness is provably regularized by GD with large learning rates (Ahn et al., 2022). We verify in Appendix I.9 that our observed trade-off persists with other sharpness measures, including reparameterization-invariant ones.*

**Generalization and Norm.** In sparse resp. low-rank recovery, good generalization is provably achieved via $\ell_1$-resp. nuclear norm minimization (Foucart & Rauhut, 2013). This well-established theory offers a possible explanation for the occasionally observed correlation between flatness and generalization. Ding et al. (2024) show for (overparameterized) matrix regression that sharpness and nuclear norm ($\ell_1$-norm on the spectrum) minimizers lie close to each other. Hence, the good generalization of flat minima might just be a consequence of flat minima lying close to nuclear norm minimizers. This spectral perspective is further supported by Bartlett et al. (2017). The observation that a single bias causes generalization might only stem from special situations in which several independent biases agree, a phenomenon also observed in scalar factorization Wang et al. (2022a, Appendix F.2). This point of view is supported by Wen et al. (2023) and aligns with our observations.

## 2. Conflicting Biases

Across a wide range of training setups with varying architectures, activations, loss functions, and datasets, we consistently observe a trade-off between sharpness and norm of the final parameters as soon as the learning rate increases above a critical value. In Figure 2, we show examples of this transition, revealing two distinct regimes: The *flow-aligned regime* where both final sharpness and norm remain nearly constant with respect to the learning rate, and the *Edge-of-Stability (EoS) regime* where sharpness decreases hyperbolically and the $\ell_1$-norm increases approximately linearly. For GD trained until loss $\varepsilon$ the critical learning rate at which this phase transition occurs depends on the gradient flow solution and is approximately given by $\eta_c := 2/s_{\mathrm{GF}}^\varepsilon$. Here, $s_{\mathrm{GF}}^\varepsilon := \max_{t \le t_\varepsilon} S_{\mathcal{L}}(\boldsymbol{\theta}(t_\varepsilon))$ denotes the maximal sharpness of the GF solution $\boldsymbol{\theta}$ until time $t_\varepsilon := \inf\{t \colon \mathcal{L}(\boldsymbol{\theta}(t)) \le \varepsilon\}$, see Figure 3. When $\varepsilon$ is clear from the context, we just write $s_{\mathrm{GF}}$. We emphasize that this regime transition occurs when comparing final GD iterates initialized identically over the choice of learning rate, and does not correspond to the transition from Progressive Sharpening to EoS at $t_\eta := \inf\{t \colon S_{\mathcal{L}}(\boldsymbol{\theta}_t) \ge 2/\eta\}$ observed for fixed learning rate $\eta$ over the iterates $\boldsymbol{\theta}_k$ of GD (Cohen et al., 2021).

### 2.1. Systematic Experimental Analysis

To systematically investigate the trade-off between sharpness and norm minimization, we conduct experiments on standard vision datasets using both simple and moderately complex architectures. Since computing the sharpness during training involves estimating the largest eigenvalue of the Hessian, which scales with both model and dataset size, we primarily use compact models to allow for evaluation across a broad range of learning rates.

Following the experimental setup of Cohen et al. (2021), our base configuration consists of a fully connected ReLU network with two dense layers with 200 hidden neurons each, trained on the first 5,000 training examples from both MNIST and CIFAR-10 (LeCun et al., 2010; Krizhevsky et al., 2014). These two datasets provide complementary complexity levels and help ensure that the observed effects are not specific to a single data distribution.

We train using full-batch gradient descent in order to cleanly

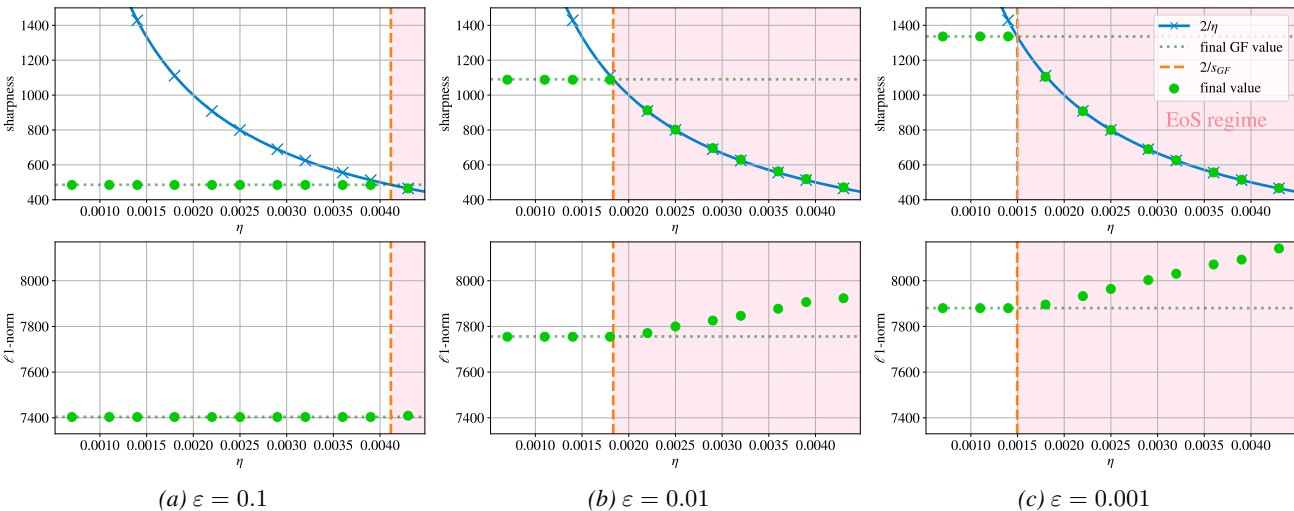

*Figure 3.* Sharpness and $\ell_1$-norm of final FCN models with tanh activation trained via MSE loss on CIFAR-10-5k for three different loss thresholds $\varepsilon$. Axis scales are equal for all three instances. Each plot illustrates a sharp regime transition as the learning rate crosses the critical threshold $\eta_c \approx 2/s_{\mathrm{GF}}^\varepsilon$, shifting from the flow-aligned regime with nearly constant sharpness and norm to the EoS regime where sharpness decreases and the norm increases.

isolate the fundamental trade-off between norm and sharpness bias driven by the learning rate $\eta$, avoiding confounding factors such as stochasticity or momentum.

To ensure comparable convergence across settings, we train to a fixed (training) loss threshold using identical, small-scale initializations across learning rates. We provide experimental and methodological details in Appendix H, as well as our research code.

We perform a systematic investigation by varying the following core components of the training setup.

(i) **Dataset size.** We train on the full MNIST and CIFAR-10 datasets, see Appendix I.1.

(ii) **Architecture.** We vary the width and depth of the FCN and study a convolutional neural network, a ResNet and a Vision Transformer (LeCun et al., 1998; He et al., 2016; Dosovitskiy et al., 2021), see Appendix I.2.

(iii) **Activation function.** We study ReLU and tanh activations, see Appendix I.3.

(iv) **Loss function.** On most settings, we compare both cross-entropy loss (CE) and mean squared error (MSE). The phase transitions are similar though differences in the time evolution exist, see Appendix I.4.

(v) **Loss threshold.** For every experiment, we vary the loss threshold to which we train, cf. Figure 3 and Appendix I.5. Note that varying the loss threshold can be interpreted as early stopping.

(vi) **Initialization.** Varying the initialization changes the

GF solution $s_{\mathrm{GF}}$, accordingly shifting the critical learning rate, see Appendix I.6.

(vii) **Parameterization**. We train FCNs with varying widths in the $\mu$P and kernel parameterizations (Yang et al., 2021; Jacot et al., 2018) in Appendix I.7 where for $\mu$P we observe a certain width-independence of the spectral properties, cf. Noci et al. (2024).

Across all variations, we consistently observe the same trade-off between sharpness and norm, and the emergence of the flow-aligned and EoS regimes. Most figures showing these variations are deferred to Appendix J due to the page limit, along with further noteworthy observations from our experiments being noted in Appendix I.

Next, we provide a high-level summary of our findings in Sections 2.2, 2.3, and 2.4.

## 2.2. Flow-aligned Regime

In the flow-aligned regime ($\eta < \eta_c$), the behavior of GD closely mirrors that of continuous-time gradient flow. This regime is characterized by stable convergence of GD and minimal deviation from the gradient flow dynamics in terms of sharpness and norm. Intuitively, the sharpness of the solution in this regime stays within the stability limits set by the learning rate in (2), i.e., $S_{\mathcal{L}}(\theta_k) \leq 2/\eta$, allowing the discrete updates to track the continuous trajectory. However, we note that contrary to previous findings such as by Arora et al. (2022), the absolute deviation from the GF trajectory is not necessarily negligible, see Appendix I.10. Nonetheless, the limits of GF and GD share nearly equal sharpness and norm values.

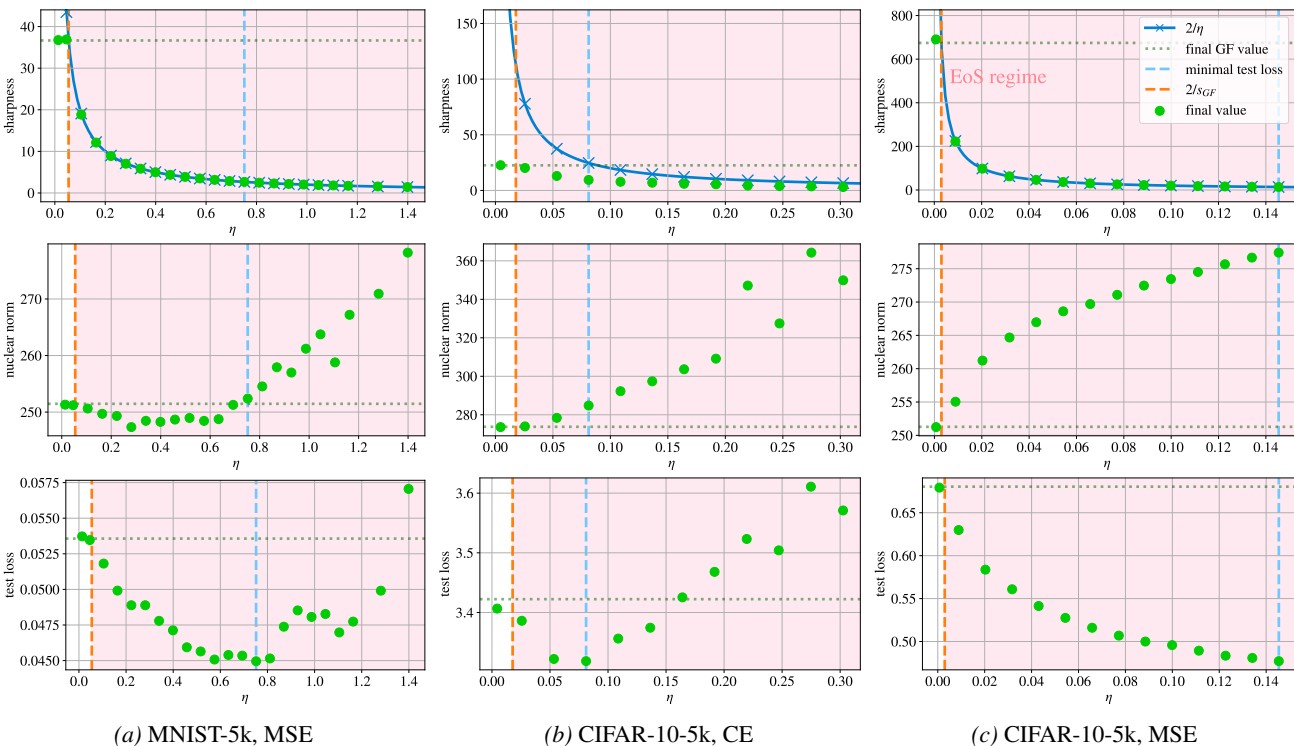

*(a) MNIST-5k, MSE*       *(b) CIFAR-10-5k, CE*       *(c) CIFAR-10-5k, MSE*

*Figure 4.* Final sharpness, nuclear norm, and test loss versus learning rate for three FCNs. On MNIST-5k with MSE loss (left), a clear U-shaped test loss indicates a trade-off between low sharpness and low nuclear norm. CIFAR-10-5k with CE loss (middle) shows a similar, though weaker trend. The best generalization typically occurs at intermediate learning rates where norm and sharpness biases are balanced. However, this is not universal — for instance CIFAR-10-5k with MSE loss (right) does not follow this pattern.

## 2.3. Edge-of-Stability Regime

As the learning rate exceeds the critical threshold $\eta_c = 2/s_{GF}$, the dynamics of GD enter the EoS regime. Here, training is governed by EoS (Cohen et al., 2021): while the loss continues to decrease on average over time, the decrease is no longer monotone and the curvature of the loss at the iterates (as measured by $S_\mathcal{L}$) fluctuates just above $2/\eta$. As GD is unable to converge to an overly sharp solution (cf. Theorem B.2), the iterates oscillate towards flatter regions. If training ends during or just after this EoS phase, the solution sharpness will therefore be near $2/\eta$.

In this regime, the sharpness $S_\mathcal{L}$ of the final network parameters thus decreases hyperbolically with the learning rate, closely tracking the function $\eta \mapsto 2/\eta$. At the same time, the norm of the final parameters increases. In some cases, there is an initial, temporary decrease in norm before the overarching trend of increasing norm and decreasing sharpness takes over at larger learning rates. We highlight that this increase in norm is not specific to the choice of norm: we observe the same qualitative trend for the $\ell_1$, $\ell_2$-norm and the nuclear norm, suggesting a general increase in model complexity as the learning rate increases, see Appendix I.9. We highlight that the observed increase in norm is not an artifact of oscillation. In our temporal analysis of these values

during training in Appendix I.11, we see that the sharpness evolves oscillatory in the EoS regime while the parameter norm grows monotonically over time, indicating that the training trajectory visits regions of the loss landscape of higher norm.

## 2.4. Generalization

When comparing the test error of the produced solutions, see Figure 4, we note that minimal norm solutions in the flow-aligned regime never lead to optimal generalization, i.e., if the test error decreases towards one extreme, it is always towards higher learning rates and increasing norm in line with prior work finding that EoS is beneficial for good generalization (Ahn et al., 2023) and the empirical success of Sharpness-aware minimization (Foret et al., 2021).

In some settings (see Table 1 for a systematic overview) we even observe a U-curve of the test error, i.e. large learning rates harm good generalization. This suggests that GD generalizes best when norm and sharpness biases are well-balanced, see Figure 4. The learning rate can then be interpreted as a regularization hyperparameter that controls generalization capacity of the resulting model. This aligns with recent independent experiments by Andriushchenko et al. (2023a).

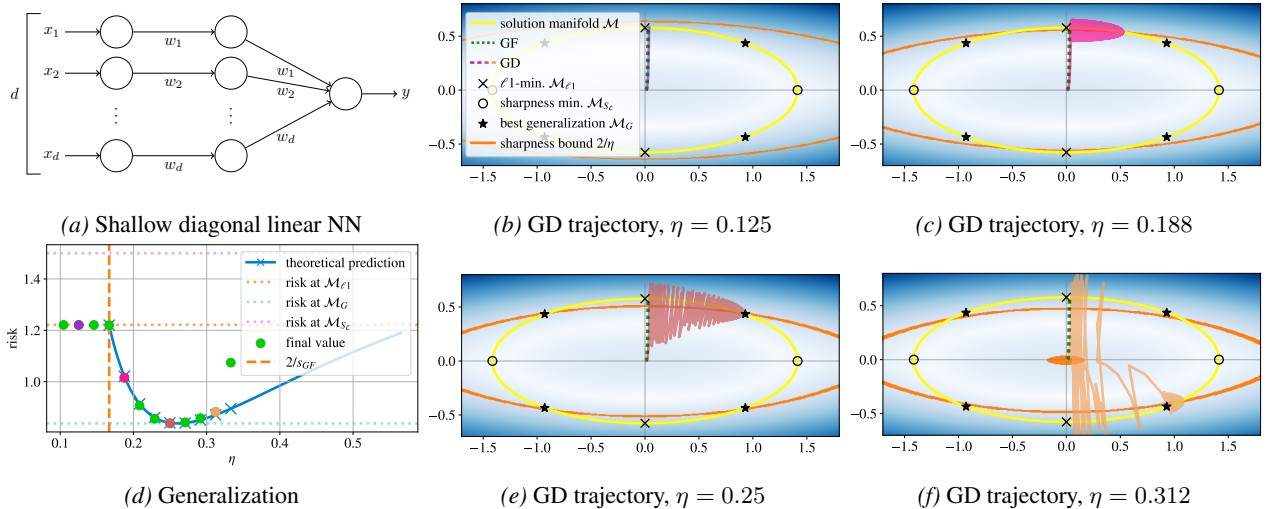

*Figure 5.* Two-layer diagonal linear model with weight sharing, shown in (5a). In (5b), (5c), (5e), and (5f) we show evolutions of weight iterates throughout training for different learning rates, where (5b) operates in the flow-aligned regime, and the others in the EoS regime. The background color map represents loss sharpness from low (white) to high (blue). The U-shaped generalization error is shown in (5d).

# 3. An Elementary Study of How Implicit Biases Interact

To shed some light on the empirical observations of Section 2, we study the implicit biases of GF and GD in the EoS regime in a simple regression task and show that for this setup, the norm and sharpness minimizers of the interpolating manifold are distinct, and neither is sufficient for best generalization. Assuming a *single data point* $(\mathbf{x}, y) \in \mathbb{R}^d \times \mathbb{R}$, we train a shallow diagonal linear network with shared weights $\mathbf{w} \in \mathbb{R}^d$ and without bias

$$\phi_{\mathbf{w}} \colon \mathbb{R}^d \to \mathbb{R}, \qquad \phi_{\mathbf{w}}(\mathbf{z}) = \mathbf{w}^T \mathbf{D}_{\mathbf{w}} \mathbf{z}, \qquad (3)$$

see Figure 5a, via the square loss

$$\mathcal{L}(y', y) = \frac{1}{2}(y' - y)^2.$$

The training objective is then

$$\min_{\mathbf{w} \in \mathbb{R}^d} \mathcal{L}(\mathbf{w}) = \min_{\mathbf{w} \in \mathbb{R}^d} \frac{1}{2} \left( \langle \mathbf{w}^{\odot 2}, \mathbf{x} \rangle - y \right)^2, \qquad (4)$$

where we overload the notation $\mathcal{L}(\phi_{\mathbf{w}}(\mathbf{x}), y) =: \mathcal{L}(\mathbf{w})$ for the sake of simplicity. Note that $\odot$ denotes the Hadamard product and $\mathbf{z}^{\odot k} = \mathbf{z} \odot \cdots \odot \mathbf{z}$ the $k$-th Hadamard power. We define the set of parameters of interpolating solutions $\phi_{\mathbf{w}}$ as

$$\mathcal{M} = \{ \mathbf{w} \in \mathbb{R}^d \colon \mathcal{L}(\mathbf{w}) = 0 \} \qquad (5)$$

and note in the following lemma that $\mathcal{M}$ is a Riemannian manifold in general. We provide the proof in Appendix C.

**Lemma 3.1.** *For $\mathcal{L}$ as in (4), define $\mathcal{M}$ as in (5) and assume that $\mathcal{M} \neq \emptyset$. If $\mathbf{x} \in \mathbb{R}_{\neq 0}^d$ and $y \neq 0$, then $\mathcal{M}$ is a Riemannian manifold with tangent space $T_{\mathbf{w}} \mathcal{M} = (\mathbf{x} \odot \mathbf{w})^\perp$ at $\mathbf{w} \in \mathcal{M}$.*

While this training model is strongly simplistic, it allows us to explicitly compare the implicit biases induced by GF and by EoS, and to compute their generalization errors w.r.t. the realization of $(\mathbf{x}, y)$. Indeed, it is known that in this setting GF initialized at $\mathbf{w}_0 = \alpha \mathbf{1}$, for $\alpha > 0$ small, converges to an end-to-end model $\mathbf{w}_\star^{\odot 2}$ that approximately minimizes the $\ell_1$-norm among all interpolating solutions (Chou et al., 2023), see Theorem B.1 in Appendix B.[3] Similarly, under mild technical conditions on $\mathcal{L}$, which are fulfilled in the present study, it is well-known for GD with learning rate $\eta > 0$ that for almost every initialization $\mathbf{w}_0 \in \mathbb{R}^d$ the iterates $\mathbf{w}_k$ can only converge to stationary points $\mathbf{w}_\infty$ with $S_{\mathcal{L}}(\mathbf{w}_\infty) \leq 2/\eta$ (Ahn et al., 2022), see Theorem B.2 in Appendix B. In consequence, GD is implicitly restricted to limits with low sharpness if $\eta$ is chosen sufficiently large.

The following result now characterizes how the norm- and sharpness-minimizers of (4) relate. In particular, it illustrates that they are clearly distinct in general.

**Proposition 3.2.** *For $\mathbf{x} \in \mathbb{R}_{\neq 0}^d$ and $\mathcal{L}$ as in (4) with $\mathcal{M} \neq \emptyset$ as in (5), the following hold:*

*(i) To have*

$$\mathbf{w} \in \mathcal{M}_{\ell_1} := \arg \min_{\mathbf{z} \in \mathcal{M}} \| \mathbf{z}^{\odot 2} \|_1,$$

---

[3]In consequence, the network parameters $\mathbf{w}_\star$ minimize the squared $\ell_2$-norm.

*it is necessary that* $\mathbf{x}|_{\mathrm{supp}(\mathbf{w})} = x_{\max} \cdot \mathbf{1}|_{\mathrm{supp}(\mathbf{w})}$, *for* $x_{\max} = \max_i |x_i|$. *If* $\mathbf{x} \in \mathbb{R}^d_{>0}$, *this condition is also sufficient. In particular, we have in this case that*

$$\mathcal{M}_{\ell_1} = \left\{ \mathbf{w} \in \mathbb{R}^d \colon \|\mathbf{w}\|^2_2 = \frac{y}{x_{\max}} \right. \tag{6}$$
$$\left. and\ \mathrm{supp}(\mathbf{w}) \subset \arg\max_i x_i \right\}.$$

*(ii) To have*

$$\mathbf{w} \in \mathcal{M}_{S_{\mathcal{L}}} := \arg\min_{\mathbf{z} \in \mathcal{M}} S_{\mathcal{L}}(\mathbf{z}),$$

*it is necessary that* $\mathbf{x}|_{\mathrm{supp}(\mathbf{w})} = x_0 \cdot \mathbf{1}|_{\mathrm{supp}(\mathbf{w})}$, *for some* $x_0 \in \mathbb{R}$.
*If* $\mathbf{x} \in \mathbb{R}^d_{>0}$, *it is necessary and sufficient that the previous condition holds with* $x_0 = x_{\min} = \min_i x_i$. *In particular, we have in this case that*

$$\mathcal{M}_{S_{\mathcal{L}}} = \left\{ \mathbf{w} \in \mathbb{R}^d \colon \|\mathbf{w}\|^2_2 = \frac{y}{x_{\min}} \right. \tag{7}$$
$$\left. and\ \mathrm{supp}(\mathbf{w}) \subset \arg\min_i x_i \right\}.$$

*Proof sketch:* To derive the necessary conditions, we calculate Riemannian gradients and Hessians along $\mathcal{M}$ and use the respective first- and second-order necessary conditions. To derive the sufficient conditions and the explicit representations in (6) and (7), we construct simple minimizers based on canonical basis elements. The full proof is in Appendix D. □

Proposition 3.2 shows that, in general, the norm- and sharpness-minimizer on $\mathcal{M}$ do not agree. We mention that the assumption $\mathbf{x} \in \mathbb{R}^d_{\neq 0}$ is not restrictive since any zero coordinate of $\mathbf{x}$ can be removed by reducing the problem dimension. In view of Theorems B.1 and B.2, we see that depending on the learning rate, GD with initialization $\mathbf{w}_0 = \alpha\mathbf{1}$, for $\alpha > 0$ close to zero, is implicitly more biased to two disjoint sets. For $\eta \to 0$, the limit of stable GD will lie close to the set in (6); as $\eta$ increases, the limit of unstable GD (as far as it exists) will lie close to the set in (7). For $d = 2$, the situation is illustrated in Figure 5.

*Remark* 3.3. In Appendix E, we show for *multiple Gaussian* data points and an accordingly generalized loss $\mathcal{L}$ that $S_{\mathcal{L}}(\mathbf{w})$ is approximately proportional to $\|\mathbf{w}\|_\infty$. Hence, the implication of Proposition 3.2 that on (4) the implicitly regularized solutions of GD with small and large learning rates are clearly separated in space carries over from the single data point setting to multiple data points. For a more detailed discussion, we refer to Appendix E.

Despite its simplicity, our toy model can reproduce the characteristic phase transitions of norm and sharpness (Figure 2) and the U-shaped generalization curve (Figure 4).
For this, let us assume that the data follows a simple linear

regression model with $\mathbf{x} \sim \mathcal{N}(\mathbf{0}, \mathbf{I})$ and $y = \langle \mathbf{1}, \mathbf{x} \rangle + \varepsilon$, for independent $\varepsilon \sim \mathcal{N}(0, 1)$. Then, the risk $\mathcal{R}$ under $\mathcal{L}$ can be computed explicitly and the best achievable generalization error of $\phi_\mathbf{w}$ trained via (4) can be identified, see Lemma F.1.

Assume we are given a generic draw of the single data point $(\mathbf{x}_0, y_0) \sim (\mathbf{x}, y)$ with $\mathbf{x}_0 \in \mathbb{R}^d_{\geq 0}$, i.e., we consider a draw $(\mathbf{x}_0, y_0)$ from the conditional distribution $p((\mathbf{x}, y)|\mathbf{x} \geq \mathbf{0})$.[4] Note that almost surely $\mathbf{x}_0$ will satisfy $|\mathrm{supp}(\mathbf{x}_0)| \geq 2$, and have a unique minimal entry $x_{\min}$ at index $k_{\min}$ and a unique maximal entry $x_{\max}$ at index $k_{\max}$ such that the sets in (6) and (7) consist of two points each which only differ by a sign.

On this model, GD with learning rate $\eta$ will minimize $\mathcal{L}$ under constraints $S_{\mathcal{L}} \leq \frac{2}{\eta}$ due to its implicit sharpness regularization. We can now compare the limit of GD with initialization $\mathbf{w}_0 = \alpha\mathbf{1}$, for $\alpha > 0$ small, to three *idealized* training algorithms which, given input $(\mathbf{x}_0, y_0)$, output the weight vector $\mathbf{w} \in \mathbb{R}^d$ of an interpolating solution $\phi_\mathbf{w}$:

(i) **Minimal norm:** $\mathcal{A}_{\ell_1} \colon \mathbb{R}^d \times \mathbb{R} \to \mathbb{R}^d$ with $\mathcal{A}_{\ell_1}(\mathbf{x}_0, y_0) = \sqrt{\frac{y_0}{x_{\max}}} \mathbf{e}_{k_{\max}}$. This corresponds to the solution computed by GD with vanishing learning rate.

(ii) **Minimal sharpness:** $\mathcal{A}_{S_{\mathcal{L}}} \colon \mathbb{R}^d \times \mathbb{R} \to \mathbb{R}^d$ with $\mathcal{A}_{S_{\mathcal{L}}}(\mathbf{x}_0, y_0) = \sqrt{\frac{y_0}{x_{\min}}} \mathbf{e}_{k_{\min}}$. This corresponds to the solution that would be computed by GD with extremely large learning rate if convergence still happened.

(iii) **Minimal generalization error:** $\mathcal{A}_{\mathrm{opt}} \colon \mathbb{R}^d \times \mathbb{R} \to \mathbb{R}^d$ with $\mathcal{A}_{\mathrm{opt}}(\mathbf{x}_0, y_0)$ returning a risk minimizer in $\mathcal{M}_G$ (best generalizing points in $\mathcal{M}$).

Figure 5 shows four snapshots of the training dynamics for growing $\eta$. Figure 5b reflects the situation where GD has no sharpness induced restrictions on $\mathcal{M}$ and converges to a minimizer in $\mathcal{M}_{\ell_1}$, i.e. the output of $\mathcal{A}_{\ell_1}$. As long as $\eta$ is not too large (Figure 5c), the generalization minimizer falls inside the feasible set. Due to EoS, the model finds a solution with sharpness around $2/\eta$ yielding suboptimal generalization error, though risk improves over $\mathcal{M}_{\ell_1}$. For carefully tuned $\eta$, Figure 5e shows convergence of GD to a point close to the output of $\mathcal{A}_{\mathrm{opt}}$. For too large $\eta$, the sharpness constraints exclude $\mathcal{M}_G$ and GD moves closer to $\mathcal{M}_{S_{\mathcal{L}}}$. As Figure 5d illustrates, our toy model exhibits the U-shaped generalization curve observed in various training simulations, and explains it by an interpolation between implicit norm- and sharpness biases.

We note that in this example both $\mathcal{M}_{\ell_1}$ and $\mathcal{M}_{S_{\mathcal{L}}}$ lead to

---

[4]In this discussion, $(\mathbf{x}_0, y_0)$ takes the role of the single data point $(\mathbf{x}, y)$ from before and we condition to non-negative data in order to apply Proposition 3.2. We examine removing the latter limitation in Section F.1.

suboptimal generalization with $\mathcal{R}(\mathcal{M}_{\ell_1}) < \mathcal{R}(\mathcal{M}_{S_{\mathcal{L}}})$. Due to its instability, GD already diverges for many $\eta$ where the feasible set of the constrained optimization problem is non-empty, i.e., although there exist points on the solution manifold with sharpness $< 2/\eta$. Consequently, all convergent trajectories in the EoS regime achieve better generalization than $\mathcal{R}(\mathcal{M}_{\ell_1})$, although the sharpness minimizer induces a higher risk. This might be an explanation for why the U-shaped generalization curve is not always visible in our experiments.

We provide additional numerical experiments for the diagonal network in Appendix I.13. In particular, note that the GD limit is often close to a KKT point of a sharpness-restricted risk minimization on $\mathcal{M}$ (Figure 17 and Lemma F.1). In Appendix G, we analyze a comparably simplified classification model for which sharpness minimization leads to better generalization performance than norm-minimization.

## 4. Discussion

Our experiments demonstrate that a single implicit bias of gradient descent is not sufficient to explain the good generalization performance in deep learning. While solutions obtained with vanishing learning rates may have an implicit bias towards simple structures, the bias changes with increasing learning rate. This insight provides an explanation for the strong empirical influence of the learning rate on model performance. Our theoretical analysis further demonstrates that low-norm-and low sharpness minima can be geometrically distinct, with optimal generalization obtained by neither. This indicates the learning rate balances between various implicit biases, and that good generalization performance is only reached by careful fine-tuning of such hyperparameters of GD. These insights from our simplified model open the door to a broader perspective on implicit regularization which accounts for the interaction between multiple biases shaped by the optimization dynamics. Future work extending our insights to additional known biases and more realistic optimizers will be important to fully translate these insights into practical training settings.

**Limitations.** Our theoretical analysis is restricted to simple models due to the difficulty in explicitly characterizing the implicit biases of GD in more general setups. In combination with our empirical studies, it nevertheless provides consistent evidence for the observed phenomena. Our study is further limited by only considering full-batch gradient descent as well as two specific implicit biases.

## Acknowledgments

VF and MM acknowledge the financial support by the Munich Center for Machine Learning (MCML). JM and GK received partial support from MCML. JM received partial support by the Deutsche Forschungsgemeinschaft (DFG, German Research Foundation) – GRK 3081/1 – Project number 534429653. MM and GK are supported in part by the DAAD programme Konrad Zuse Schools of Excellence in Artificial Intelligence, sponsored by the German Federal Ministry of Research, Technology and Space.

## Impact Statement

This paper presents work whose goal is to advance the field of Machine Learning. There are many potential societal consequences of our work, none of which we feel must be specifically highlighted here.

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

## Supplement to the Paper "Conflicting Biases at the Edge of Stability: Norm versus Sharpness Regularization"

In this supplement, we provide additional numerical simulations and proofs that were skipped in the main paper.

## A. Related Works — Extended Discussion

We provide a more detailed review of the related literature here.

**Implicit Bias of GF.** To understand the remarkable generalization properties of unregularized gradient-based learning procedures for deep neural networks (Zhang et al., 2021; Belkin et al., 2019), a recent line of works has been analyzing the implicit bias of GD towards parsimoniously structured solutions in simplified settings such as linear classification (Soudry et al., 2018; Ji & Telgarsky, 2019), matrix factorization (Gunasekar et al., 2017; Arora et al., 2019; Chou et al., 2024a), training linear networks (Geyer et al., 2020; Stöger & Soltanolkotabi, 2021), training two-layer networks for classification (Chizat & Bach, 2020; Frei et al., 2022), and training linear diagonal networks for regression (Vaskevicius et al., 2019; Woodworth et al., 2020; Azulay et al., 2021; Chou et al., 2023). All of these results analyze GD with small or vanishing learning rate, i.e., the implicit biases identified therein can be ascribed to the underlying GF dynamics. Building on these results, parameter norm is widely used as a proxy for model complexity and structural simplicity in realistic deep networks (Neyshabur et al., 2015). This relationship is further motivated by its central role in explicit regularization such as weight decay (Goodfellow et al., 2016, Chapter 7.1.1).

**Other Types of Implicit Regularization of GD.** It is worth noting that there are other mechanisms inducing algorithmic regularization such as label noise (Pesme et al., 2021; Vivien et al., 2022) or weight normalization (Chou et al., 2024b), momentum gradient descent (Papazov et al., 2024), smoothed sign descent (Wang & Klabjan, 2025) and explicit regularization into the mirror flow (Jacobs et al., 2025). In (Andriushchenko et al., 2023b; Even et al., 2023) an intriguing connection regarding implicit regularization induced by large step sizes coupled with SGD noise has been discussed. In particular, for shallow diagonal linear networks it has been shown that SGD with large learning rates implicitly regularizes certain parameter norms (Wu & Su, 2023). For a broader overview on the topic including further references we refer to the survey by Vardi (2023).

**Edge of Stability.** Whereas most of the above works rely on vanishing learning rates, results by Cohen et al. (2021) on EoS suggest that GD under finite, realistic learning rates behaves notably differently from its infinitesimal limit. In the past few years, subsequent works have started to theoretically analyze the EoS regime. It is noted in Ahn et al. (2022) that GD with fixed learning rate $\eta > 0$ can only converge to stationary points $\boldsymbol{\theta}_\star$ of a loss $\mathcal{L}$ if $S_{\mathcal{L}}(\boldsymbol{\theta}_\star) < 2/\eta$. In Chemnitz & Engel (2025), this stability criterion of stationary points has been generalized to SGD. Note that EoS was first observed for SGD (Wu et al., 2018), for which the analogous sharpness bounds also depend on the batch size (Wu et al., 2022). Arora et al. (2022) relate normalized GD on a loss $\mathcal{L}$ to GD on the modified loss $\sqrt{\mathcal{L}}$ and show that EoS occurs $\mathcal{O}(\eta)$-close to the manifold of interpolating solutions. Under various restrictive assumptions, progressive sharpening and EoS have been analyzed by Wang et al. (2022b); Chen & Bruna (2023); Zhu et al. (2023); Kreisler et al. (2023). Recently, a thorough analysis of EoS has been provided for training linear classifiers (Wu et al., 2024) and shallow near-homogeneous networks (Cai et al., 2024) on the logistic loss via GD. The authors show that large learning rates allow a loss decay of $\mathcal{O}(1/k^2)$ which exceeds the best known rates for vanilla GD from classical optimization. Cohen et al. (2021) extended their empirical study of EoS to adaptive GD-methods for which the stability criterion becomes more involved (Cohen et al., 2023). Finally, let us mention that applying early stopping to label noise SGD with small learning rate can also induce sharpness minimization and structural simplicity of the learned weights (Gatmiry et al., 2024). As opposed to our definition of sharpness, sometimes called *worst-case sharpness*, in the latter work sharpness is measured by the trace of $\nabla^2 \mathcal{L}$ also known as *average-case sharpness*. Additionally, Ghosh et al. (2025) show that when deep linear networks are trained with very large learning rates, gradient descent operates in a so-called beyond–EoS regime characterized by sustained oscillations around the balanced minimum which is of minimum sharpness. In contrast, we only consider converged trajectories, not ones which are in stable oscillations. Finally, we highlight that for models with normalization layers, the sharpness scales inversely with the squared parameter norm (Li et al., 2021; Lyu et al., 2022). Although this corresponds to a different GD dynamics due to the explicit regularization, the resulting trade-off aligns with our main observation.

**Sharpness and Generalization.** In the past, various notions of sharpness have been studied in connection to generalization. The idea that flat minima benefit generalization dates back to Wolpert (1993), who argued this from a minimal description length perspective. Later, Hochreiter & Schmidhuber (1994; 1997) proposed an algorithm designed to locate flat minima, defining them as "large regions of connected acceptable minima," where an acceptable minimum is any point with empirical mean squared error below a certain threshold. Notably, their formulation does not explicitly involve the Hessian. Following these early works, many authors have conjectured that flatter solutions should generalize better (Xing et al., 2018; Zhou et al., 2020; Park & Kim, 2022; Lyu et al., 2022). The prevailing intuition is that solutions lying in flatter regions of the loss landscape are more robust to perturbations (Keskar et al., 2017), which may contribute to improved generalization.

Inspired by this idea, sharpness-aware minimization (SAM) has been proposed by Foret et al. (2021) as an explicit regularization method that penalizes sharpness, successfully applied in improving model generalization on benchmark datasets such as CIFAR-10 and CIFAR-100. In Tahmasebi et al. (2024), SAM was extended to sharpness measures that are general functions of the (spectrum of the) Hessian of the loss. The general sharpness formulation presented therein encompasses various common notions of sharpness such as worst-case and average-case sharpness.

Despite these theoretical and empirical arguments, the relationship between flatness and generalization remains disputed (Andriushchenko & Flammarion, 2022). Studies have found little correlation between sharpness and generalization performance (Jiang et al., 2020; Kaur et al., 2023). Furthermore, a re-parameterization argument by Dinh et al. (2017) shows that sharpness measures such as $S_{\mathcal{L}}$ can be made arbitrarily large without affecting generalization, challenging the notion that flatness is a necessary condition for good performance. Even when using scaling invariant sharpness measures like *adaptive sharpness* (Kwon et al., 2021), the empirical studies performed by Andriushchenko et al. (2023a) show that there is no notable correlation between low sharpness and good generalization. On the contrary, in various cases the correlation is negative, i.e., sharper minima generalize better. What is most interesting about the latter work from our perspective, is that it observes correlation of generalization with parameters such as the learning rate, which agrees with the herein presented idea of an implicit bias trade-off that is governed by hyperparameters of GD.

**Generalization and Norm.** A possible explanation for the occasionally observed correlation between flatness and generalization can be deduced from Ding et al. (2024). Therein the authors show for (overparameterized) matrix factorization of $\mathbf{X}_{\star} \in \mathbb{R}^{d_1 \times d_2}$ via

$$\min_{\mathbf{U} \in \mathbb{R}^{d_1 \times k}, \mathbf{V} \in \mathbb{R}^{d_2 \times k}} \|\mathbf{U}\mathbf{V}^T - \mathbf{X}_{\star}\|_F^2,$$

where $k \geq \mathrm{rank}(\mathbf{X}_{\star})$ is arbitrarily large, that sharpness and nuclear norm ($\ell_1$-norm on the spectrum) minimizers coincide. For (overparameterized) matrix regression

$$\min_{\mathbf{U} \in \mathbb{R}^{d_1 \times k}, \mathbf{V} \in \mathbb{R}^{d_2 \times k}} \|\mathcal{A}(\mathbf{U}\mathbf{V}^T) - \mathbf{y}\|_2^2, \tag{8}$$

where $\mathbf{y} = \mathcal{A}(\mathbf{X}_{\star}) + \mathbf{e}$, for $\mathcal{A} \colon \mathbb{R}^{d_1 \times d_2} \to \mathbb{R}^m$ and unknown noise $\mathbf{e} \in \mathbb{R}^m$, they relate the distance between sharpness and nuclear norm minimizers to how close the measurement operator $\mathcal{A}$ is to identity. Good generalization of a solution $(\hat{\mathbf{U}}, \hat{\mathbf{V}})$ of (8), i.e., $\hat{\mathbf{U}}\hat{\mathbf{V}}^T \approx \mathbf{X}_{\star}$, is then proved if $\mathcal{A}$ satisfies an appropriate *restricted isometry property (RIP)* for low-rank matrices. However, it is not really clear which of the two types of regularization explains the generalization. In view of the well-established theory of sparse resp. low-rank recovery via $\ell_1$- resp. nuclear norm minimization (Foucart & Rauhut, 2013), one may assume in this specific setting that good generalization of flat minima is just a consequence of the fact that flat minima lie close to nuclear norm minimizers, which provably generalize well in low-rank recovery. This spectral perspective is further supported by Bartlett et al. (2017). The observation that a single bias causes generalization might only stem from special situations in which several independent biases agree. This is also the case in scalar factorization Wang et al. (2022a, Appendix F.2.), where the sharpness of a minimizer is equal to squared norm and the biases thus coincide. This point of view is supported by Wen et al. (2023) and aligns with our observations.

## B. Implicit Norm and Sharpness Regularization

In this section, we recall two established results on implicit bias of GF and GD. In the setting of Section 3, it is known that GF converges to an end-to-end model $\mathbf{w}_{\star}^{\odot 2}$ that approximately minimizes a weighted $\ell_1$-norm among all interpolating solutions $\phi_{\mathbf{w}}(\mathbf{x}) = y$ if initialized close to the origin (Chou et al., 2023) where the weights of the $\ell_1$-norm depend on the chosen initialization. To avoid unnecessary technicalities, we formulate the result only for $\mathbf{w}_0 = \alpha \mathbf{1}$ which induces a bias towards the unweighted $\ell_1$-norm.

**Theorem B.1** (Implicit $\ell_1$-bias of GF (Chou et al., 2023)). *Let $\mathcal{L}$ be defined as in (4) with $\mathcal{M}$ as in (5). Assume that $\mathcal{M} \cap \mathbb{R}_{\geq 0}^d$ is non-empty and GF is applied with $\mathbf{w}_0 = \alpha\mathbf{1}$, for $\alpha > 0$. Then, GF converges to $\mathbf{w}_\infty \in \mathbb{R}^d$ with*

$$\|\mathbf{w}_\infty^{\odot 2}\|_1 \leq \left( \min_{\mathbf{w} \in \mathcal{M} \cap \mathbb{R}_{\geq 0}^d} \|\mathbf{w}^{\odot 2}\|_1 \right) + \varepsilon(\alpha),$$

*where $\varepsilon(\alpha) > 0$ satisfies $\varepsilon(\alpha) \searrow 0$, for $\alpha \to 0$.*[5]

The implicit sharpness regularization of GD for large learning rates can be deduced from the following result.

**Theorem B.2** (Dynamic stability of GD (Ahn et al., 2022)). *Let $\eta > 0$ and $X \subset \mathbb{R}^p$. Let $\mathcal{L}$ be twice continuously differentiable such that the operator $F \colon \mathbb{R}^p \to \mathbb{R}^p$, $F(w) = w - \eta\nabla\mathcal{L}(w)$ satisfies that $F^{-1}(S)$ is a set of Lebesgue-measure zero, for any set $S \subset \mathbb{R}^p$ of measure zero. Assume furthermore that $\frac{1}{\eta}$ is not an eigenvalue of $\nabla^2\mathcal{L}(w_\star)$ for every stationary point $w_\star$ of $\mathcal{L}$. Let $w_k$ be the iterates of GD with learning rate $\eta$. If $\|\nabla^2\mathcal{L}(w)\|_2 > 2/\eta$ for every $w \in X$, then there exists a zero Lebesgue measure set $A_X$ such that*

- *either $w_0 \in A_X$*

- *or $w_k$ does not converge to any $w \in X$.*

## C. Proof of Lemma 3.1

Lemma 3.1 is a special case of the following result for training diagonal linear $L$-layer networks with shared weights on a single data point. In this case, the loss $\mathcal{L}$ is given by

$$\mathcal{L}(\mathbf{w}) = \frac{1}{2}(\langle \mathbf{x}, \mathbf{w}^{\odot L} \rangle - y)^2. \tag{9}$$

**Lemma C.1.** *For $\mathcal{L}$ as in (9), define $\mathcal{M}$ as in (5). If $\mathbf{x} \in \mathbb{R}_{\neq 0}^n$ and $y \neq 0$, then $\mathcal{M}$ is a Riemannian manifold with tangent space $T_\mathbf{w}\mathcal{M} = (\mathbf{x} \odot \mathbf{w}^{\odot(L-1)})^\perp$ at $\mathbf{w} \in \mathcal{M}$.*

*Proof.* Note that $\mathbf{w} \in \mathcal{M}$ is equivalent to

$$h(\mathbf{w}) := \langle \mathbf{x}, \mathbf{w}^{\odot L} \rangle - y = 0,$$

where $h \colon \mathbb{R}^d \to \mathbb{R}$. Since $Dh(\mathbf{w}) = L(\mathbf{x} \odot \mathbf{w}^{\odot L-1})^T$ and $\mathbf{w} \neq \mathbf{0}$ for any $\mathbf{w} \in \mathcal{M}$ due to $y \neq 0$, we have that $\mathrm{rank}(Dh(\mathbf{w})) = 1$ for all $\mathbf{w} \in \mathcal{M}$. Hence, $\mathcal{M}$ is a $(d-1)$-dimensional submanifold in $\mathbb{R}^d$ with tangent spaces

$$T_\mathbf{w}\mathcal{M} = \ker(Dh(\mathbf{w})) = (\mathbf{x} \odot \mathbf{w}^{L-1})^\perp,$$

e.g., see Boumal (2023). Smoothness of the manifold follows by equipping $T_\mathbf{w}\mathcal{M}$ with the Euclidean metric of $\mathbb{R}^d$. $\square$

## D. Proof of Proposition 3.2

Before we prove Proposition 3.2, we note that the $\ell_1$-norm of $\mathbf{w}^{\odot 2}$ can be written as

$$\|\mathbf{w}^{\odot 2}\|_1 = \|\mathbf{w}\|_2^2 \tag{10}$$

and that the sharpness $S_\mathcal{L}(\mathbf{w})$ of $\mathcal{L}$ at $\mathbf{w}$ satisfies

$$S_\mathcal{L}(\mathbf{w}) = 4\|\mathbf{x} \odot \mathbf{w}\|_2^2, \tag{11}$$

for any $\mathbf{w} \in \mathcal{M}$, where we used that

$$\nabla^2\mathcal{L}(\mathbf{w}) = \mathbf{D}_{2(\langle \mathbf{x}, \mathbf{w}^{\odot 2}\rangle - y)\cdot\mathbf{x}} + 4(\mathbf{x} \odot \mathbf{w})(\mathbf{x} \odot \mathbf{w})^T. \tag{12}$$

The necessary conditions of Proposition 3.2 are proven in the following lemma.

---

[5]Note that the restriction of Theorem B.1 to non-negative parameters is not limiting the analysis since (6) always contains such solutions, i.e., in our setting an $\ell_1$-minimizer on $\mathcal{M} \cap \mathbb{R}_{\geq 0}^d$ is also a minimizer on $\mathcal{M}$.

**Lemma D.1.** *For* $\mathbf{x} \in \mathbb{R}^d_{\neq 0}$ *and* $\mathcal{L}$ *as in* (4) *with* $\mathcal{M}$ *as in* (5)*, the following hold:*

*(i) To have*

$$\mathbf{w} \in \arg\min_{\mathbf{z} \in \mathcal{M}} \|\mathbf{z}^{\odot 2}\|_1,$$

*it is necessary that* $\mathbf{x}|_{\text{supp}(\mathbf{w})} = x_0 \cdot \mathbf{1}|_{\text{supp}(\mathbf{w})}$, *for* $x_0 = \max_i |x_i|$.

*(ii) To have*

$$\mathbf{w} \in \arg\min_{\mathbf{z} \in \mathcal{M}} S_{\mathcal{L}}(\mathbf{z}),$$

*it is necessary that* $\mathbf{x}|_{\text{supp}(\mathbf{w})} = x_0 \cdot \mathbf{1}|_{\text{supp}(\mathbf{w})}$, *for some* $x_0 \in \mathbb{R}$. *Furthermore, if* $\mathbf{x} \in \mathbb{R}^d_{>0}$, *it is additionally necessary that* $x_0 = \min_i x_i$.

*Proof.* In the proof we compute the Riemannian gradient $\text{grad} f$ and the Riemannian Hessian $\text{Hess} f$ of a function $f$ on $\mathcal{M}$. Note that

$$\text{grad} f(\mathbf{w}) = \mathbb{P}_{T_\mathbf{w}\mathcal{M}} \nabla f(\mathbf{w})$$

and

$$[\text{Hess} f(\mathbf{w})](\mathbf{u}) = \mathbb{P}_{T_\mathbf{w}\mathcal{M}}([\nabla \text{grad} f(\mathbf{w})](\mathbf{u})),$$

for any $\mathbf{w} \in \mathcal{M}$ and $\mathbf{u} \in T_\mathbf{w}\mathcal{M}$, where $\mathbb{P}_U$ denotes the orthogonal projection onto the linear subspace $U \subset \mathbb{R}^d$ (Boumal, 2023).

We begin with $(i)$. Define $f(\mathbf{w}) = \frac{1}{2}\mathbf{w}^T\mathbf{w}$ and note that $f(\mathbf{w}) = \frac{1}{2}\|\mathbf{w}^{\odot 2}\|_1$ by (10). Hence,

$$\text{grad} f(\mathbf{w}) = \mathbb{P}_{T_\mathbf{w}\mathcal{M}} \nabla f(\mathbf{w}) = \mathbf{w} - \frac{1}{\|\mathbf{D}_\mathbf{x}\mathbf{w}\|_2^2} \mathbf{D}_\mathbf{x}\mathbf{w}\mathbf{w}^T\mathbf{D}_\mathbf{x} \cdot \mathbf{w}.$$

To have $\text{grad} f(\mathbf{w}) = \mathbf{0}$, $\mathbf{w}$ has to be an eigenvector of $\mathbf{D}_\mathbf{x}\mathbf{w}\mathbf{w}^T\mathbf{D}_\mathbf{x}$ with eigenvalue $\|\mathbf{D}_\mathbf{x}\mathbf{w}\|_2^2$ which is equivalent to $\mathbf{x}|_{\text{supp}(\mathbf{w})} = x_0 \cdot \mathbf{1}|_{\text{supp}(\mathbf{w})}$, for some $x_0 \in \mathbb{R}$. This is the first necessary condition.

Now define $G(\mathbf{w}) = \text{grad} f(\mathbf{w})$. Then,

$$[\nabla G(\mathbf{w})]_{ij} = \partial_j G(\mathbf{w})_i$$
$$= \begin{cases} \frac{2}{\|\mathbf{D}_\mathbf{x}\mathbf{w}\|_2^4} \cdot x_j^2 w_j \cdot x_i w_i \langle \mathbf{w}, \mathbf{D}_\mathbf{x}\mathbf{w}\rangle - \frac{2}{\|\mathbf{D}_\mathbf{x}\mathbf{w}\|_2^2} \cdot x_i x_j w_i w_j & i \neq j, \\ 1 - \frac{1}{\|\mathbf{D}_\mathbf{x}\mathbf{w}\|_2^2} \cdot (x_i\langle\mathbf{w},\mathbf{D}_\mathbf{x}\mathbf{w}\rangle + 2x_i^2 w_i^2) + \frac{2}{\|\mathbf{D}_\mathbf{x}\mathbf{w}\|_2^4}x_i^2 w_i \cdot x_i w_i \langle\mathbf{w},\mathbf{D}_\mathbf{x}\mathbf{w}\rangle & i = j, \end{cases}$$

such that

$$\nabla G(\mathbf{w}) = \mathbf{D}_{\mathbf{1} - \frac{\langle\mathbf{w},\mathbf{D}_\mathbf{x}\mathbf{w}\rangle}{\|\mathbf{D}_\mathbf{x}\mathbf{w}\|_2^2} \cdot \mathbf{x}} - \frac{2}{\|\mathbf{D}_\mathbf{x}\mathbf{w}\|_2^2} \mathbf{D}_\mathbf{x}\mathbf{w}\mathbf{w}^T\mathbf{D}_\mathbf{x} + \frac{2\langle\mathbf{w},\mathbf{D}_\mathbf{x}\mathbf{w}\rangle}{\|\mathbf{D}_\mathbf{x}\mathbf{w}\|_2^4}\mathbf{D}_\mathbf{x}\mathbf{w}\mathbf{w}^T\mathbf{D}_\mathbf{x}^2.$$

Consequently, we have that

$$[\text{Hess} f(\mathbf{w})](\mathbf{u}) = \mathbb{P}_{T_\mathbf{w}\mathcal{M}}([\nabla G(\mathbf{w})](\mathbf{u}))$$
$$= (\mathbf{I} - \frac{1}{\|\mathbf{D}_\mathbf{x}\mathbf{w}\|_2^2}\mathbf{D}_\mathbf{x}\mathbf{w}\mathbf{w}^T\mathbf{D}_\mathbf{x}) \cdot \left[(\mathbf{1} - \frac{\langle\mathbf{w},\mathbf{D}_\mathbf{x}\mathbf{w}\rangle}{\|\mathbf{D}_\mathbf{x}\mathbf{w}\|_2^2} \cdot \mathbf{x}) \odot \mathbf{u}\right].$$

For any $\mathbf{w}$ satisfying the first necessary condition, we thus have that

$$\langle\mathbf{u}, [\text{Hess} f(\mathbf{w})](\mathbf{u})\rangle = \mathbf{u}^T \cdot (\mathbf{I} - \frac{\mathbf{w}\mathbf{w}^T}{\|\mathbf{w}\|_2^2}) \cdot (\mathbf{1} - \frac{\mathbf{x}}{x_0}) \odot \mathbf{u} = \|\mathbf{u}\|_2^2 - \langle\mathbf{u}, \frac{\mathbf{x}}{x_0} \odot \mathbf{u}\rangle,$$

where we used in the second equality that $\mathbf{x}|_{\mathrm{supp}(\mathbf{w})} = x_0 \cdot \mathbf{1}|_{\mathrm{supp}(\mathbf{w})}$ by which $(\mathbf{1} - \frac{\mathbf{x}}{x_0})|_{\mathrm{supp}(\mathbf{w})} = \mathbf{0}$. Hence, $\langle \mathbf{u}, [\mathrm{Hess} f(\mathbf{w})](\mathbf{u}) \rangle \geq 0$ can only hold for all $\mathbf{u} \in T_{\mathbf{w}} \mathcal{M}$ if $x_0 = \arg\max_i |x_i|$.

To show $(ii)$, we proceed analogously but consider $f(\mathbf{w}) = \frac{1}{2} \mathbf{D}_{\mathbf{x}} \mathbf{w}^T \mathbf{w} \mathbf{D}_{\mathbf{x}}$, and note that $f(\mathbf{w}) = \frac{1}{8} S_{\mathcal{L}}(\mathbf{w})$ by (11). Then, one can easily check that

$$\mathrm{grad} f(\mathbf{w}) = \mathbf{D}_{\mathbf{x}}^2 \mathbf{w} - \frac{1}{\|\mathbf{D}_{\mathbf{x}} \mathbf{w}\|_2^2} \mathbf{D}_{\mathbf{x}} \mathbf{w} \mathbf{w}^T \mathbf{D}_{\mathbf{x}}^3 \cdot \mathbf{w},$$

which implies the same first necessary condition. Now assume $\mathbf{x} \in \mathbb{R}_{>0}^d$. Then,

$$\nabla^2 G(\mathbf{w}) = \mathbf{D}_{\mathbf{x}^{\odot 2} - \frac{\langle \mathbf{w}, \mathbf{D}_{\mathbf{x}}^3 \mathbf{w} \rangle}{\|\mathbf{D}_{\mathbf{x}} \mathbf{w}\|_2^2} \cdot \mathbf{x}} - \frac{2}{\|\mathbf{D}_{\mathbf{x}} \mathbf{w}\|_2^2} \mathbf{D}_{\mathbf{x}} \mathbf{w} \mathbf{w}^T \mathbf{D}_{\mathbf{x}}^3 + \frac{2\langle \mathbf{w}, \mathbf{D}_{\mathbf{x}}^3 \mathbf{w} \rangle}{\|\mathbf{D}_{\mathbf{x}} \mathbf{w}\|_2^4} \mathbf{D}_{\mathbf{x}} \mathbf{w} \mathbf{w}^T \mathbf{D}_{\mathbf{x}}^2,$$

such that

$$[\mathrm{Hess} f(\mathbf{w})](\mathbf{u}) = \left( \mathbf{I} - \frac{1}{\|\mathbf{D}_{\mathbf{x}} \mathbf{w}\|_2^2} \mathbf{D}_{\mathbf{x}} \mathbf{w} \mathbf{w}^T \mathbf{D}_{\mathbf{x}} \right) \cdot \left( \mathbf{x}^{\odot 2} - \frac{\langle \mathbf{w}, \mathbf{D}_{\mathbf{x}}^3 \mathbf{w} \rangle}{\|\mathbf{D}_{\mathbf{x}} \mathbf{w}\|_2^2} \cdot \mathbf{x} \right) \odot \mathbf{u}.$$

For any $\mathbf{w}$ satisfying the first necessary condition, we thus have that

$$\langle \mathbf{u}, [\mathrm{Hess} f(\mathbf{w})](\mathbf{u}) \rangle = \langle \mathbf{u}, \mathbf{D}_{\mathbf{x}}^2 \mathbf{u} \rangle - x_0 \langle \mathbf{u}, \mathbf{D}_{\mathbf{x}} \mathbf{u} \rangle$$

which implies for $\mathbf{x} \in \mathbb{R}_{>0}^d$ that $\langle \mathbf{u}, [\mathrm{Hess} f(\mathbf{w})](\mathbf{u}) \rangle \geq 0$ can only hold for all $\mathbf{u} \in T_{\mathbf{w}} \mathcal{M}$ if $x_0 = \arg\min_i x_i$. $\quad\square$

The sufficient conditions are stated in the following lemma.

**Lemma D.2.** *For $\mathbf{x} \in \mathbb{R}_{>0}^d$ and $\mathcal{L}$ as in* (4) *with $\mathcal{M}$ as in* (5), *we have the following:*

*(i) To have*

$$\mathbf{w} \in \arg\min_{\mathbf{z} \in \mathcal{M}} \|\mathbf{z}^{\odot 2}\|_1,$$

*it is sufficient for $\mathbf{w} \in \mathcal{M}$ that $\mathrm{supp}(\mathbf{w}) \subset \arg\max_k x_k$.*

*(ii) To have*

$$\mathbf{w} \in \arg\min_{\mathbf{z} \in \mathcal{M}} S_{\mathcal{L}}(\mathbf{z}),$$

*it is sufficient for $\mathbf{w} \in \mathcal{M}$ that $\mathrm{supp}(\mathbf{w}) \subset \arg\min_k x_k$.*

*Proof.* First recall (10) and (11). We begin with $(i)$. Let $k_* \in \arg\max_k x_k$. Since $\|\mathbf{w}\|_2^2 < y/x_{k_*}$ implies by our assumption on $\mathbf{x}$ that $\langle \mathbf{x}, \mathbf{w}^{\odot 2} \rangle \leq x_{k_*} \|\mathbf{w}\|_2^2 < y$, i.e., $\mathbf{w} \notin \mathcal{M}$, and

$$\sqrt{\frac{y}{x_{k_*}}} \mathbf{e}_{k_*} \in \mathcal{M} \quad \text{satisfies} \quad \left\| \sqrt{\frac{y}{x_{k_*}}} \mathbf{e}_{k_*} \right\|_2^2 = \frac{y}{x_{k_*}},$$

we know by (10) that

$$\min_{\mathbf{z} \in \mathcal{M}} \|\mathbf{z}^{\odot 2}\|_1 = \frac{y}{x_{k_*}}.$$

For any $\mathbf{w} \in \mathcal{M}$ with $\mathrm{supp}(\mathbf{w}) \subset \arg\max_k x_k$, we have that

$$y = \langle \mathbf{x}, \mathbf{w}^{\odot 2} \rangle = x_{k_*} \|\mathbf{w}\|_2^2 = x_{k_*} \|\mathbf{w}^{\odot 2}\|_1$$

and the claim in $(i)$ follows.

To see $(ii)$ we proceed analogously. Let $k_* \in \arg\min_k x_k$. Since $\|\mathbf{D_x w}\|_2^2 < yx_{k_*}$ implies by our assumption on $\mathbf{x}$ that $\langle \mathbf{x}, \mathbf{w}^{\odot 2} \rangle \leq \frac{1}{x_{k_*}} \|\mathbf{D_x w}\|_2^2 < y$, i.e., $\mathbf{w} \notin \mathcal{M}$, and

$$\sqrt{\frac{y}{x_{k_*}}} \mathbf{e}_{k_*} \in \mathcal{M} \quad \text{satisfies} \quad \left\| \mathbf{D_x} \cdot \sqrt{\frac{y}{x_{k_*}}} \mathbf{e}_{k_*} \right\|_2^2 = yx_{k_*},$$

we know by (11) that

$$\min_{\mathbf{z} \in \mathcal{M}} S_{\mathcal{L}}(\mathbf{z}) = yx_{k_*}.$$

For any $\mathbf{w} \in \mathcal{M}$ with $\text{supp}(\mathbf{w}) \subset \arg\min_k x_k$, we have that

$$y = \langle \mathbf{x}, \mathbf{w}^{\odot 2} \rangle = x_{k_*} \|\mathbf{w}\|_2^2 = \frac{1}{x_{k_*}} S_{\mathcal{L}}(\mathbf{w})$$

and the claim in $(ii)$ follows. □

The specific shape of the minimizing sets (6) and (7) can easily be derived from the previous two lemmas.

## E. Generalizing Proposition 3.2 to Multiple Data Points

In this section, we detail our claims in Remark 3.3. For multiple data points, (4) becomes

$$\min_{\mathbf{w} \in \mathbb{R}^d} \mathcal{L}(\mathbf{w}) = \min_{\mathbf{w} \in \mathbb{R}^d} \frac{1}{8} \|\mathbf{X w}^{\odot 2} - \mathbf{y}\|_2^2, \tag{13}$$

where the rows of $\mathbf{X} \in \mathbb{R}^{m \times d}$ contain $m$ data points $\mathbf{x}_i \in \mathbb{R}^d$ and, for notational convenience, we changed the factor $\frac{1}{2}$ to $\frac{1}{8}$ to normalize the Hessian. The following lemma generalizes (12) to this setting.

**Lemma E.1** (Hessian for multiple data points). *Let $\mathbf{X} \in \mathbb{R}^{m \times d}$, $\mathbf{y} \in \mathbb{R}^m$, and $L \in \mathbb{N}$ with $L \geq 2$. Then the loss function*

$$\mathcal{L}(\mathbf{w}) := \frac{1}{2L^2} \|\mathbf{X w}^{\odot L} - \mathbf{y}\|_2^2 \tag{14}$$

*has a Hessian of the form*

$$\mathbf{H}(\mathbf{w}) := \nabla^2 \mathcal{L}(\mathbf{w}) = \frac{L-1}{L} \cdot \mathbf{H}_1(\mathbf{w}) + \mathbf{H}_2(\mathbf{w}). \tag{15}$$

*The matrix $\mathbf{H}_1$ is related to the gradient of $\mathcal{L}$ in the following way*

$$\mathbf{H}_1(\mathbf{w}) = \mathbf{D}_{\mathbf{w}}^{L-2} \mathbf{D}_{\mathbf{X}^\top (\mathbf{X w}^{\odot L} - \mathbf{y})}, \tag{16}$$

*while $\mathbf{H}_2$ is a symmetric positive semi-definite matrix given by*

$$\mathbf{H}_2(\mathbf{w}) = \mathbf{D}_{\mathbf{w}}^{L-1} \mathbf{X}^\top \mathbf{X} \mathbf{D}_{\mathbf{w}}^{L-1}. \tag{17}$$

*Here, $\mathbf{D_z}$ is the diagonal matrix whose diagonal equals to the vector $\mathbf{z}$.*

*Proof.* The gradient and Hessian of $\mathcal{L}$ are given by

$$\nabla \mathcal{L}(\mathbf{w}) = \frac{1}{L} \cdot \mathbf{D}_{\mathbf{w}}^{L-1} \mathbf{X}^\top (\mathbf{X w}^{\odot L} - \mathbf{y}) \tag{18}$$

and

$$\nabla^2 \mathcal{L}(\mathbf{w}) = \frac{L-1}{L} \cdot \underbrace{\mathbf{D}_{\mathbf{w}}^{L-2} \mathbf{D}_{\mathbf{X}^\top (\mathbf{X w}^{\odot L} - \mathbf{y})}}_{=: \mathbf{H}_1(\mathbf{w})} + \underbrace{\mathbf{D}_{\mathbf{w}}^{L-1} \mathbf{X}^\top \mathbf{X} \mathbf{D}_{\mathbf{w}}^{L-1}}_{=: \mathbf{H}_2(\mathbf{w})}$$

This completes the proof. □

Whereas for general $\mathbf{X}$ there does not exist a simple relation between $S_{\mathcal{L}}$ and a norm of $\mathbf{w}$, one can (approximately) obtain such a relation if $\mathbf{X}$ is assumed to be Gaussian. The following result generalizes the observations of Andriushchenko et al. (2023a) to non-whitened Gaussian data.

**Theorem E.2** (Concentration of sharpness). *Let $\mathbf{X}$ be a random matrix in $\mathbb{R}^{m \times d}$ with independent mean-zero Gaussian entries with unit variance, let $\mathbf{y} \in \mathbb{R}^m$ be fixed, and consider the loss function in* (14). *For any data interpolating solution $\mathbf{w}$ with $\mathcal{L}(\mathbf{w}) = 0$, the sharpness $\|\nabla^2 \mathcal{L}(\mathbf{w})\|_2$ is a sub-exponential variable satisfying*

$$\|\|\nabla^2 \mathcal{L}(\mathbf{w})\| - \mathbb{E}\|\nabla^2 \mathcal{L}(\mathbf{w})\|\|_{\psi_1} \lesssim (\sqrt{m} + \sqrt{d} + 1)\|\mathbf{w}^{\odot(L-1)}\|_\infty^2 \tag{19}$$

*and*

$$m \cdot \|\mathbf{w}^{\odot(L-1)}\|_\infty^2 \leq \mathbb{E}\|\nabla^2 \mathcal{L}(\mathbf{w})\| \lesssim (\sqrt{m} + \sqrt{d})^2 \cdot \|\mathbf{w}^{\odot(L-1)}\|_\infty^2. \tag{20}$$

Before we provide the proof of Theorem E.2, let us discuss its implications. For $m > 1$, the solution manifold $\mathcal{M}$ of (13) is an intersection of different ellipsoids. It is thus hard to visualize and analyze the $\ell_2$-parameter-norm and sharpness minimizers as detailed as in the single sample setting.

However, we can consider generic data points $\mathbf{x}_1, \ldots, \mathbf{x}_m \in \mathbb{R}^d$ with (approximate) unit norm. These could be modeled by a properly normalized Gaussian matrix $\frac{1}{\sqrt{d}}\mathbf{X} \in \mathbb{R}^{m \times d}$. If we consider the overparameterized regime $m = cd$, for $c \in (0, 1)$ and set $L = 2$ for convenience, Theorem E.2 shows in this case that $S_{\mathcal{L}}(\mathbf{w})$ concentrates around its mean $\mathbb{E} \, S_{\mathcal{L}}(\mathbf{w}) \simeq \|\mathbf{w}^{\odot 2}\|_\infty^2$ with deviation $|S_{\mathcal{L}}(\mathbf{w}) - \mathbb{E} \, S_{\mathcal{L}}(\mathbf{w})| = \mathcal{O}(1/\sqrt{m})$.

Considering the results in Appendix B, we thus know that on (13) GD with small learning rate will converge to interpolating parameters (approximately) minimizing $\|\mathbf{w}^{\odot 2}\|_1$, while GD with large learning rate will converge to interpolating parameters (approximately) minimizing $\|\mathbf{w}^{\odot 2}\|_\infty$. Due to the fundamentally different geometry of $\ell_1$- and $\ell_\infty$-ball, these (approximately) minimal norm solutions of the linear system $\mathbf{X}\mathbf{w}^{\odot 2} = \mathbf{y}$ are clearly separated in space, for $\mathbf{y} \neq 0$. We thus argue that our toy analysis of a single data point in Section 3 is representative for the multi-sample case as well, although many quantities such as $\mathcal{M}_{\ell_1}$, $S_{\mathcal{L}}$, etc. have no simple closed-form expressions anymore.

*Proof of Theorem E.2.* Recall Lemma E.1. For $\mathcal{L}(\mathbf{w}) = 0$ and $L \geq 2$, we have $\mathbf{H}_1 = 0$ and hence

$$\mathbf{H}(\mathbf{w}) := \nabla^2 \mathcal{L}(\mathbf{w}) = \mathbf{D}_{\mathbf{w}}^{L-1} \mathbf{X}^\top \mathbf{X} \mathbf{D}_{\mathbf{w}}^{L-1}. \tag{21}$$

For the sake of conciseness, we will abbreviate $\mathbf{D}_{\mathbf{w}}^{L-1}$ by $\mathbf{D}$ in the following calculations, and define $S := \|\mathbf{X}\mathbf{D}\|$ so that $S^2 = \|\mathbf{X}\mathbf{D}\|^2 = \|\mathbf{D}\mathbf{X}^\top \mathbf{X}\mathbf{D}\| = \|\mathbf{H}\|$. We first bound $\mathbb{E}S^2$ and then prove that $S$ is sub-gaussian, which implies that $S^2$ is sub-exponential.

Next, we show the upper and lower bound of the expectation. For the upper bound

$$\mathbb{E}S^2 = \mathbb{E}\|\mathbf{X}\mathbf{D}\|^2 \leq \mathbb{E}\|\mathbf{X}\|^2 \cdot \|\mathbf{D}\|^2 \lesssim (\sqrt{m} + \sqrt{d})^2 \cdot \|\mathbf{D}\|^2 \tag{22}$$

follows from basic properties of Gaussian random matrices (Vershynin, 2018). For the lower bound, let $\mathbf{x}^{(i)} \in \mathbb{R}^m$ denote the $i$-th column of $\mathbf{X}$, $\mathbf{e}_i$ the $i$-th canonical basis vector, and $j \in \arg\max_i |D_{ii}|$ so that $|D_{jj}| = \|\mathbf{D}\|$. Then

$$\|\mathbf{X}\mathbf{D}\| \geq \|\mathbf{X}\mathbf{D}\mathbf{e}_j\|_2 = |D_{jj}| \cdot \|\mathbf{x}^{(j)}\|_2 = \|\mathbf{D}\| \cdot \|\mathbf{g}\|_2$$

where $\mathbf{g} \sim \mathcal{N}(0, \mathbf{I})$. Hence

$$\mathbb{E}S^2 \geq \|\mathbf{D}\|^2 \cdot \mathbb{E}\|\mathbf{g}\|_2^2 = m \cdot \|\mathbf{D}\|^2 \tag{23}$$

Altogether we have

$$m \cdot \|\mathbf{D}\|^2 \leq \mathbb{E}S^2 \lesssim (\sqrt{m} + \sqrt{d})^2 \cdot \|\mathbf{D}\|^2. \tag{24}$$

Next, consider the map $f(\mathbf{X}) = \|\mathbf{X}\mathbf{D}\|_2$. Note that $f$ is $\|\mathbf{D}\|_2$-Lipschitz since

$$|f(\mathbf{X}) - f(\mathbf{X}')| \leq \|(\mathbf{X} - \mathbf{X}')\mathbf{D}\|_F \leq \|\mathbf{X} - \mathbf{X}'\|_F \|\mathbf{D}\| = \|\mathbf{D}\|\|\text{vec}(\mathbf{X}) - \text{vec}(\mathbf{X}')\|_2.$$

By concentration of Gaussian random vectors over Lipschitz functions, $\|f(\mathbf{X}) - \mathbb{E}f(\mathbf{X})\|_{\psi_2} \lesssim \|\mathbf{D}\|$ (Vershynin, 2018). Hence $S$ is sub-gaussian. Since subtracting a constant preserves Lipschitzness, we also have for $\Delta := S - \mathbb{E}S$ that $\|\Delta\|_{\psi_2} \lesssim \|\mathbf{D}\|$, i.e., $\Delta$ is sub-gaussian.

Using $S = \Delta + \mathbb{E}S$, we can write

$$S^2 - \mathbb{E}S^2 = (\Delta^2 - \mathbb{E}\Delta^2) + 2(\mathbb{E}S)\Delta.$$

Consequently, by the triangle inequality and the facts that (i) $\Delta^2 - \mathbb{E}\Delta^2$ is sub-exponential with $\|\Delta^2 - \mathbb{E}\Delta^2\|_{\psi_1} \lesssim \|\Delta\|_{\psi_2}^2$ and (ii) $\|\Delta\|_{\psi_1} \lesssim \|\Delta\|_{\psi_2}$, we obtain

$$
\begin{aligned}
\|S^2 - \mathbb{E}S^2\|_{\psi_1} &\leq \|\Delta^2 - \mathbb{E}\Delta^2\|_{\psi_1} + \|2(\mathbb{E}S)\Delta\|_{\psi_1} \\
&\lesssim \|\Delta\|_{\psi_2}^2 + 2(\mathbb{E}S)\|\Delta\|_{\psi_2} \lesssim \|\mathbf{D}\|^2 + 2(\mathbb{E}S)\|\mathbf{D}\|.
\end{aligned}
\tag{25}
$$

In particular, $S^2 - \mathbb{E}S^2$ is sub-exponential.

Combining (25) with

$$\mathbb{E}S = \mathbb{E}\|\mathbf{XD}\| \leq \mathbb{E}\|\mathbf{X}\| \cdot \|\mathbf{D}\| \leq (\sqrt{m} + \sqrt{d}) \cdot \|\mathbf{D}\|$$

Vershynin (2018) yields

$$\|S^2 - \mathbb{E}S^2\|_{\psi_1} \lesssim \big(1 + 2(\sqrt{m} + \sqrt{d})\big)\|\mathbf{D}\|^2 \lesssim (\sqrt{m} + \sqrt{d} + 1)\|\mathbf{D}\|^2. \tag{26}$$

Finally, recall that $S^2 = \|\mathbf{H}\|$ is the sharpness and

$$\|\mathbf{D}\| = \|\mathbf{D}_{\mathbf{w}}^{L-1}\| = \|\mathbf{w}^{\odot(L-1)}\|_\infty. \tag{27}$$

Plugging those back to (24) and (26), we obtain

$$\|S^2 - \mathbb{E}S^2\|_{\psi_1} \lesssim (\sqrt{m} + \sqrt{d} + 1)\|\mathbf{w}^{\odot(L-1)}\|_\infty^2 \tag{28}$$

and

$$m \cdot \|\mathbf{w}^{\odot(L-1)}\|_\infty^2 \leq \mathbb{E}S^2 \lesssim (\sqrt{m} + \sqrt{d})^2 \cdot \|\mathbf{w}^{\odot(L-1)}\|_\infty^2. \tag{29}$$

This concludes the proof. $\qquad\square$

*Remark* E.3 (Normalization of Loss). We state the bounds in Theorem E.2 for the unnormalized quadratic loss, for which $\|\nabla^2\mathcal{L}(\mathbf{w})\|_2$ scales with the operator norm of $\mathbf{X}^\top\mathbf{X}$ and thus grows with the number of data points, $m$. When we consider the *averaged* empirical risk instead

$$\mathcal{L}_{\text{avg}}(\mathbf{w}) := \frac{1}{2L^2 m}\|\mathbf{Xw}^{\odot L} - \mathbf{y}\|_2^2, \tag{30}$$

for any interpolating solution $\mathbf{w}$ with $\mathcal{L}_{\text{avg}}(\mathbf{w}) = 0$, Lemma E.1 implies

$$\nabla^2\mathcal{L}_{\text{avg}}(\mathbf{w}) = \frac{1}{m}\mathbf{D}_{\mathbf{w}}^{L-1}\mathbf{X}^\top\mathbf{X}\mathbf{D}_{\mathbf{w}}^{L-1}. \tag{31}$$

Hence, the corresponding sharpness satisfies

$$\|\nabla^2\mathcal{L}_{\text{avg}}(\mathbf{w})\| = \frac{1}{m}\|\nabla^2\mathcal{L}(\mathbf{w})\|.$$

Using that Orlicz norms scale linearly, Theorem E.2 yields

$$\big\|\|\nabla^2\mathcal{L}_{\text{avg}}(\mathbf{w})\| - \mathbb{E}\|\nabla^2\mathcal{L}_{\text{avg}}(\mathbf{w})\|\big\|_{\psi_1} \lesssim \frac{\sqrt{m} + \sqrt{d} + 1}{m}\|\mathbf{w}^{\odot(L-1)}\|_\infty^2, \tag{32}$$

and

$$\|\mathbf{w}^{\odot(L-1)}\|_\infty^2 \leq \mathbb{E}\|\nabla^2\mathcal{L}_{\text{avg}}(\mathbf{w})\| \lesssim \frac{(\sqrt{m} + \sqrt{d})^2}{m}\|\mathbf{w}^{\odot(L-1)}\|_\infty^2. \tag{33}$$

In particular, for fixed $d$ the variances scaling in (32) is $\mathcal{O}(m^{-1/2})$, i.e., the sharpness concentrates more tightly as the number of samples increases.

# F. An Elementary Study of How Implicit Biases Interact — Generalization

Recalling the setting outlined in Section 3, let us assume that our data follows a simple linear regression model with $\mathbf{x} \sim \mathcal{N}(\mathbf{0}, \mathbf{I})$ and $y = \langle \mathbf{1}, \mathbf{x} \rangle + \varepsilon$, for independent $\varepsilon \sim \mathcal{N}(0, 1)$. Then, the risk under $\mathcal{L}$ can be computed explicitly and, given a single training data point $(\mathbf{x}_0, y_0)$ with $\mathbf{x}_0 \in \mathbb{R}^d_{\geq 0}$, the best achievable generalization error of $\phi_\mathbf{w}$ trained via (4) can be computed as follows.[6]

**Lemma F.1.** *Let $\mathcal{L}$ be as in (4) and let $\mathbf{x} \sim \mathcal{N}(\mathbf{0}, \mathbf{I}_{d \times d})$ and $y = \langle \mathbf{1}, \mathbf{x} \rangle + \varepsilon$, for independent $\varepsilon \sim \mathcal{N}(0, 1)$. Then,*

$$\mathcal{R}(\mathbf{w}) = \mathbb{E}_{(\mathbf{x},y)} \mathcal{L}(\mathbf{w}) = \frac{1}{2} \|\mathbf{w}\|_4^4 - \|\mathbf{w}\|_2^2 + \frac{1}{2}(d+1).$$

*Let now $\eta > 0$ and $(\mathbf{x}_0, y_0) \in \mathbb{R}^d_{\geq 0} \times \mathbb{R}$, and define the corresponding risk minimization under sharpness constraints $S_\mathcal{L}(\mathbf{w}) \leq \frac{2}{\eta}$ as*

$$\min_{\mathbf{w} \in \mathbb{R}^d} \mathcal{R}(\mathbf{w}), \quad s.t. \quad \langle \mathbf{x}_0, \mathbf{w}^{\odot 2} \rangle = y_0, \quad S_\mathcal{L}(\mathbf{w}) \leq \frac{2}{\eta}. \tag{34}$$

*Fix any support $S_w \subset [d]$ with $S_w \cap \operatorname{supp}(\mathbf{x}_0) \neq \emptyset$. Let $\mathbf{w}$ be any vector such that $\operatorname{supp}(\mathbf{w}) = S_w$ and*

$$\mathbf{w}|_{S_w}^{\odot 2} = (\mathbf{1} - 2\lambda\eta\mathbf{x}_0^{\odot 2} - \nu\mathbf{x}_0)|_{S_w},$$

*for $(\lambda, \nu)$ as defined below:*

- *If $S_\mathcal{L}(\mathbf{w}) \leq \frac{2}{\eta}$ and*

$$\lambda = 0$$
$$\nu = \frac{\|\mathbf{x}_0|_{S_w}\|_1 - y_0}{\|\mathbf{x}_0|_{S_w}\|_2^2}$$

  *with $\nu\|\mathbf{x}_0|_{S_w}\|_\infty < 1$, then $\mathbf{w}$ is a KKT point of (34).*

- *If $\mathbf{x}_0 \neq \alpha\mathbf{1}$, for all $\alpha \neq 0$, and*

$$\lambda = \frac{y_0\|\mathbf{x}_0|_{S_w}\|_3^3 + \|\mathbf{x}_0|_{S_w}\|_2^4 - \|\mathbf{x}_0|_{S_w}\|_1\|\mathbf{x}_0|_{S_w}\|_3^3 - \frac{1}{2\eta}\|\mathbf{x}_0|_{S_w}\|_2^2}{2\eta(\|\mathbf{x}_0|_{S_w}\|_2^2\|\mathbf{x}_0|_{S_w}\|_4^4 - \|\mathbf{x}_0|_{S_w}\|_3^6)}$$

$$\nu = \frac{y_0\|\mathbf{x}_0|_{S_w}\|_4^4 + \|\mathbf{x}_0|_{S_w}\|_3^3\|\mathbf{x}_0|_{S_w}\|_2^2 - \|\mathbf{x}_0|_{S_w}\|_1\|\mathbf{x}_0|_{S_w}\|_4^4 - \frac{1}{2\eta}\|\mathbf{x}_0|_{S_w}\|_3^3}{\|\mathbf{x}_0|_{S_w}\|_3^6 - \|\mathbf{x}_0|_{S_w}\|_2^2\|\mathbf{x}_0|_{S_w}\|_4^4} \tag{35}$$

  *or $\mathbf{x}_0 = \alpha\mathbf{1}$, for some $\alpha \neq 0$, and $(\lambda, \nu)$ satisfying*

$$\|\mathbf{x}_0|_{S_w}\|_1 - 2\eta\lambda\|\mathbf{x}_0|_{S_w}\|_3^3 - \nu\|\mathbf{x}_0|_{S_w}\|_2^2 = y_0, \tag{36}$$

  *both with $\lambda \geq 0$ and $2\lambda\eta(x_0)_i + \nu(x_0)_i < 1$, for all $i \in S_w$, then $\mathbf{w}$ is a KKT point of (34).*

*This characterizes all KKT points of (34).*

*Proof.* First note that

$$\mathcal{R}(\mathbf{w}) = \mathbb{E}_{(\mathbf{x},y)} \mathcal{L}(\mathbf{w}) = \frac{1}{2} \mathbb{E}_{(\mathbf{x},y)}(\langle \mathbf{w}^{\odot 2}, \mathbf{x} \rangle - y)^2$$

$$= \frac{1}{2}\left((\mathbf{w}^{\odot 2})^T \mathbb{E}(\mathbf{x}\mathbf{x}^T)\mathbf{w}^{\odot 2} - 2\mathbb{E}(y\mathbf{x}^T)\mathbf{w}^{\odot 2} + \mathbb{E} y^2\right)$$

$$= \frac{1}{2}\|\mathbf{w}^{\odot 2}\|_2^2 - \langle \mathbf{1}, \mathbf{w}^{\odot 2} \rangle + \frac{1}{2}(d+1)$$

$$= \frac{1}{2}\|\mathbf{w}\|_4^4 - \|\mathbf{w}\|_2^2 + \frac{1}{2}(d+1),$$

---

[6]Note that $(\mathbf{x}_0, y_0)$ takes in this section the role of the single data point $(\mathbf{x}, y)$ from before and that we condition to non-negative data in order to apply Proposition 3.2.

where we used in the penultimate line that $\mathbb{E}\left(y\mathbf{x}^T\right) = \mathbf{1}^T$ and $\mathbb{E}\left(y^2\right) = d+1$, and in the ultimate line that $\langle \mathbf{1}, \mathbf{w}^{\odot 2}\rangle = \|\mathbf{w}\|_2^2$ and $\|\mathbf{w}^{\odot 2}\|_2^2 = \|\mathbf{w}\|_4^4$.

For the KKT analysis of Equation (34), we will drop the additive constant $\frac{1}{2}(d+1)$. We first re-write Equation (34) as

$$\min_{\mathbf{w}\in\mathbb{R}^d} f(\mathbf{w}), \quad \text{s.t.} \quad h(\mathbf{w}) = 0, \quad g(\mathbf{w}) \le 0.$$

where

$$f(\mathbf{w}) = \frac{1}{2}\|\mathbf{w}\|_4^4 - \|\mathbf{w}\|_2^2$$
$$h(\mathbf{w}) = \langle \mathbf{x}_0, \mathbf{w}^{\odot 2}\rangle - y_0$$
$$g(\mathbf{w}) = 2\eta\|\mathbf{x}_0 \odot \mathbf{w}\|_2^2 - 1.$$

The point $\mathbf{w}$ satisfies the KKT conditions if there exists $\lambda, \nu \in \mathbb{R}$ such that

$$\nabla f(\mathbf{w}) + \nu\nabla h(\mathbf{w}) + \lambda\nabla g(\mathbf{w}) = \mathbf{0}$$
$$h(\mathbf{w}) = 0$$
$$g(\mathbf{w}) \le 0$$
$$\lambda g(\mathbf{w}) = 0$$
$$\lambda \ge 0.$$

Plugging in, we obtain

$$2\mathbf{w}^{\odot 3} - 2\mathbf{w} + 2\nu\mathbf{x}_0 \odot \mathbf{w} + 4\lambda\eta\mathbf{x}_0^{\odot 2} \odot \mathbf{w} = \mathbf{0} \tag{37}$$
$$\langle \mathbf{x}_0, \mathbf{w}^{\odot 2}\rangle - y_0 = 0 \tag{38}$$
$$2\eta\|\mathbf{x}_0 \odot \mathbf{w}\|_2^2 - 1 \le 0 \tag{39}$$
$$\lambda(2\eta\|\mathbf{x}_0 \odot \mathbf{w}\|_2^2 - 1) = 0 \tag{40}$$
$$\lambda \ge 0. \tag{41}$$

By rewriting (37) as

$$(\mathbf{w}^{\odot 2} - \mathbf{1} + \nu\mathbf{x}_0 + 2\lambda\eta\mathbf{x}_0^{\odot 2}) \odot \mathbf{w} = \mathbf{0},$$

we see that, for any $i \in [d]$, we have

$$w_i = 0 \quad \text{or} \quad w_i^2 = 1 - \nu(x_0)_i - 2\lambda\eta(x_0)_i^2. \tag{42}$$

Consider any $\mathbf{w}$ with $\text{supp}(\mathbf{w}) = S_w$ satisfying the KKT conditions.

If $\lambda = 0$, we get that $\mathbf{w}|_{S_w}^{\odot 2} = (\mathbf{1} - \nu\mathbf{x}_0)|_{S_w}$ such that (38) yields that

$$\|\mathbf{x}_0|_{S_w}\|_1 - \nu\|\mathbf{x}_0|_{S_w}\|_2^2 = y_0 \quad \Leftrightarrow \quad \nu = \frac{\|\mathbf{x}_0|_{S_w}\|_1 - y_0}{\|\mathbf{x}_0|_{S_w}\|_2^2},$$

which implies that a suitable $\nu$ exists iff $S_w \cap \text{supp}(\mathbf{x}_0) \ne \emptyset$ and $\nu < \min_{i\in S_w\cap\text{supp}(\mathbf{x}_0)} \frac{1}{(x_0)_i}$. The latter condition stems from the fact that non-zero entries of $\mathbf{w}^{\odot 2}$ have to be positive. Finally, to be a KKT point, $\mathbf{w}$ has to satisfy (39).

If $\lambda \ne 0$, we get that $\mathbf{w}|_{S_w}^{\odot 2} = (\mathbf{1} - 2\lambda\eta\mathbf{x}_0^{\odot 2} - \nu\mathbf{x}_0)|_{S_w}$ such that (38) and (40) yield that

$$\|\mathbf{x}_0|_{S_w}\|_1 - 2\eta\lambda\|\mathbf{x}_0|_{S_w}\|_3^3 - \nu\|\mathbf{x}_0|_{S_w}\|_2^2 = y_0$$
$$\|\mathbf{x}_0|_{S_w}\|_2^2 - 2\eta\lambda\|\mathbf{x}_0|_{S_w}\|_4^4 - \nu\|\mathbf{x}_0|_{S_w}\|_3^3 = \frac{1}{2\eta},$$

which is a solvable linear system iff $S_w \cap \text{supp}(\mathbf{x}_0) \ne \emptyset$. If $\mathbf{x}_0|_{S_w} \ne \alpha\mathbf{1}|_{S_w}$, for all $\alpha \ne 0$, the unique solution is given by (35). Else, the system is underdetermined and only yields the relation in (36). Finally, if $\lambda \ge 0$ and $(2\lambda\eta\mathbf{x}_0^{\odot 2} + \nu\mathbf{x}_0)|_{S_w} < \mathbf{1}|_{S_w}$ (positivity constraint for non-zero entries of $\mathbf{w}^{\odot 2}$), any resulting $\mathbf{w}$ yields the second type of KKT point. $\qquad\square$

While it is cumbersome to analytically extract for general $d$ which of the KKT points of Lemma F.1 corresponds to a global minimizer, we can easily evaluate this numerically in our toy example from Figure 5, see Section 3. The exemplary data point used in the shown experiments is $\mathbf{x}_0 = (0.5, 3)$, $y = 1$.

### F.1. A More General Regression Analysis

Since it is more natural to have unconditioned training data, let us now assume that our data follows a general distribution $(\mathbf{x}, y) \sim \mathcal{D}$. Then, the risk for a parameter choice $\mathbf{w}$ under the model in (3)-(4) is given by

$$\mathcal{R}(\mathbf{w}) = \mathbb{E}_{(\mathbf{x},y)} \mathcal{L}(\mathbf{w}) = \frac{1}{2}\left((\mathbf{w}^{\odot 2})^T \boldsymbol{\Sigma} \mathbf{w}^{\odot 2} - 2\boldsymbol{\mu}^T \mathbf{w}^{\odot 2} + \sigma^2\right), \tag{43}$$

where we define $\boldsymbol{\Sigma} = \mathbb{E}(\mathbf{x}\mathbf{x}^T)$, $\boldsymbol{\mu} = \mathbb{E}(y\mathbf{x})$, and $\sigma^2 = \mathbb{E}\, y^2$. Under mild technical assumptions on $\mathcal{D}$ and considering a single training data point $(\mathbf{x}_0, y_0) \sim (\mathbf{x}, y)$, we can compare the three (idealized) training algorithms $\mathcal{A}_{\ell_1}$, $\mathcal{A}_{S_{\mathcal{L}}}$, and $\mathcal{A}_{\mathrm{opt}}$ from above which minimize $\ell_1$-norm, sharpness, and generalization error on $\mathcal{M}$, respectively.

**Proposition F.2.** *Assume that $\mathcal{D}$ is a distribution such that $\boldsymbol{\Sigma}, \boldsymbol{\mu}, \sigma^2$ are well-defined and finite, that $\boldsymbol{\Sigma}$ is invertible, that $\mathbf{x} \in \mathbb{R}^d_{\geq 0}$ a.s., and that the entries of $\mathbf{x}$ are a.s. distinct. Then, given a single training data point $(\mathbf{x}_0, y_0) \sim (\mathbf{x}, y)$ we have that*

*(i)* $\mathcal{A}_{\ell_1}(\mathbf{x}_0, y_0) = \sqrt{\frac{y_0}{x_{\max}}}\mathbf{e}_{k_{\max}}$, *where $k_{\max}$ is the index of the maximal entry of $\mathbf{x}_0$. The expected generalization error is given by*

$$\mathbb{E}_{(\mathbf{x}_0,y_0)} \mathcal{R}(\mathcal{A}_{\ell_1}(\mathbf{x}_0, y_0)) = \frac{1}{2}\left(\sigma^2 + \mathbb{E}\left(\frac{\Sigma_{k_{\max}k_{\max}}y_0^2}{x_{\max}^2}\right) + \mathbb{E}\left(\frac{\mu_{k_{\max}}y_0}{x_{\max}}\right)\right).$$

*(ii)* $\mathcal{A}_{S_{\mathcal{L}}}(\mathbf{x}_0, y_0) = \sqrt{\frac{y_0}{x_{\min}}}\mathbf{e}_{k_{\min}}$, *where $k_{\min}$ is the index of the minimal entry of $\mathbf{x}_0$. The expected generalization error is given by*

$$\mathbb{E}_{(\mathbf{x}_0,y_0)} \mathcal{R}(\mathcal{A}_{S_{\mathcal{L}}}(\mathbf{x}_0, y_0)) = \frac{1}{2}\left(\sigma^2 + \mathbb{E}\left(\frac{\Sigma_{k_{\min}k_{\min}}y_0^2}{x_{\min}^2}\right) + \mathbb{E}\left(\frac{\mu_{k_{\min}}y_0}{x_{\min}}\right)\right).$$

*(iii)* $\mathcal{A}_{\mathrm{opt}}(\mathbf{x}_0, y_0) = \left(\boldsymbol{\Sigma}^{-\frac{1}{2}}\left(\mathcal{P}^{\perp}_{\mathbf{x}_{\boldsymbol{\Sigma}}}\boldsymbol{\mu}_{\boldsymbol{\Sigma}} + \frac{y_0}{\|\mathbf{x}_{\boldsymbol{\Sigma}}\|_2^2}\mathbf{x}_{\boldsymbol{\Sigma}}\right)\right)^{\odot\frac{1}{2}}$, *where $\mathcal{P}_{\mathbf{z}}$ denotes the orthogonal projection onto $\mathrm{span}\{\mathbf{z}\}$, $\mathbf{x}_{\boldsymbol{\Sigma}} = \boldsymbol{\Sigma}^{-\frac{1}{2}}\mathbf{x}_0$, and $\boldsymbol{\mu}_{\boldsymbol{\Sigma}} = \boldsymbol{\Sigma}^{-\frac{1}{2}}\boldsymbol{\mu}$. The expected generalization error is given by*

$$\mathbb{E}_{(\mathbf{x}_0,y_0)} \mathcal{R}(\mathcal{A}_{\mathrm{opt}}(\mathbf{x}_0, y_0))$$
$$= \frac{1}{2}\left(\sigma^2 + \mathbb{E}\left(\frac{y_0^2}{\|\mathbf{x}_{\boldsymbol{\Sigma}}\|_2^2}\right) - 2\boldsymbol{\mu}_{\boldsymbol{\Sigma}}^T \mathbb{E}\left(\frac{y_0}{\|\mathbf{x}_{\boldsymbol{\Sigma}}\|_2^2}\mathbf{x}_{\boldsymbol{\Sigma}}\right) - \boldsymbol{\mu}_{\boldsymbol{\Sigma}}^T \mathbb{E}\,\mathcal{P}^{\perp}_{\mathbf{x}_{\boldsymbol{\Sigma}}}\boldsymbol{\mu}_{\boldsymbol{\Sigma}}\right).$$

Although it is not possible to analytically evaluate the expectations on this level of generality, the expected generalization error of $\mathcal{A}_{S_{\mathcal{L}}}(\mathbf{x}_0, y_0)$ will presumably be larger than the one of $\mathcal{A}_{\ell_1}(\mathbf{x}_0, y_0)$ since $x_{\min} < x_{\max}$; just like in the specific setting in the beginning of Section F.

*Proof of Proposition F.2.* By our assumptions on the distribution of $\mathbf{x}_0$, Points $(i)$ and $(ii)$ follow from applying Proposition 3.2, and inserting the resulting minimizer into (43).

To derive $(iii)$, we abbreviate $\tilde{\mathbf{w}} = \boldsymbol{\Sigma}^{\frac{1}{2}}\mathbf{w}^{\odot 2}$, $\boldsymbol{\mu}_{\boldsymbol{\Sigma}} = \boldsymbol{\Sigma}^{-\frac{1}{2}}\boldsymbol{\mu}$, and $\mathbf{x}_{\boldsymbol{\Sigma}} = \boldsymbol{\Sigma}^{-\frac{1}{2}}\mathbf{x}_0$, and consider the linearly constrained optimization problem

$$\min_{\mathbf{w}\in\mathcal{M}} \mathcal{R}(\mathbf{w}) = \frac{1}{2}\min_{\tilde{\mathbf{w}}\in\mathbb{R}^d} \|\tilde{\mathbf{w}}\|_2^2 - 2\boldsymbol{\mu}_{\boldsymbol{\Sigma}}^T\tilde{\mathbf{w}} + \sigma^2, \quad \text{s.t. } \mathbf{x}_{\boldsymbol{\Sigma}}^T\tilde{\mathbf{w}} = y_0. \tag{44}$$

Since the objective is convex and the constraints are linear, the KKT-conditions of (44)

$$\begin{cases} 2\tilde{\mathbf{w}} - 2\boldsymbol{\mu}_{\boldsymbol{\Sigma}} + \lambda\mathbf{x}_{\boldsymbol{\Sigma}} = 0 \\ \mathbf{x}_{\boldsymbol{\Sigma}}^T\tilde{\mathbf{w}} = y_0 \end{cases} \iff \begin{cases} \tilde{\mathbf{w}} = \boldsymbol{\mu}_{\boldsymbol{\Sigma}} - \frac{1}{2}\lambda\mathbf{x}_{\boldsymbol{\Sigma}} \\ \mathbf{x}_{\boldsymbol{\Sigma}}^T\boldsymbol{\mu}_{\boldsymbol{\Sigma}} - \frac{1}{2}\lambda\|\mathbf{x}_{\boldsymbol{\Sigma}}\|_2^2 = y_0 \end{cases} \iff \begin{cases} \tilde{\mathbf{w}} = \boldsymbol{\mu}_{\boldsymbol{\Sigma}} - \frac{1}{2}\lambda\mathbf{x}_{\boldsymbol{\Sigma}} \\ \frac{1}{2}\lambda = \frac{1}{\|\mathbf{x}_{\boldsymbol{\Sigma}}\|_2^2}(\mathbf{x}_{\boldsymbol{\Sigma}}^T\boldsymbol{\mu}_{\boldsymbol{\Sigma}} - y_0) \end{cases}$$

are sufficient and necessary, and yield the unique minimizer

$$\tilde{\mathbf{w}}_\star = \left(\mathbf{I} - \frac{\mathbf{x}_\Sigma \mathbf{x}_\Sigma^T}{\|\mathbf{x}_\Sigma\|_2^2}\right)\boldsymbol{\mu}_\Sigma + \frac{y_0}{\|\mathbf{x}_\Sigma\|_2^2}\mathbf{x}_\Sigma$$

with

$$
\begin{aligned}
\mathcal{R}(\mathcal{A}_{\mathrm{opt}}(\mathbf{x}_0, y_0)) &= \frac{1}{2}\left(\|\tilde{\mathbf{w}}_\star\|_2^2 - 2\boldsymbol{\mu}_\Sigma^T\tilde{\mathbf{w}}_\star + \sigma^2\right) \\
&= \frac{1}{2}\left(\left\|\mathcal{P}_{\mathbf{x}_\Sigma}^\perp\boldsymbol{\mu}_\Sigma + \frac{y_0}{\|\mathbf{x}_\Sigma\|_2^2}\mathbf{x}_\Sigma\right\|_2^2 - 2\boldsymbol{\mu}_\Sigma^T\left(\mathcal{P}_{\mathbf{x}_\Sigma}^\perp\boldsymbol{\mu}_\Sigma + \frac{y_0}{\|\mathbf{x}_\Sigma\|_2^2}\mathbf{x}_\Sigma\right) + \sigma^2\right) \\
&= \frac{1}{2}\left(\boldsymbol{\mu}_\Sigma^T\mathcal{P}_{\mathbf{x}_\Sigma}^\perp\boldsymbol{\mu}_\Sigma + \left\|\frac{y_0}{\|\mathbf{x}_\Sigma\|_2^2}\mathbf{x}_\Sigma\right\|_2^2 - 2\boldsymbol{\mu}_\Sigma^T\mathcal{P}_{\mathbf{x}_\Sigma}^\perp\boldsymbol{\mu}_\Sigma - 2\frac{y_0}{\|\mathbf{x}_\Sigma\|_2^2}\boldsymbol{\mu}_\Sigma^T\mathbf{x}_\Sigma + \sigma^2\right) \\
&= \frac{1}{2}\left(\frac{y_0^2}{\|\mathbf{x}_\Sigma\|_2^2} - 2\frac{y_0}{\|\mathbf{x}_\Sigma\|_2^2}\boldsymbol{\mu}_\Sigma^T\mathbf{x}_\Sigma - \boldsymbol{\mu}_\Sigma^T\mathcal{P}_{\mathbf{x}_\Sigma}^\perp\boldsymbol{\mu}_\Sigma + \sigma^2\right).
\end{aligned}
$$

Consequently,

$$
\begin{aligned}
&\mathbb{E}_{(\mathbf{x}_0, y_0)}\,\mathcal{R}(\mathcal{A}_{\mathrm{opt}}(\mathbf{x}_0, y_0)) \\
&= \frac{1}{2}\left(\sigma^2 + \mathbb{E}\left(\frac{y_0^2}{\|\mathbf{x}_\Sigma\|_2^2}\right) - 2\boldsymbol{\mu}_\Sigma^T\,\mathbb{E}\left(\frac{y_0}{\|\mathbf{x}_\Sigma\|_2^2}\mathbf{x}_\Sigma\right) - \boldsymbol{\mu}_\Sigma^T\,\mathbb{E}\,\mathcal{P}_{\mathbf{x}_\Sigma}^\perp\boldsymbol{\mu}_\Sigma\right).
\end{aligned}
$$

$\square$

We can now use Proposition F.2 to examine a regression task in which the feature distribution is a folded Gaussian and thus restricted to the positive orthant. Let $\mathbf{x} \sim |\mathcal{N}(0, \mathbf{I}_n)|$ and $y = \langle\mathbf{1}, \mathbf{x}\rangle$. Then $\boldsymbol{\Sigma}$, $\boldsymbol{\mu}$, and $\sigma^2$ are given by

$$\Sigma_{ij} = \mathbb{E}(\mathbf{x}_i\mathbf{x}_j) = \begin{cases} 1 & \text{if } i = j \\ \frac{2}{\pi} & \text{if } i \neq j \end{cases}$$

$$\mu_i = \mathbb{E}(y\mathbf{x}_i) = \mathbb{E}(\mathbf{x}_i^2) + \sum_{j:j\neq i}\mathbb{E}(\mathbf{x}_i\mathbf{x}_j) = 1 + \frac{2(n-1)}{\pi}$$

$$\sigma^2 = \mathbb{E}(y^2) = \sum_i\mathbb{E}(\mathbf{x}_i^2) + \sum_{i,j:i\neq j}\mathbb{E}(\mathbf{x}_i\mathbf{x}_j) = n + \frac{2n(n-1)}{\pi}$$

By Proposition F.2, we obtain the following results: For $\mathcal{A}_{\ell_1}(\mathbf{x}_0, y_0)$, the expected generalization error is given by

$$\frac{1}{2}\left(\frac{n(2n-2+\pi)}{\pi} + \mathbb{E}\left(\frac{\langle\mathbf{1}, \mathbf{x}_0\rangle^2}{x_{\max}^2}\right) + \frac{2n-2+\pi}{\pi}\,\mathbb{E}\left(\frac{\langle\mathbf{1}, \mathbf{x}_0\rangle}{x_{\max}}\right)\right).$$

Since $\langle\mathbf{1}, \mathbf{x}_0\rangle \leq nx_{\max}$, the above expectation terms are bounded by

$$\mathbb{E}\,\frac{\langle\mathbf{1}, \mathbf{x}_0\rangle^2}{x_{\max}^2} \leq n^2, \quad \mathbb{E}\,\frac{\langle\mathbf{1}, \mathbf{x}_0\rangle}{x_{\max}} \leq n.$$

For $\mathcal{A}_{S_\mathcal{L}}(\mathbf{x}_0, y_0)$, the expected generalization error is given by

$$\frac{1}{2}\left(\frac{n(2n-2+\pi)}{\pi} + \mathbb{E}\left(\frac{\langle\mathbf{1}, \mathbf{x}_0\rangle^2}{x_{\min}^2}\right) + \frac{2n-2+\pi}{\pi}\,\mathbb{E}\left(\frac{\langle\mathbf{1}, \mathbf{x}_0\rangle}{x_{\min}}\right)\right).$$

However, in this case due to $x_{\min}$ the expectation blows up to infinity as shown below.

$$
\begin{aligned}
\mathbb{E}\,\frac{\langle\mathbf{1}, \mathbf{x}_0\rangle}{x_{\min}} &\geq \left(\frac{2}{\pi}\right)^{n/2}\int_{[0,1]\times[1,2]^{n-1}}\frac{x_1 + \cdots + x_n}{\min_i x_i}e^{-\frac{1}{2}(x_1^2 + \cdots + x_n^2)}dx_1\cdots dx_n \\
&\geq \left(\frac{2}{\pi}\right)^{n/2}\underbrace{\int_{[0,1]}\frac{n-1}{x_1}e^{-\frac{1}{2}x_1^2}dx_1}_{=\infty}\underbrace{\int_{[1,2]^{n-1}}e^{-\frac{1}{2}(x_2^2 + \cdots + x_n^2)}dx_2\cdots dx_n}_{>0} = \infty.
\end{aligned}
$$

Consequently, as in the simpler setting above we see that the implicit GF-regularization leads to smaller generalization error than the sharpness regularization.

## F.2. Multiple Data Points

Finally, we extend the results from Lemma F.1 to multiple Gaussian data points. Recall from Appendix E that in this case the sharpness $S_{\mathcal{L}}$ concentrates around its mean, which is approximately proportional to $\|w\|_\infty^2$. If $m = cd$ for some $c \in (0, 1)$, we furthermore know that any limit point $\mathbf{w}$ of GD with learning rate $\eta$ satisfies $m \|\mathbf{w}\|_\infty^2 \approx S_{\mathcal{L}} \leq \frac{2}{\eta}$ by combining Theorems E.2 and B.2. We can thus analyze the KKT points of the corresponding risk minimization problem as follows.

**Lemma F.3** (KKT points for multiple data points). *Let* $\mathcal{R}(\mathbf{w}) = \mathbb{E}_{(\mathbf{x},y)} \mathcal{L}(\mathbf{w})$ *be the population risk from Lemma F.1, i.e.*

$$\mathcal{R}(\mathbf{w}) = \frac{1}{2}\|\mathbf{w}\|_4^4 - \|\mathbf{w}\|_2^2 + \frac{1}{2}(d+1). \tag{45}$$

*Let* $\eta > 0$, $\mathbf{X} \in \mathbb{R}^{m \times d}$, *and* $\mathbf{y} \in \mathbb{R}^m$. *Consider*

$$\min_{\mathbf{w} \in \mathbb{R}^d} \mathcal{R}(\mathbf{w}), \quad s.t. \quad \mathbf{X}\mathbf{w}^{\odot 2} = \mathbf{y}, \quad \|\mathbf{w}\|_\infty^2 \leq \frac{2}{\eta m}. \tag{46}$$

*Fix a support* $S_w \subset [d]$ *and define*

$$A := \left\{ i \in S_w : \ w_i^2 = \frac{2}{\eta m} \right\}, \qquad F := S_w \setminus A. \tag{47}$$

*Then* $\mathbf{w}$ *is a KKT point of* (46) *with* $\mathrm{supp}(\mathbf{w}) = S_w$ *and active set* $A$ *if and only if there exists* $\nu \in \mathbb{R}^m$ *such that*

$$(\mathbf{X}|_F \mathbf{X}|_F^\top)\nu = \mathbf{X}|_F \mathbf{1}_F + \frac{2}{\eta m}\mathbf{X}|_A \mathbf{1}_A - \mathbf{y}, \tag{48}$$

*and (componentwise)*

$$0 < \mathbf{1}_F - \mathbf{X}|_F^\top \nu < \frac{2}{\eta m}\mathbf{1}_F, \qquad \mathbf{X}|_A^\top \nu \leq \left(1 - \frac{2}{\eta m}\right)\mathbf{1}_A. \tag{49}$$

*In that case, the squared coordinates are given by*

$$w|_A^2 = \frac{2}{\eta m}\mathbf{1}_A, \qquad w|_F^2 = \mathbf{1}_F - \mathbf{X}|_F^\top \nu, \qquad w_{S_w^c} = 0,$$

*and one may choose KKT multipliers* $\lambda \in \mathbb{R}_{\geq 0}^d$ *with* $\lambda|_F = 0$, $\lambda_{S_w^c} = 0$, *and*

$$\lambda|_A = \frac{2}{\eta m}\left(\left(1 - \frac{2}{\eta m}\right)\mathbf{1}_A - \mathbf{X}|_A^\top \nu\right) \geq 0. \tag{50}$$

*This uniquely determines* $\mathbf{w}$ *up to sign choices on* $S_w$.

*If* $\mathbf{X}|_F \mathbf{X}|_F^\top$ *is invertible, the unique* $\nu$ *satisfying* (48) *is*

$$\nu = (\mathbf{X}|_F \mathbf{X}|_F^\top)^{-1}\left(\mathbf{X}|_F \mathbf{1}_F + \frac{2}{\eta m}\mathbf{X}|_A \mathbf{1}_A - \mathbf{y}\right). \tag{51}$$

*This characterizes all KKT points of* (46).

We remark that, as $\mathbf{X}$ has i.i.d. Gaussian entries, if $|F| \geq m$ then $\mathbf{X}|_F \mathbf{X}|_F^\top$ is almost surely invertible.

*Proof.* We drop the additive constant $\frac{1}{2}(d + 1)$, define our minimization objective

$$f(\mathbf{w}) = \frac{1}{2}\|\mathbf{w}\|_4^4 - \|\mathbf{w}\|_2^2,$$

the equality constraint

$$h(\mathbf{w}) := \mathbf{X}\mathbf{w}^{\odot 2} - \mathbf{y} = \mathbf{0}.$$

and write the infinity-norm constraint $g(\mathbf{w}) \leq \mathbf{0}$ coordinate-wise as

$$g_i(\mathbf{w}) := \frac{\eta m}{2}w_i^2 - 1 \leq 0, \qquad i \in [d].$$

A point $\mathbf{w}$ satisfies the KKT conditions if there exists $\lambda \in \mathbb{R}_{\geq 0}^d, \nu \in \mathbb{R}^m$ such that

$$\nabla f(\mathbf{w}) + \nabla h(\mathbf{w})\nu + \nabla g(\mathbf{w})\lambda = \mathbf{0}$$
$$h(\mathbf{w}) = 0$$
$$g(\mathbf{w}) \leq \mathbf{0}$$
$$\lambda^\top g(\mathbf{w}) = 0$$
$$\lambda \geq \mathbf{0}.$$

We plug in and obtain

$$2\mathbf{w}^{\odot 3} - 2\mathbf{w} + 2(\mathbf{X}^\top \nu) \odot \mathbf{w} + \eta m\, \lambda \odot \mathbf{w} = \mathbf{0}, \tag{52}$$
$$\mathbf{X}\mathbf{w}^{\odot 2} - \mathbf{y} = \mathbf{0} \tag{53}$$
$$\frac{\eta m}{2} w_i^2 - 1 \leq 0, \qquad i \in [d]. \tag{54}$$
$$\lambda_i\Big(\frac{\eta m}{2} w_i^2 - 1\Big) = 0, \qquad i \in [d]. \tag{55}$$
$$\lambda_i \geq 0, \qquad i \in [d]. \tag{56}$$

By rewriting (52) we see

$$\Big(\mathbf{w}^{\odot 2} - \mathbf{1} + \mathbf{X}^\top \nu + \frac{\eta m}{2}\lambda\Big) \odot \mathbf{w} = \mathbf{0}$$

which means for each $i \in [d]$ that

$$w_i = 0 \quad \text{or} \quad w_i^2 = 1 - (\mathbf{X}^\top \nu)_i - \frac{\eta m}{2}\lambda_i. \tag{57}$$

Now consider any $\mathbf{w}$ with $\mathrm{supp}(\mathbf{w}) = S_w$ satisfying the KKT conditions. We define the sets

$$A := \Big\{ i \in S_w : w_i^2 = \frac{2}{\eta m} \Big\}, \qquad F := S_w \setminus A.$$

For $i \in F$ we have $w_i^2 < \frac{2}{\eta m}$, and hence $\lambda_i = 0$. Then (57) gives

$$w|_F^{\odot 2} = \mathbf{1}_F - \mathbf{X}|_F^\top \nu. \tag{58}$$

For $i \in A$ we have $w_i^2 = \frac{2}{\eta m}$, so (57) gives

$$\frac{2}{\eta m} = 1 - (\mathbf{X}^\top \nu)_i - \frac{\eta m}{2}\lambda_i \quad \Longleftrightarrow \quad \lambda_i = \frac{2}{\eta m}\Big(1 - (\mathbf{X}^\top \nu)_i - \frac{2}{\eta m}\Big), \qquad i \in A,$$

and by definition $w|_A^{\odot 2} = \frac{2}{\eta m}\mathbf{1}_A$. Note that $\lambda_i \geq 0$ holds due to (56)

Substituting $w|_F^{\odot 2}$ and $w|_A^{\odot 2}$ into the equality constraint $\mathbf{X}\mathbf{w}^{\odot 2} = \mathbf{y}$ yields

$$\mathbf{X}|_F(\mathbf{1}_F - \mathbf{X}|_F^\top \nu) + \frac{2}{\eta m}\mathbf{X}|_A\mathbf{1}_A = \mathbf{y}. \tag{59}$$

If $\mathbf{X}|_F\mathbf{X}|_F^\top$ is invertible, (59) has the unique solution

$$\nu = (\mathbf{X}|_F\mathbf{X}|_F^\top)^{-1}\Big(\mathbf{X}|_F\mathbf{1}_F + \frac{2}{\eta m}\mathbf{X}|_A\mathbf{1}_A - \mathbf{y}\Big).$$

This uniquely determines $\mathbf{w}^{\odot 2}$ on $S_w$ via $w|_A^2 = \frac{2}{\eta m}\mathbf{1}_A$ and (58). Finally, $\mathbf{w}$ itself is determined up to independent sign choices on $S_w$.

In the other case, rearranging (59) gives the linear system

$$(\mathbf{X}|_F\mathbf{X}|_F^\top)\nu = \mathbf{X}|_F\mathbf{1}_F + \frac{2}{\eta m}\mathbf{X}|_A\mathbf{1}_A - \mathbf{y}. \tag{60}$$

If $\nu_1, \nu_2$ solve (60), then $(\mathbf{X}|_F \mathbf{X}|_F^\top)(\nu_1 - \nu_2) = 0$, hence

$$0 = (\nu_1 - \nu_2)^\top \mathbf{X}|_F \mathbf{X}|_F^\top (\nu_1 - \nu_2) = \|\mathbf{X}|_F^\top (\nu_1 - \nu_2)\|_2^2,$$

so $\mathbf{X}|_F^\top \nu_1 = \mathbf{X}|_F^\top \nu_2$. Therefore $w|_F^2 = \mathbf{1}_F - \mathbf{X}|_F^\top \nu$ is uniquely determined by $(S_w, A)$ even when $\mathbf{X}|_F \mathbf{X}|_F^\top$ is singular. Next, for $i \in A$ we have $w_i^2 = \frac{2}{\eta m}$ and (57) yields

$$\lambda_i = \frac{2}{\eta m}\left(1 - (\mathbf{X}^\top \nu)_i - \frac{2}{\eta m}\right), \qquad i \in A.$$

Thus dual feasibility $\lambda_i \geq 0$ is equivalent to

$$(\mathbf{X}^\top \nu)_i \leq 1 - \frac{2}{\eta m}, \qquad i \in A,$$

and we may write compactly

$$\lambda|_A = \frac{2}{\eta m}\left(\left(1 - \frac{2}{\eta m}\right)\mathbf{1}_A - \mathbf{X}|_A^\top \nu\right) \geq 0, \qquad \lambda|_F = 0, \qquad \lambda_{S_w^c} = 0.$$

Finally, since $\operatorname{supp}(\mathbf{w}) = S_w$ and $F = S_w \setminus A$, the definition of $F$ implies the strict component-wise feasibility $0 < w_i^2 < \frac{2}{\eta m}$ for all $i \in F$, i.e. $0 < \mathbf{1}_F - \mathbf{X}|_F^\top \nu < \frac{2}{\eta m}\mathbf{1}_F$.

A computation confirms that the reverse also holds, i.e. such constructed points satisfy the KKT conditions.

$\qquad\qquad\qquad\qquad\qquad\qquad\qquad\qquad\qquad\qquad\qquad\qquad\qquad\qquad\qquad\qquad\qquad\quad$ $\square$

Similarly to the single-data-point case, Lemma F.3 shows that KKT points of (46) split into two cases: *inactive sharpness constraints* ($A = \emptyset$, equivalently $\lambda = \mathbf{0}$ on $S_w$), and *active sharpness constraints* ($A \neq \emptyset$, i.e. at least one coordinate satisfies $|w_i|^2 = \frac{2}{\eta m}$). Empirically, as $\eta$ varies in the EoS regime we often observe convergent GD trajectories whose limits satisfy $m\|\mathbf{w}_\infty\|_\infty^2 \approx S_{\mathcal{L}}(\mathbf{w}) \approx \frac{2}{\eta}$, consistent with convergence to KKT points of the risk minimization problem with $A \neq \emptyset$. For appropriately tuned $\eta$, GD finds a solution of the unconstrained minimization problem. Hence, the multi-data point case is a direct analogue of the single-sample picture from Section 3: changing $\eta$ interpolates between regimes where GD behaves like an (unconstrained) risk minimizer on the interpolation manifold and regimes where GD is effectively restricted by sharpness constraints. In particular, the qualitative algorithmic conclusions from Section 3 extend to the multi-data-point setup.

## G. An Elementary Study of How Implicit Biases Interact II — Classification

In this section, we extend our insights from Section 3 to a simple classification set-up. To this end, define for data $D = \{(\mathbf{x}_i, y_i)\}_{i=1}^n \subset \mathbb{R}^{d+1} \times \{0, 1\}$ the logistic loss

$$\mathcal{L}(\mathbf{w}) = \frac{1}{n}\sum_{i=1}^n \left(y_i \log(g(\langle \mathbf{w}, \mathbf{x}_i \rangle)) + (1 - y_i)\log(1 - g(\langle \mathbf{w}, \mathbf{x}_i \rangle))\right),$$

where

$$g\colon \mathbb{R} \to \mathbb{R} \quad \text{with} \quad g(z) = \frac{1}{1 + e^{-z}}$$

is the logistic function. Here, we assume that $\mathbf{w} = (\tilde{\mathbf{w}}, b)^T$ and that the data points are of the form $\mathbf{x} = (\tilde{\mathbf{x}}, 1)^T$ such that the linear classifier $h_\mathbf{w}$ corresponding to parameters $\mathbf{w}$ is given by

$$h_\mathbf{w}(\mathbf{x}) = \mathbf{1}_{\{\mathbf{z} = (\tilde{\mathbf{z}}, 1)\colon \langle \mathbf{w}, \mathbf{z} \rangle > 0\}}(\mathbf{x}) = \mathbf{1}_{\{\tilde{\mathbf{z}}\colon \langle \tilde{\mathbf{w}}, \tilde{\mathbf{z}} \rangle + b > 0\}}(\mathbf{x}).$$

In the simplest possible case, we only have two data points with different labels. W.l.o.g. we assume that one of the two data points is centered at the origin and that their distance is normalized to one. Then we know the following.

**Theorem G.1.** *Let* $D = \{(\mathbf{x}_1, 0), (\mathbf{x}_2, 1)\} \subset \mathbb{R}^{d+1} \times \{0, 1\}$ *where* $\mathbf{x}_i = (\tilde{\mathbf{x}}_i, 1)^T$ *with* $\tilde{\mathbf{x}}_1 = \mathbf{0}$ *and* $\|\tilde{\mathbf{x}}_2\|_2 = 1$. *Then,*

(i) *the max-margin classifier of $D$ is parameterized by any positive scalar multiple of $\mathbf{w} = (\tilde{\mathbf{w}}, b)^T$ with $\tilde{\mathbf{w}} = \tilde{\mathbf{x}}_2$ and $b = -1/2$.*

(ii) *the parameters minimizing the sharpness of $\mathcal{L}$ over*

$$\mathcal{M} = \{\mathbf{w} = (\tilde{\mathbf{w}}, b) \colon h_{\mathbf{w}}(\mathbf{x}_1) = 0, \ h_{\mathbf{w}}(\mathbf{x}_2) = 1, \ and \ \|\tilde{\mathbf{w}}\|_2 = 1\}$$

*are given by a min-margin classifier parameterized by $\mathbf{w} = (\tilde{\mathbf{w}}, b)$ with $\tilde{\mathbf{w}} = \tilde{\mathbf{x}}_2$ and $b = 0$.*

*Proof.* To see (i), just note that the decision boundary of the max-margin classifier in $\mathbb{R}^d$ must be orthogonal to $\tilde{\mathbf{x}}_2 - \tilde{\mathbf{x}}_1$ with $h_{\mathbf{w}}(\mathbf{x}_2) = 1$, i.e., $\tilde{\mathbf{w}} = \alpha(\tilde{\mathbf{x}}_2 - \tilde{\mathbf{x}}_1) = \alpha\tilde{\mathbf{x}}_2$, for $\alpha > 0$, and that it must contain $\frac{1}{2}(\mathbf{x}_1 + \mathbf{x}_2)$ which implies that $0 = \langle \tilde{\mathbf{w}}, \frac{1}{2}(\mathbf{x}_1 + \mathbf{x}_2)\rangle + b = \frac{1}{2}\alpha\|\tilde{\mathbf{x}}_2\|_2^2 + b$, i.e., $b = -\frac{1}{2}\alpha$.

For (ii), we compute that

$$\mathcal{L}(\mathbf{w}) = \frac{1}{2}\left(\log(1 - g(\langle \mathbf{w}, \mathbf{x}_1\rangle)) + \log(g(\langle \mathbf{w}, \mathbf{x}_2\rangle))\right)$$

$$= \frac{1}{2}\left(\log(1 - g(b)) + \log(g(\langle \mathbf{w}, \mathbf{x}_2\rangle))\right).$$

By using that $g'(z) = g(z)(1 - g(z))$, we then get that

$$\nabla\mathcal{L}(\mathbf{w}) = \frac{1}{2}\left(-g(\langle \mathbf{w}, \mathbf{x}_1\rangle)\cdot\mathbf{x}_1 + (1 - g(\langle \mathbf{w}, \mathbf{x}_2\rangle))\cdot\mathbf{x}_2\right)$$

and

$$\nabla^2\mathcal{L}(\mathbf{w}) = -\frac{1}{2}\left(g'(\langle \mathbf{w}, \mathbf{x}_1\rangle)\cdot\mathbf{x}_1\mathbf{x}_1^T + g'(\langle \mathbf{w}, \mathbf{x}_2\rangle)\cdot\mathbf{x}_2\mathbf{x}_2^T\right).$$

To deduce the sharpness $S(\mathbf{w}) = \left\|\nabla^2\mathcal{L}(\mathbf{w})\right\|$, we will compute the eigenvalues of the Hessian. First note, that any vector in the image of $\nabla^2\mathcal{L}(\mathbf{w})$ can be expressed as $\mathbf{x} = \alpha\mathbf{e}_{d+1} + \beta\mathbf{x}_2$. Now assume $\mathbf{x} \neq \mathbf{0}$ is an eigenvector with eigenvalue $\lambda \neq 0$. Then, since $\mathbf{x}_1 = \mathbf{e}_{d+1}$,

$$\nabla^2\mathcal{L}(\mathbf{w})\mathbf{x} = -\frac{1}{2}\left(g'(b)\left(\alpha + \beta\right)\mathbf{e}_{d+1} + g'(\langle \mathbf{w}, \mathbf{x}_2\rangle)\left(\alpha + 2\beta\right)\mathbf{x}_2\right)$$

$$= \lambda(\alpha\mathbf{e}_{d+1} + \beta\mathbf{x}_2),$$

where we used that $\mathbf{x}_2^T\mathbf{e}_{d+1} = \mathbf{e}_{d+1}^T\mathbf{x}_2 = 1$, $\mathbf{x}_2^T\mathbf{x}_2 = 2$, and $\mathbf{e}_{d+1}^T\mathbf{e}_{d+1} = 1$. Matching coefficients, we obtain the system

$$\begin{pmatrix} \frac{1}{2}g'(b) + \lambda & \frac{1}{2}g'(b) \\ \frac{1}{2}g'(\langle \mathbf{w}, \mathbf{x}_2\rangle) & g'(\langle \mathbf{w}, \mathbf{x}_2\rangle) + \lambda \end{pmatrix}\begin{pmatrix}\alpha \\ \beta\end{pmatrix} = 0.$$

Since $(\alpha, \beta) \neq \mathbf{0}$, this implies that the matrix has determinant zero and leads to the quadratic equation

$$\lambda^2 + \left(\frac{1}{2}g'(b) + g'(\langle \mathbf{w}, \mathbf{x}_2\rangle)\right)\lambda + \frac{1}{4}g'(b)\cdot g'(\langle \mathbf{w}, \mathbf{x}_2\rangle) = 0.$$

Since $g'(b), g'(\langle \mathbf{w}, \mathbf{x}_2\rangle) > 0$, the maximal solution of the latter system, i.e., the leading eigenvalue of $\nabla^2\mathcal{L}(\mathbf{w})$, is

$$S(\mathbf{w}) = \left\|\nabla^2\mathcal{L}(\mathbf{w})\right\| = \frac{\frac{1}{2}g'(b) + g'(\langle \mathbf{w}, \mathbf{x}_2\rangle) + \sqrt{\frac{1}{4}g'(b)^2 + g'(\langle \mathbf{w}, \mathbf{x}_2\rangle)^2}}{2}$$

$$= \frac{1}{4}\left(g'(b) + 2g'(\langle \tilde{\mathbf{w}}, \tilde{\mathbf{x}}_2\rangle + b) + \sqrt{g'(b)^2 + 4\cdot g'(\langle \tilde{\mathbf{w}}, \tilde{\mathbf{x}}_2\rangle + b)^2}\right).$$

The parameter minimizing the sharpness is then

$$\min_{\mathbf{w} \in \mathcal{M}} S_{\mathcal{L}}(\mathbf{w})$$

$$= \frac{1}{4} \min_{\|\tilde{\mathbf{w}}\|_2 = 1} g'(b) + 2g'(\langle \tilde{\mathbf{w}}, \tilde{\mathbf{x}}_2 \rangle + b) +$$

$$\sqrt{g'(b)^2 + (2g'(\langle \tilde{\mathbf{w}}, \tilde{\mathbf{x}}_2 \rangle + b))^2}, \qquad \text{s.t.} \begin{cases} b = \langle \mathbf{w}, \mathbf{x}_1 \rangle \leq 0 \\ \langle \tilde{\mathbf{w}}, \tilde{\mathbf{x}}_2 \rangle + b > 0 \end{cases}$$

$$= \frac{1}{4} \min_{z \in (0,1]} g'(b) + 2g'(z+b) + \sqrt{g'(b)^2 + (2g'(z+b))^2}, \qquad \text{s.t.} \ -z < b \leq 0$$

$$\approx 0.277$$

The minimum of the function is attained at $(z, b) = (1, 0)$ which means that $\tilde{\mathbf{w}} = \tilde{\mathbf{x}}_2$. $\qquad \square$

Analogously to the regression case, we can now evaluate the max-margin and the sharpness minimizing classifiers in terms of their expected generalization error in a toy set-up that assumes only two samples. To satisfy the requirements of Theorem G.1, we propose the following simple data generation process.

Let the samples be generated as $(\mathbf{x}_1, y_1)$ with $\tilde{\mathbf{x}}_1 = \mathbf{0}$ and $y_1 = 0$, and, for $k \geq 2$, as $(\mathbf{x}_k, y_k) \sim (\mathbf{x}, 1)$ which follows a joint distribution with $\mathbf{x} \sim \frac{\mathbf{g}}{\|\mathbf{g}\|_2}$, where $\mathbf{g} \sim \mathcal{N}(\boldsymbol{\mu}, I)$ for $\boldsymbol{\mu} \neq \mathbf{0}$. The classification task is thus to separate a Gaussian cluster that is projected to the unit sphere from the origin. Given two samples $(\mathbf{x}_1, y_1)$ and $(\mathbf{x}_2, y_2)$ one can use Theorem G.1 and numerically evaluate that the expected generalization error (Mohri et al., 2018). To get a feeling of it, let us consider the two cases where $\|\boldsymbol{\mu}\| \ll 1$ and $\|\boldsymbol{\mu}\| \gg 1$. Let $\mathbf{g}_0$ and $\mathbf{g}_0'$ be independent and distributed as $\mathcal{N}(0, I)$.

Suppose $\|\boldsymbol{\mu}\| \ll 1$. The expected generalization error for the max-margin classifier $\mathbf{w}_{max} = (\tilde{\mathbf{w}}_{max}, b_{max})^T$ is

$$\mathbb{E}_{\tilde{\mathbf{x}}_2} \mathbb{P}_{\mathbf{x}}[h_{\mathbf{w}_{max}(\mathbf{x}) \neq 1}] = \mathbb{E}_{\tilde{\mathbf{x}}_2} \mathbb{P}_{\mathbf{g}} \left[ \left\langle \tilde{\mathbf{x}}_2, \frac{\mathbf{g}}{\|\mathbf{g}\|_2} \right\rangle \leq \frac{1}{2} \right]$$

$$\approx \mathbb{E}_{\mathbf{g}_0'} \mathbb{P}_{\mathbf{g}_0} \left[ \left\langle \frac{\mathbf{g}_0'}{\|\mathbf{g}_0'\|_2}, \frac{\mathbf{g}_0}{\|\mathbf{g}_0\|_2} \right\rangle \leq \frac{1}{2} \right]$$

$$\approx \frac{\gamma(\frac{d}{2} + \frac{1}{2})}{\gamma(\frac{d}{2})\gamma(\frac{1}{2})} \int_{-1}^{\frac{1}{2}} (1 - x^2)^{\frac{d}{2} - 1} dx$$

$$\to 1 \ (\text{as } d \text{ grows})$$

because $(1 - x^2)^{\frac{d}{2} - 1}$ concentrates well around $x = 0$. On the other hand, the expected generalization error for the sharpness minimizing classifier $\mathbf{w}_{min} = (\tilde{\mathbf{w}}_{min}, b_{min})$ is

$$\mathbb{E}_{\tilde{\mathbf{x}}_2} \mathbb{P}_{\mathbf{x}}[h_{\mathbf{w}_{min}(\mathbf{x}) \neq 1}] = \mathbb{E}_{\tilde{\mathbf{x}}_2} \mathbb{P}_{\mathbf{g}} \left[ \left\langle \tilde{\mathbf{x}}_2, \frac{\mathbf{g}}{\|\mathbf{g}\|_2} \right\rangle \leq 0 \right]$$

$$\approx \mathbb{E}_{\mathbf{g}_0'} \mathbb{P}_{\mathbf{g}_0} \left[ \left\langle \frac{\mathbf{g}_0'}{\|\mathbf{g}_0'\|_2}, \frac{\mathbf{g}}{\|\mathbf{g}_0\|_2} \right\rangle \leq 0 \right]$$

$$= \frac{1}{2},$$

where we used symmetry of the distribution in the last step. We see that in contrast to Section F here the sharpness minimizer leads to a significantly smaller expected generalization error than the GF-induced regularization.

Now suppose that $\|\boldsymbol{\mu}\| \gg 1$. The expected generalization error for the max-margin classifier is

$$
\begin{aligned}
\mathbb{E}_{\tilde{\mathbf{x}}_2} \mathbb{P}_{\mathbf{x}}[h_{\mathbf{w}_{max}(\mathbf{x}) \neq 1}] &= \mathbb{E}_{\mathbf{g}'} \mathbb{P}_{\mathbf{g}}\left[\left\langle \frac{\mathbf{g}'}{\|\mathbf{g}'\|_2}, \frac{\mathbf{g}}{\|\mathbf{g}\|_2}\right\rangle \leq \frac{1}{2}\right] \\
&\approx \mathbb{E}_{\mathbf{g}'} \mathbb{P}_{\mathbf{g}}\left[\left\langle \frac{\mathbf{g}_0' + \boldsymbol{\mu}}{\|\boldsymbol{\mu}\|_2}, \frac{\mathbf{g}_0 + \boldsymbol{\mu}}{\|\boldsymbol{\mu}\|_2}\right\rangle \leq \frac{1}{2}\right] \\
&\approx \mathbb{E}_{\mathbf{g}'} \mathbb{P}_{\mathbf{g}}\left[\langle \mathbf{g}_0' + \mathbf{g}_0, \boldsymbol{\mu}\rangle \leq -\frac{1}{2}\|\boldsymbol{\mu}\|_2^2\right] \\
&= \frac{1}{\sqrt{2}(2\pi)^{\frac{d}{2}}} \int_{-\infty}^{-\frac{1}{2}\|\boldsymbol{\mu}\|_2} e^{-\frac{1}{4}x^2} dx \\
&= \frac{1}{(2\pi)^{\frac{d-1}{2}}} \cdot \Phi\left(-\frac{1}{2\sqrt{2}}\|\boldsymbol{\mu}\|_2\right)
\end{aligned}
$$

where $\Phi$ denotes the cumulative distribution function of the standard normal distribution. Similarly, the expected generalization error for the sharpness minimizing classifier is

$$
\begin{aligned}
\mathbb{E}_{\tilde{\mathbf{x}}_2} \mathbb{P}_{\mathbf{x}}[h_{\mathbf{w}_{min}(\mathbf{x}) \neq 1}] &= \mathbb{E}_{\tilde{\mathbf{x}}_2} \mathbb{P}_{\mathbf{g}}\left[\left\langle \tilde{\mathbf{x}}_2, \frac{\mathbf{g}}{\|\mathbf{g}\|_2}\right\rangle \leq 0\right] \\
&\approx \mathbb{E}_{\mathbf{g}_0'} \mathbb{P}_{\mathbf{g}_0}\left[\langle \mathbf{g}_0' + \mathbf{g}_0, \boldsymbol{\mu}\rangle \leq -\|\boldsymbol{\mu}\|_2^2\right] \\
&= \frac{1}{\sqrt{2}(2\pi)^{d/2}} \int_{-\infty}^{-\|\boldsymbol{\mu}\|_2} e^{-\frac{1}{4}x^2} dx \\
&= \frac{1}{(2\pi)^{\frac{d-1}{2}}} \cdot \Phi\left(-\frac{1}{\sqrt{2}}\|\boldsymbol{\mu}\|_2\right).
\end{aligned}
$$

Here, both expected generalization errors are small.

## H. Methodology

To ensure reproducibility, we follow a standard procedure for each experimental configuration, which is defined by a specific combination of dataset, architecture, activation function, and loss function. To isolate the effect of the learning rate, we fix the initialization across all runs within a configuration. We initialize using the default PyTorch scheme, which is a modified LeCun initialization (LeCun et al., 2002): Fixing a random seed, initial entries of each weight matrix are uniformly sampled from the interval $\left(-1/\sqrt{n_{l-1}}, 1/\sqrt{n_{l-1}}\right)$, where $n_{l-1}$ is the input dimension of the respective matrix.

We begin by computing the gradient flow solution using a fourth-order Runge-Kutta integrator (Runge, 1895). At each iteration step, we record the sharpness of the training loss. We also save model checkpoints whenever the training loss first drops below a power of ten (i.e., $10^{-1}$, $10^{-2}$, etc.). From this gradient flow trajectory, we extract two key statistics: the sharpness at initialization ($s_0$) and the maximum sharpness observed during the trajectory ($s_{\text{GF}}$). The values $1/s_0$ and $2/s_{\text{GF}}$ are of particular interest. Taking the learning rate of $1/s_0$ has been suggested as a heuristic for optimal step size selection for non-adaptive GD (Cohen et al., 2021), and for learning rates above $2/s_{\text{GF}}$, the well-known stability condition (2) is violated at some point of the gradient flow trajectory, suggesting that the loss decrease is not guaranteed there.

We construct the learning rate schedule for each configuration using two regular grids: a fine grid focused on the critical transition region, and a coarse grid which allows us to study the trade-off of the regularization in the EoS regime. The fine grid consists of 12 points uniformly spaced with step size $\frac{1}{2s_{\text{GF}}}$ in the interval $\left[\frac{1}{2s_{\text{GF}}}, \frac{6}{s_{\text{GF}}}\right]$. The coarse grid includes nine uniformly spaced learning rates interpolated in the interval $\left[\frac{6}{s_{\text{GF}}}, \frac{2}{s_0}\right]$, and additionally includes all learning rates sampled at the step size $\frac{1}{8} \cdot \left(\frac{2}{s_0} - \frac{6}{s_{\text{GF}}}\right)$ which are strictly greater than zero, and above until divergence. If we observe divergence already within the $\left[\frac{6}{s_{\text{GF}}}, \frac{2}{s_0}\right]$ interval, we manually refine the schedule by decreasing the step size.

For each learning rate in the schedule, we train the model using full-batch gradient descent until the training loss falls below a fixed threshold (see table 1 for the exact configuration). During training, we record the sharpness and norms every 10 epochs, and similar to the gradient flow experiments, we save the model checkpoints at every power-of-ten loss threshold. To compute the Hessian, we approximate its leading eigenvalues using the Lanczos algorithm applied to Hessian-vector products, which can be efficiently computed via backpropagation (Pearlmutter, 1994).

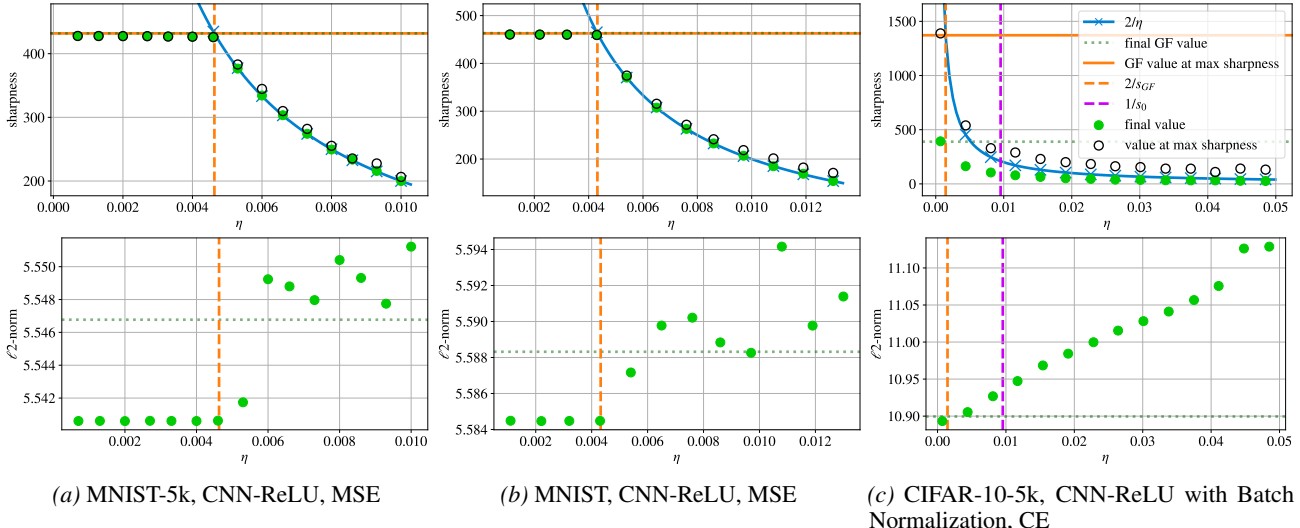

*(a)* MNIST-5k, CNN-ReLU, MSE     *(b)* MNIST, CNN-ReLU, MSE     *(c)* CIFAR-10-5k, CNN-ReLU with Batch Normalization, CE

*Figure 6.* Different configurations using the CNN architecture. We observe that the $\ell_2$-norm increase flattens out more towards larger $\eta$ in comparison to the FCN.

All experiments are fully reproducible, and the code is available at https://github.com/Maascha/conflicting-biases. Our implementation builds upon the original code by Cohen et al. (2021).

We ran the experiments on a heterogeneous computing infrastructure. Our hardware included NVIDIA A100, RTX 2080 Ti, TITAN RTX, RTX 3090 Ti, and RTX A6000 GPUs. Because GPU performance and availability varied across machines, we do not report a precise total runtime. However, the study required substantial computational effort: for each of the more than a dozen model configurations, we evaluated at least 20 learning rates, with individual runs ranging from a few minutes (for small models) to hundreds of hours (for larger models).

# I. Effect of Training Configuration on Sharpness–Norm Trade-off

As described in Section 2.1, we systematically investigate variants of our base configuration (fully-connected ReLU feed-forward network (FCN) with three layers, 200 hidden neurons each, trained on the first $5,000$ examples of MNIST or CIFAR with mean squared error) to demonstrate the relationship between sharpness and implicit regularization for varying step size.

We vary the dataset size, architecture, activation functions, loss functions, initialization and parameterization. While quantitative metrics such as the critical learning rate $\eta_c$ and absolute sharpness values differ, we consistently observe the norm-sharpness regularization trade-off.

In the following sections, we describe the findings on each variation and illustrate it with few representative plots. In all cases, we observe the same overall qualitative behavior. Additional supporting plots are included in the systematic overview of all experimental runs across configurations, provided in Appendix J and summarized in Table 1. For each of these configurations, we present both the coarse and fine-grained learning rate schedules to emphasize the transition region around $\eta_c$ as well as the behavior at larger learning rates.

## I.1. Dataset Size

Most of our experiments use a subset of $5,000$ training examples of MNIST and CIFAR-10 respectively, chosen to allow tractable estimation of sharpness across a wide range of learning rates. To confirm that our findings are not specific to the small dataset sizes, we run a limited number of configurations on the full MNIST and CIFAR-10 training sets. In Figure 6, we show the comparison of the sharpness and $\ell_2$-norm for a CNN with ReLU activation for MSE loss. The GF solution changes slightly, but the overall phenomenon persists and the values are relatively similar. We present additional figures on the full MNIST (see Appendix J.1.3, J.3.2, J.4.1) and full CIFAR (J.1.4) in Appendix J.

## I.2. Architecture

Our base model is a two-hidden-layer fully connected neural network (FCN), where each hidden layer consists of 200 neurons, with input and output layer sizes depending on the dataset.

To study the influence of the FCN architecture, we vary its widths and depths, namely experiments with $2\times$, $3\times$, and $10\times$ width, while keeping depth fixed, $2\times$ and $3\times$ depth, keeping width fixed, and $2\times$ and $3\times$ both width and depth. In other words, the considered FCN model shapes are: $200 \times 2$, $400 \times 2$, $600 \times 2$, $2000 \times 2$, $200 \times 4$, $200 \times 6$, $400 \times 4$, and $600 \times 6$ where the first number is the number of hidden neurons per hidden layer, and the second corresponds to the number of hidden layers.

While across most of these experiments the sharpness-norm trade-off is ever-present and consistent with the behavior of the standard model, increasing width alone on the MNIST-5k dataset leads to a dissolution of the trend of increasing norm. Here in the EoS regime the norm first decreases and then stays near constant (Figures 41, 42, and 43). However, we believe this to be the result of the limited range of learning rates, since for experiments increasing both width and depth we can see a similar decrease in norm at first, but a robust overall increase afterwards (Figures 46 and 47).

We further extend our analysis beyond the fully connected baseline by evaluating several alternative architectures: Convolutional networks (CNNs) with ReLU activations (Figure 6 and Appendix J.3), ResNet (Appendix J.5), and a Vision Transformer (Appendix J.4). For CNNs, the $\ell_2$-norm flattens out more for increasing $\eta$ in comparison to the FCN. For the CNN with Batch Normalization, comparably higher learning rates still converge. We do not observe a qualitative change of the phenomena for the ResNet and ViT architectures.

The CNNs (LeCun et al., 1998) consist of two convolutional layers with 32 filters, each using $3 \times 3$ kernels, stride 1, and padding 1. Each convolution is followed by an activation function (ReLU or tanh) and a $2 \times 2$ maximum pooling operation. A fully connected layer after flattening maps the features to class logits. We further include an alternative architecture that applies batch normalization within the CNN.

The ResNet-20 model (He et al., 2016) consists of three residual layers, with three blocks per layer. Each block contains two $3 \times 3$ convolutions followed by batch normalization and ReLU activation. Between stages, spatial down-sampling is performed using average pooling. To match feature dimensions across residual connections, the skip paths are adjusted using batch normalization and zero-padding along the channel dimension.

The Vision Transformer (ViT) (Dosovitskiy et al., 2021) splits the input image into non-overlapping patches ($7 \times 7$ for MNIST, $4 \times 4$ for CIFAR-10), embeds each patch into a latent space (dimension 64 for MNIST, 128 for CIFAR-10), and processes the resulting sequences with transformer encoder layers (4 for MNIST; 6 for CIFAR-10), using 4 attention heads per layer. Each configuration includes a learnable class token and positional embeddings, and ends with a linear classifier applied to the class token output.

## I.3. Activation Function

We evaluate the effect of activation functions by comparing ReLU and tanh in fully connected networks on MNIST-5k (Appendix J.1.1, J.2.1) and on CIFAR-10-5k (Appendix J.1.2, J.2.2). Across all configurations, the sharpness–norm trade-off and the transition between flow-aligned and EoS regimes are consistently observed.

## I.4. Loss Function

We compare the behavior of cross-entropy (CE) and mean squared error (MSE) for both the base configuration and additional architectures, see Figure 6 for a comparison of the trade-off for MSE and CE on MNIST-5k for a ReLU CNN and Appendix J for all other setups.

Compared to MSE, the sharpness profile for varying $\eta$ when training with CE differs. In the flow-aligned phase, the final sharpness values for CE are still similar in magnitude but consistently below the maximum sharpness of its corresponding GF. In contrast, for MSE the final sharpness is at $s_{\text{GF}}$. The transition to the EoS regime still occurs approximately at $\eta = 2/s_{\text{GF}}$. For large $\eta$, the sharpness values remain below the $2/\eta$ curve but qualitatively still decrease as $\eta$ increases for the EoS regime.

We observe for the sharpness of the iterates during training that after an initial increase (progressive sharpening) and an oscillatory phase around $2/\eta$, the sharpness subsequently decreases again significantly. This phenomenon, originally

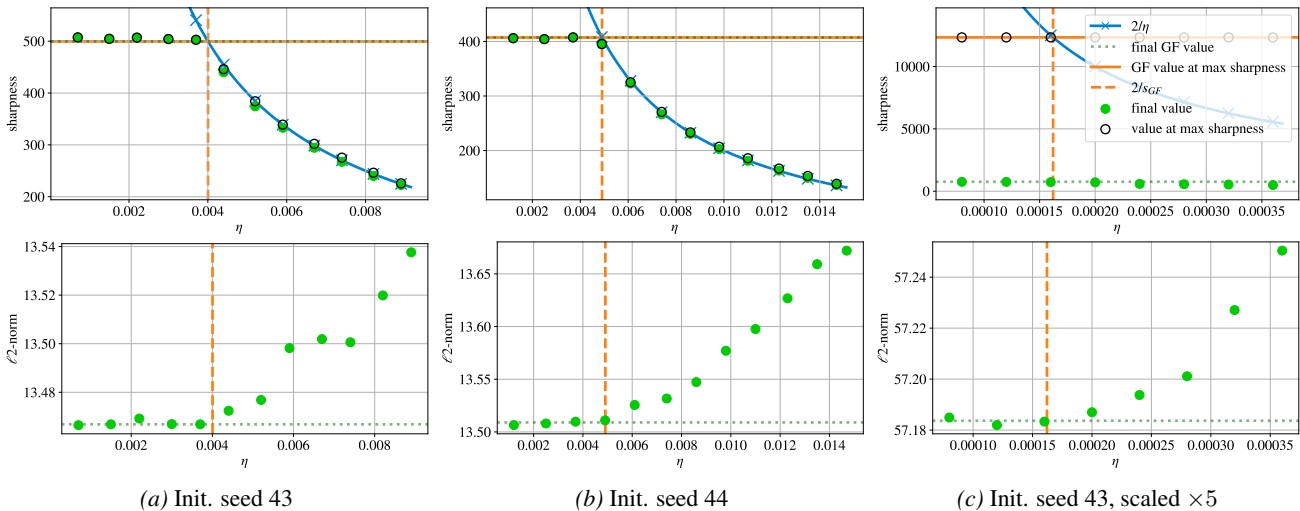

*(a)* Init. seed 43          *(b)* Init. seed 44          *(c)* Init. seed 43, scaled $\times 5$

*Figure 7.* Effect of varying initialization seed and scaling at initialization on the sharpness–norm trade-off. All columns show sharpness and $\ell_2$-norm curves for the same architecture (FCN-ReLU), dataset (CIFAR-10-5k), and loss function (MSE), all trained until loss 0.01. While the different seed does not affect the overall behavior, scaling disrupts adherence of solution sharpness to the $2/\eta$ curve. Effect on norm is however preserved.

remarked in Cohen et al. (2021), appears more pronounced in our results, as they used a higher loss-threshold beyond which the strong decrease starts occurring. Although the final sharpness values therefore do not follow the $2/\eta$ relationship, the training iterates rise toward this value and oscillate around it before the sharpness drops. In our plots, we visualize the smoothed sharpness around its maximum to highlight this trend. The effect during the training is illustrated in Figure 14 for selected learning rates.

Training with CE often fails to converge at learning rates even below $1/s_0$ ($s_0$ denoting the sharpness at initialization), while training with MSE often converges at comparatively higher values. This aligns with previous findings on the geometry of the log-loss landscape (Soudry et al., 2018), which indicate that the loss surface becomes flatter as the parameter norm increases. Because of the exponential in the CE loss equation, the loss decreases with growing parameter norm and, as a result, parameters only converge in direction. However, when the learning rate is too high early in the training, the high curvature of the loss landscape leads to instability or stagnation before this directional convergence effect.

### I.5. Loss Threshold

In Section 2, we show how the loss threshold $\varepsilon$ directly affects the critical learning rate $\eta_c$ at which (approximately) the sharpness–norm phase transition occurs, given by $2/s_{\mathrm{GF}}^{\varepsilon}$. This effect is illustrated in Figure 3 for an FCN with tanh activation on CIFAR-10-5k, trained with MSE loss. Comparing identical models trained to different loss thresholds, we observe that smaller $\varepsilon$ values yield higher $s_{\mathrm{GF}}^{\varepsilon}$, resulting in a lower $\eta_c$ and thus shifting the transition point between the flow-aligned and EoS regimes. We confirm this trend across multiple architectures in Appendix J.7.1.

This dependence on $\varepsilon$ is naturally related to early stopping: A higher loss threshold corresponds to a point before the model begins to overfit on the training set, where the test loss is still decreasing. In contrast, very small loss thresholds reflect the late phase of training, where the characteristic U-shaped test loss curve over time is evident. There, the training loss continues to drop, but the test loss increases slowly. By varying $\varepsilon$, we can thus study the sharpness and norm trade-offs under different degrees of overfitting. However, note that we do not link $\varepsilon$ to the validation loss, as it is commonly done when using early stopping as a regularizer during training.

### I.6. Initialization

We vary the initialization seed in fully connected networks trained on CIFAR-10-5k to test the sensitivity of the transition to random initialization, see Figure 7. While the critical learning rate $\eta_c$ shifts with initialization, due to a different initial sharpness $s_0$ and maximum of the flow trajectory $s_{\mathrm{GF}}$, the qualitative structure remains intact.

We also perform experiments with increased initialization scale, scaling all initial weights $\times 5$ and $\times 10$. As a result, the

maximal sharpness along the trajectory occurs already at initialization, which drastically alters the optimization dynamics and sharpness evolution. The sharpness decreases at first, and, if reaching the $2/\eta$ threshold, oscillates around this value. In general, the training is highly unstable, which leads to divergence of the training at many small learning rates. Still, the $\times 5$-scaled initializations result in somewhat similar qualitative behaviors in the observed values as our default scale. For $10\times$ scaling, the training diverges already at learning rates smaller than $\eta_c$. In addition, the final $\ell_2$-norm reaches very high values and decreases with increasing learning rate. These results suggest that the mechanism of implicit regularization differs at such large scales. We note that this aligns with previous works on EoS which often implicitly assume a sufficiently small initialization to permit progressive sharpening.

We provide further figures with varying initialization seeds and scales in Appendix J.7.2 and J.7.3, respectively.

### I.7. Parameterization

Different parameterizations of the forward pass are known to place training in qualitatively different regimes with respect to feature learning (Noci et al., 2024), which is why we test the norm-sharpness trade-off for this setup. We focus on the $\mu$P and kernel parameterizations (Yang et al., 2021; Jacot et al., 2018). The kernel parameterization corresponds to NTK-like scaling, where feature learning diminishes with width, while $\mu$P remains in the feature-learning regime with width-independent gradient magnitudes and transferable learning-rates for models of varying widths (Yang et al., 2021). Recent work by Noci et al. (2024) further suggests that the Hessian spectrum also transfers for $\mu$P.

Both used parameterizations use fully connected feed-forward networks with ReLU activations. Each hidden layer of width $n_l$ computes

$$h_l = \frac{1}{\sqrt{n_{l-1}}}\sigma\left(W_l h_{l-1}\right)$$

with weights initialized as $(W_l)_{ij} \sim \mathcal{N}(0, 1)$. In the kernel parameterization the final layer is obtained as $f(x) = W_L h_L$, while in the $\mu$P parameterization the logits are rescaled by the width of the last hidden layer $f(x) = \frac{1}{\sqrt{n_L}} W_L h_L$. This differs from the normal parameterization in all other experiments where the $1/\sqrt{n_{l-1}}$ factor in the forward pass is missing and the weights are initialized uniformly with variance $1/(3n_{l-1})$. The hypothesis spaces are the same in both settings, however the reparameterization changes the dynamics and is hence of interest with respect to implicit regularization.

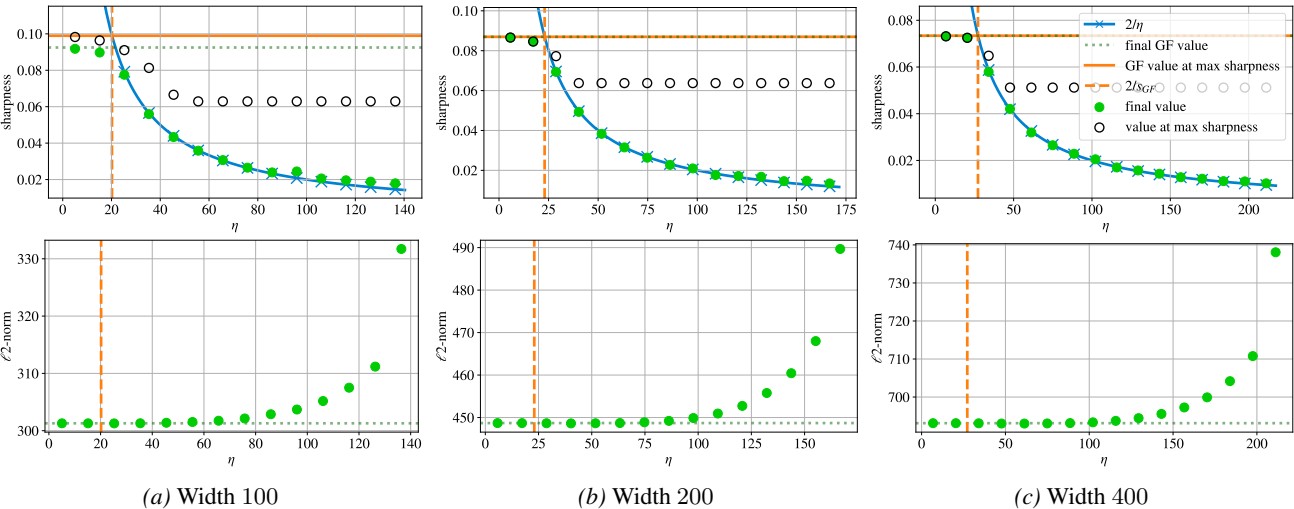

*(a)* Width 100        *(b)* Width 200        *(c)* Width 400

*Figure 8.* Sharpness (top row) and $\ell_2$-norm of final classifiers (bottom row) for $\mu$P parameterization with widths 100, 200, and 400 on MNIST-5k with MSE and loss goal 0.1.

For the $\mu$P parameterization, the sharpness plots (top row of Figure 8) show approximately constant sharpness for small learning rates and a decrease along the $2/\eta$ curve for larger learning rates, with similar values in the flow-aligned regime across widths. The $\ell_2$-norm plots (bottom row) reveal the usual pattern across widths of increasing final parameter $\ell_2$ for increasing learning rate. The absolute norms differ due to model size, but the growth of the norm as $\eta$ increases is approximately consistent (though divergence happens slightly earlier for smaller models). This is expected for the $\mu$P parameterization, as the parameter update magnitudes are independent of the model width. After rescaling the learning

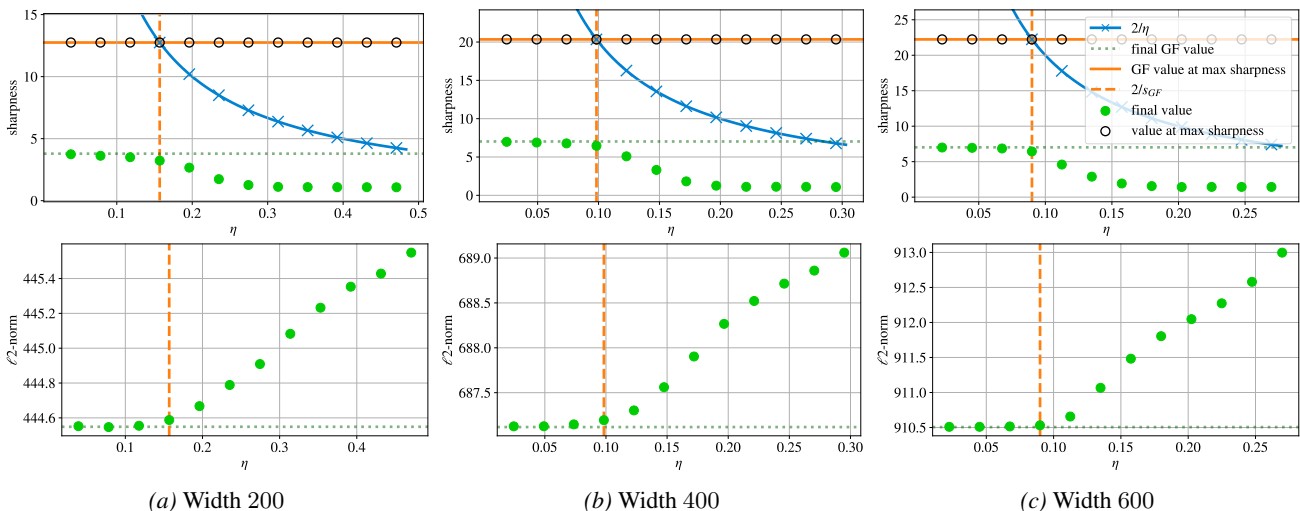

*Figure 9.* Sharpness (top row) and $\ell_2$-norm of final classifiers (bottom row) for kernel parameterization with widths 200, 400, and 600 on MNIST-5k with MSE and loss goal 0.1.

rate proportionally to width, the results align across the models of different widths which matches the results by Noci et al. (2024).

For the kernel parameterization we observe that the $\ell_2$-norm of the parameters (bottom row of Figure 9) remains stable for small learning rates and starts to increase once $\eta$ crosses the critical threshold, with the transition occurring at learning rates of the same order across widths[7]. The sharpness plots (top row) show that the maximum sharpness coincides with the sharpness at initialization, similar to the large-initialization experiments in Section I.6. Because of the different parameterization, sharpness no longer tracks the $2/\eta$ curve, yet the qualitative pattern is consistent across widths: sharpness stays flat below the threshold and decreases gradually thereafter.

### I.8. Number of Iterations

A notable difference between the two regimes lies in the relationship between learning rate and convergence speed. While the small learning rates of the flow-aligned regime lead to slower convergence in absolute terms, increasing the step size within this regime significantly accelerates optimization, with the number of iterations required to reach a fixed training loss decreasing at an approximate rate of $1/\eta$. As further shown in Section J.8.1, this rate of convergence speed acceleration with respect to the learning rate is higher in the flow-aligned regime than in the EoS regime.

### I.9. Alternative Norms and Sharpness Measures

We emphasize that our findings are robust to the choice of norm. Accordingly, the figures in the main text illustrate results across a variety of norms. To ensure consistency, the appendix figures primarily focus on the $\ell_1$-norm of the GD solution. In Figure 10, we compare the $\ell_1$-norm with the nuclear and $\ell_2$-norms, which exhibit qualitatively similar behavior. We provide additional comparisons in Appendix J.8.1.

Methodologically, we calculate the $\ell_1$- and $\ell_2$-norms using the flattened parameter vector. For the nuclear norm, we sum the nuclear norms of the individual weight matrices excluding bias.

---

[7]Note that the norm of the weight matrices (after adjusting for the different widths) differs slightly due to the randomness. The change in randomness is comparable to the variance indicated by experiments when changing the initialization seed, see Section I.6.

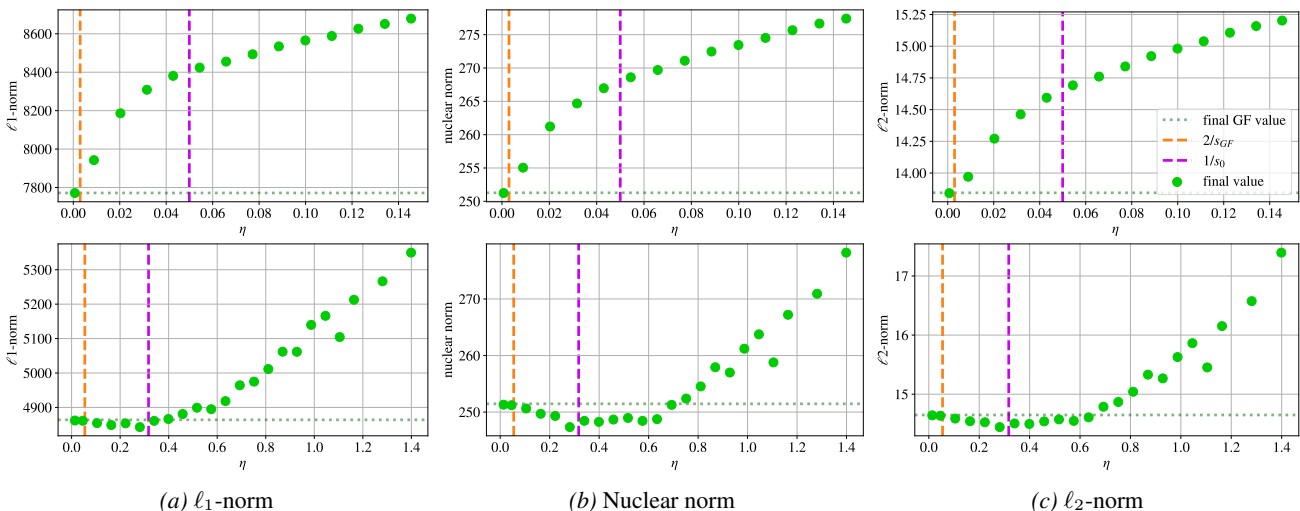

*Figure 10.* Each row shows the $\ell_1$-norm, the nuclear norm, and the $\ell_2$-norm of the solution for different models - both use FCN-ReLU with MSE loss, in the top row on CIFAR-10-5k, in the bottom row on MNIST-5k. As expected, the behavior of the different norms is approximately equivalent

Similarly, as our primary measure of sharpness we use throughout most of the paper the top eigenvalue of the loss Hessian. This notion of sharpness, though commonly used, has been shown to allow for being made arbitrarily large by means of reparameterization without affecting generalization (Dinh et al., 2017). This can make it ill-suited for studying connections to generalization performance. Therefore in Figure 11 we compare different notions of sharpness, including re-scaling invariant measures such as adaptive sharpness (Kwon et al., 2021), showing they share the overall decreasing behavior in the EoS regime similar to the worst-case sharpness.

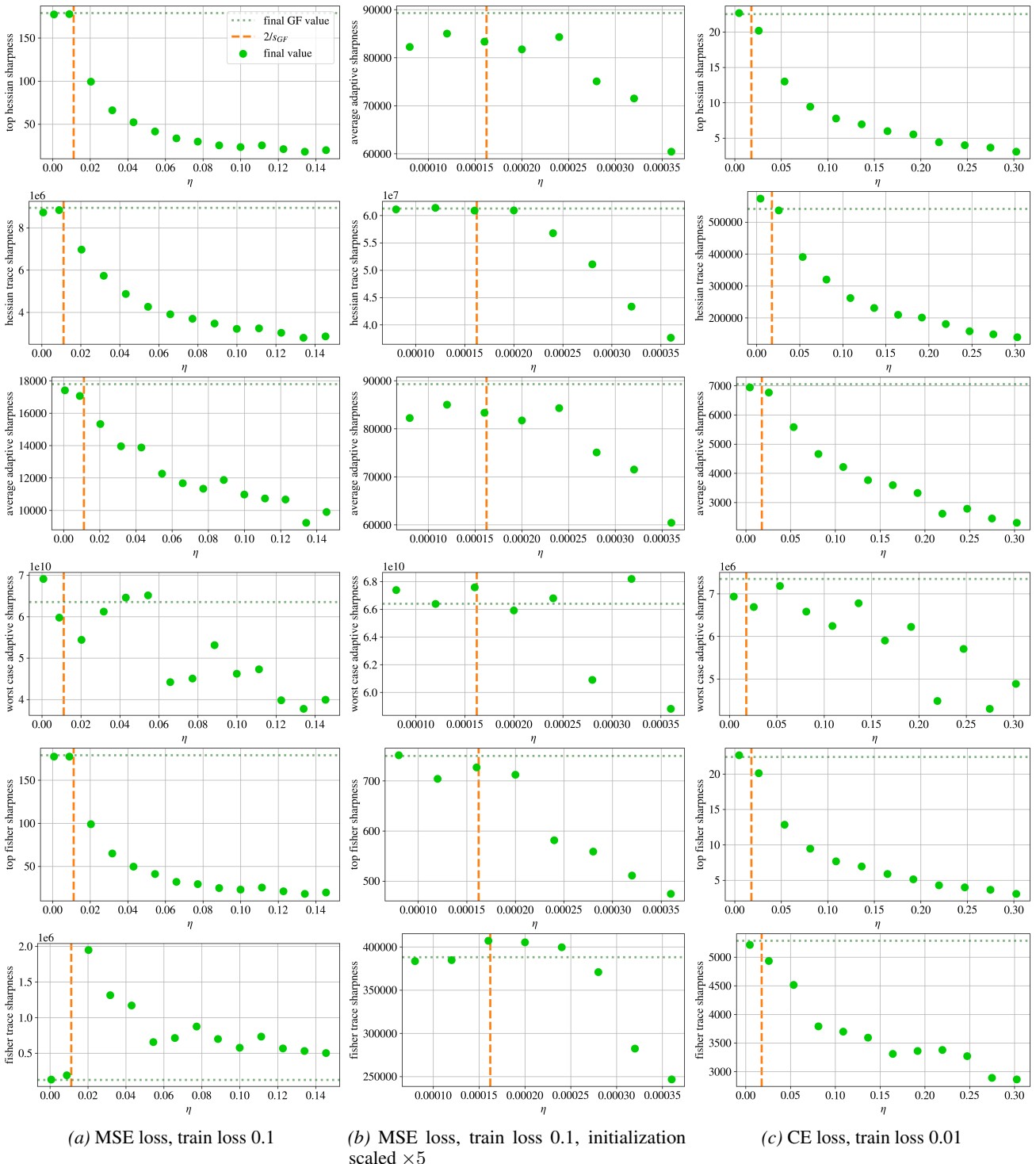

*(a)* MSE loss, train loss 0.1

*(b)* MSE loss, train loss 0.1, initialization scaled ×5

*(c)* CE loss, train loss 0.01

*Figure 11.* Each column represents a different setting: All display an FCN-ReLU network on CIFAR-10-5k, but in the first we show MSE loss with standard initialization, in the second MSE loss with scaled initialization and in the last CE loss. Each row shows a different measure of sharpness. Top to bottom these are: top eigenvalue of the loss Hessian (used throughout the paper), trace of the loss Hessian, average-case and worst-case adaptive sharpness (Kwon et al., 2021), and top eigenvalue and trace of the Fisher information matrix (Liang et al., 2019). Note that all measures display a general decreasing behavior with the exception of the Fisher trace on standard MSE loss (bottom left), where there is a sharp increase around the critical threshold $\eta_c$, from which the decreasing behavior starts. The scaled experiments show slightly more irregularity, but still preserve this general decrease.

### I.10. Gradient Descent Solution Distance

We measure the distance between the final solutions of GF and GD across different learning rates. This analysis provides insight into how closely GD tracks the continuous-time dynamics and how this relationship evolves as we move through the flow-aligned and EoS regimes.

In Figure 12, we show this relationship for two of our standard models. Comparing this with Figure 10, we can see that even though the qualitative behavior of the $\ell_1$-norm and $\ell_1$-distance from the GF solution are nearly equal, the distance of solutions for $\eta < \eta_c$ is already relatively high. This suggests that while in the flow-aligned regime, GD reaches solutions of similar sharpness and norm as GF, in absolute terms these solutions are non-negligibly different. Furthermore, comparing the scales of the two figures shows, that the increase in distance from the GF solution is much larger than the increase in absolute $\ell_1$-norm. Therefore, increasing the learning rate within the EoS regime likely results in movement of the solution in a direction more misaligned with the GF solution than the origin. Section J.8.1 shows this for further configurations.

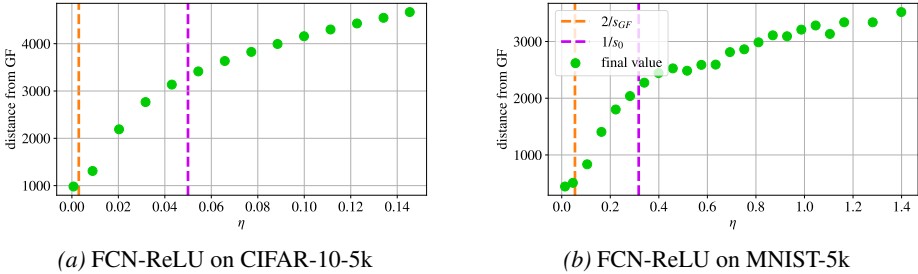

*(a)* FCN-ReLU on CIFAR-10-5k                    *(b)* FCN-ReLU on MNIST-5k

*Figure 12.* $\ell_1$-distance of the GD solution from the GF solution. Not to be confused with distance from the GF trajectory - here we measure only final values. On both examples we can see an increasing behavior similar to that of solution $\ell_1$-norm.

Additionally, in Figure 13 we compare the parameter $\ell_1$-norm to the $\ell_1$-distance from the untrained model at initialization. When examining this quantity for the final learned models plotted against the learning rate, the distance from initialization shows a similar qualitative trend as the parameter norm. In the flow-aligned regime, the distance to initialization is still approximately constant, before robustly increasing in the EoS regime. This is consistent with what can be expected since the models are initialized small relative to the norm of the final parameters.

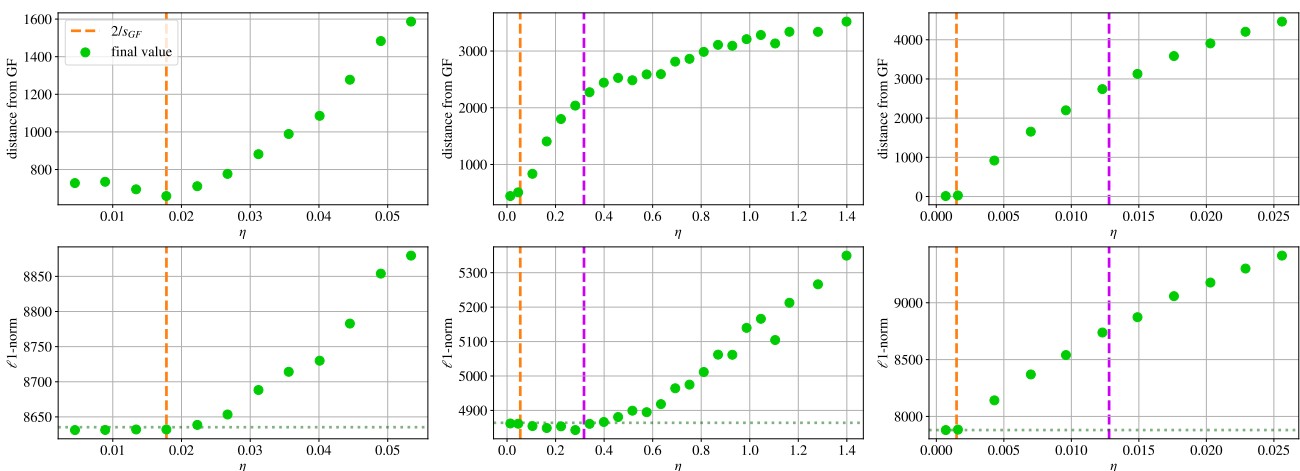

*(a)* FCN-ReLU on CIFAR-10-5k with CE loss  *(b)* FCN-ReLU on MNIST-5k with MSE loss  *(c)* FCN-tanh on CIFAR-10-5k with MSE loss

*Figure 13.* The top row shows for each setting the $\ell_1$-distance of the final models from their initialization, while the bottom row shows the absolute norm. As expected, the qualitative behavior remains almost identical.

### I.11. Evolution during Training

In Figure 14, we illustrate how sharpness, $\ell_1$-norm and loss evolve over the course of training in intrinsic time, i.e $\eta \cdot \#$ iterations. The sharpness increases initially (progressive sharpening) until reaching $2/\eta$, and then oscillates around this value. For very small learning rates, the increase stops earlier (aligned with the maximum sharpness of the corresponding GF). The norm rises without oscillation, suggesting that the oscillation occurs along a direction that preserves the parameter norm. The norm grows faster for larger learning rates. The loss decreases monotonically at first, then with oscillation after the sharpness has risen to $2/\eta$. In contrast to MSE loss, for training with CE loss, the sharpness decreases again after a period of oscillation. These dynamics in sharpness and loss were first systematically studied by Cohen et al. (2021). Our primary focus is on the dependence of final values on the learning rate, which complements these observations.

Similar to Figure 13, we compare the evolution of the parameter norm and the distance to the initialization in the second and third row of Figure 14. We observe that the distance follows closely a translated and scaled version of the parameter norm's trajectory. It naturally starts at $0$ and then grows significantly before entering the Edge of Stability. In comparison to the parameter norm evolution, here the rate of growth slows down to a larger extent after entering EoS, which supports the intuition that the chaotic EoS updates have a smaller cumulative effect on the solution's magnitude.

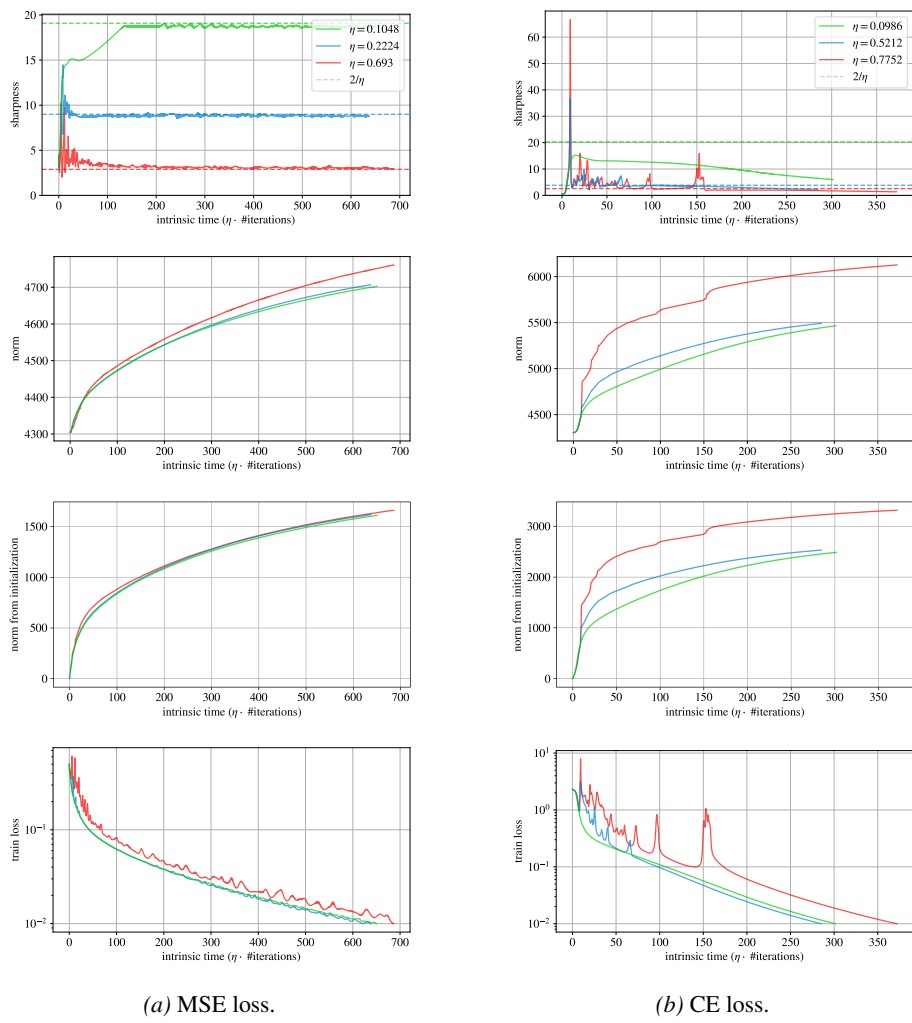

*(a)* MSE loss.                *(b)* CE loss.

*Figure 14.* For three different learning rates, we display the sharpness, $\ell_1$-norm, norm from initialization and train loss for both MSE (left) and CE loss (right column), both on MNIST-5k, FCN-ReLU, loss goal $0.01$. All plots are in intrinsic time, i.e. $\eta \cdot \#$iterations. We clearly observe the progressive sharpening and oscillations once the sharpness reaches $2/\eta$. For CE loss, the sharpness drops after an oscillatory phase.

## I.12. Per-layer Norms

In Figure 15 we present the per-layer norms when training the standard ReLU FCN on MNIST-5k and CIFAR-10-5k. All layers show an increasing trend. As one might expect, the increase is relative to the number of parameters of the respective layer. In particular, the input layer shows the largest increase.

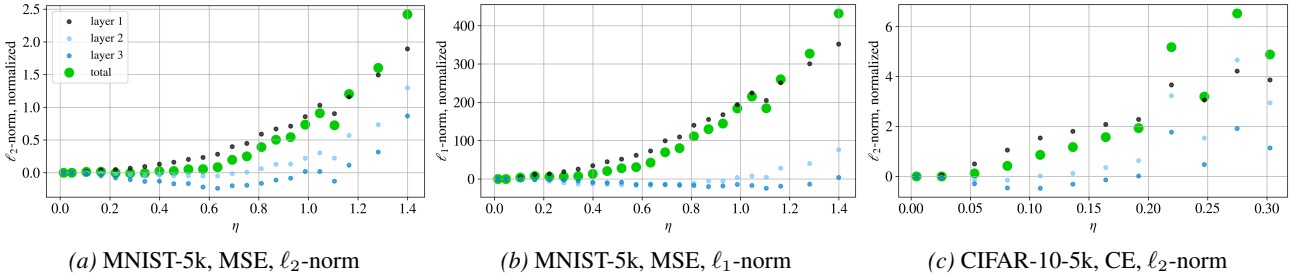

*(a)* MNIST-5k, MSE, $\ell_2$-norm          *(b)* MNIST-5k, MSE, $\ell_1$-norm          *(c)* CIFAR-10-5k, CE, $\ell_2$-norm

*Figure 15.* Layer-wise norms of the final solution on our ReLU-FCN on MNIST-5k and CIFAR-10-5k for different learning rates. We individually normalize each group by subtracting the value of the norm at the smallest learning rate. All layers show an increasing trend, which is relative to the layer size.

## I.13. The Diagonal Network

For the diagonal network discussed in Section 3, we present the sharpness, norm, and generalization values for different learning rates in Figure 16. We can explicitly compute the $\ell_1$-norm on the solution manifold under the sharpness constraint $2/\eta$, yielding the predicted line in Figure 16b. We emphasize that these curves look qualitatively similar to the more realistic models on MNIST and CIFAR-10 described throughout the empirical experiments section. Note that divergence occurs already for learning rates $\eta$ below the theoretical divergence threshold when the sharpness of all points on the solution manifold is above $2/\eta$.

We model generalization using a simple Gaussian data distribution (see Appendix F), which produces an (idealized) U-shaped curve, consistent with the behavior observed for many other realistic setups.

In Figure 17, we provide all trajectories of the iterates (cf. Figure 5).

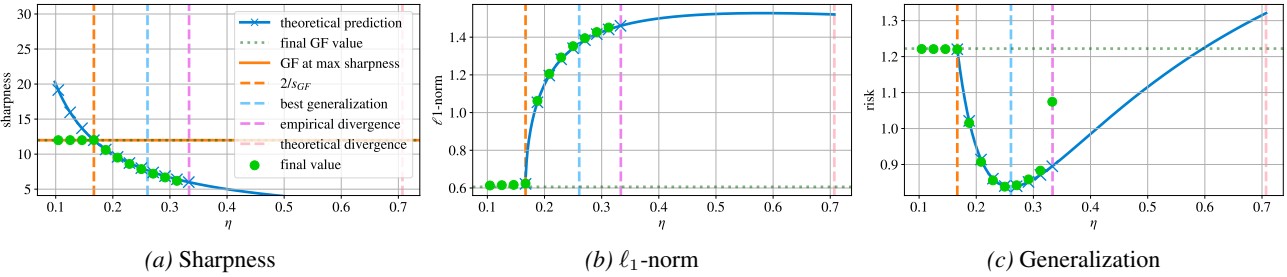

*(a)* Sharpness                  *(b)* $\ell_1$-norm                  *(c)* Generalization

*Figure 16.* Final sharpness, $\ell_1$-norm and generalization of a two-dimensional diagonal linear network with weight sharing, described in Section 3. The behavior corresponds to that of more realistic models studied throughout the paper.

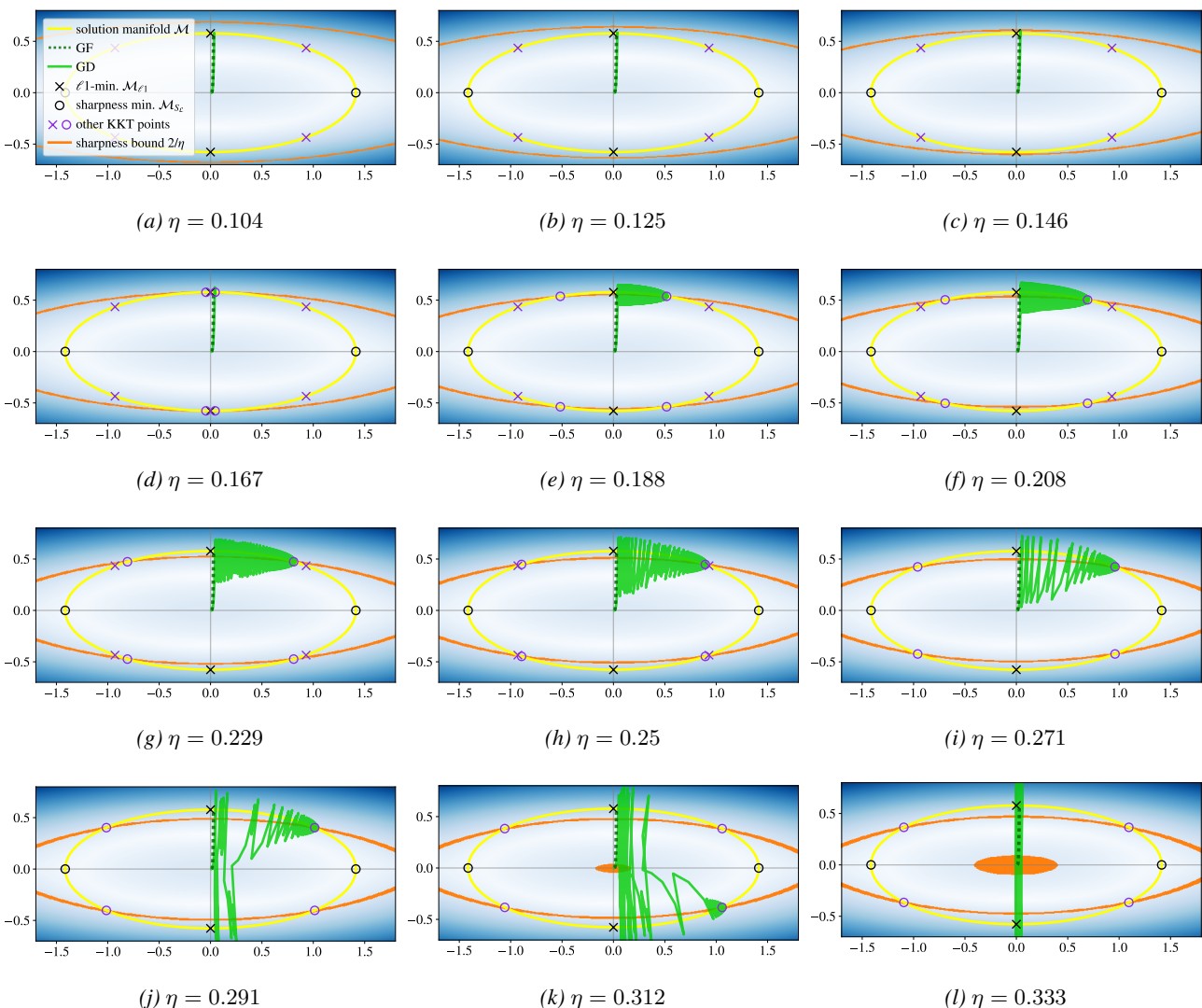

*Figure 17.* Iterates of weights of the two-dimensional diagonal linear network throughout training, for increasing learning rate. There is a clear distinction between the flow-aligned regime (17a)-(17d), where GD closely tracks the GF trajectory, and the EoS regime (17e)-(17l), where at some point GD begins to oscillate away from GF, until converging to one of the first solutions whose sharpness is less than $2/\eta$ (intersection of the yellow solution manifold $\mathcal{M}$ and orange sharpness bound $2/\eta$). This aligns with the intuition stemming from Theorem B.2. Note that for the largest learning rate tested, shown in Figure 17l, the dynamics are highly unstable. The trajectory converges to a point with sharpness strictly smaller than $2/\eta$, not an interpolator of minimal norm satisfying the sharpness constraints. In purple, we mark the KKT points from Lemma F.1 and the background color map represents loss sharpness from low (white) to high (blue).

## I.14. Other Optimizers

While our study is restricted to full-batch gradient descent to isolate as cleanly as possible the trade-off between norm bias and sharpness bias induced purely by the learning rate $\eta$, we also include evidence that the effect is present with other optimizers. Figure 18 shows stochastic gradient descent, Figure 19 the optimizer Shampoo (Gupta et al., 2018) and Figure 20 Muon (Jordan et al., 2024). For MNIST-5k with Shampoo and Muon, the sharpness increases again for larger learning rates, but for all other cases the qualitative trade-off is also visible. Notably, for CIFAR-10-5k, all the minima found for Shampoo have a smaller test loss compared to the gradient flow solution. For Muon, large learning rates still converge, but lead to exploding test loss with CE. We emphasize that for small $\eta$, values do not have to match the gradient flow solution.

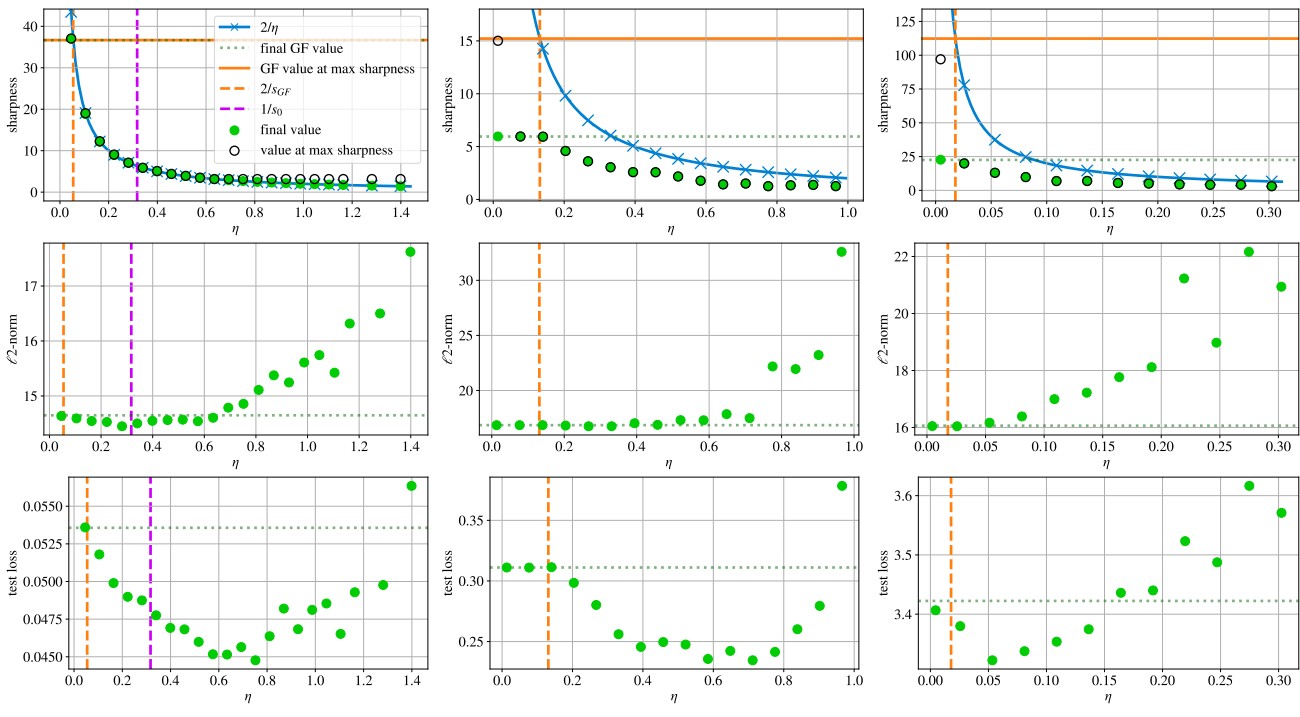

*(a)* MNIST-5k, MSE, SGD, train loss 0.0001    *(b)* MNIST-5k, CE, SGD, train loss 0.01    *(c)* CIFAR-10-5k, CE, SGD, train loss 0.01

*Figure 18.* We show the sharpness, $\ell_2$-norm and test loss for our base set-ups, but trained with stochastic gradient descent.

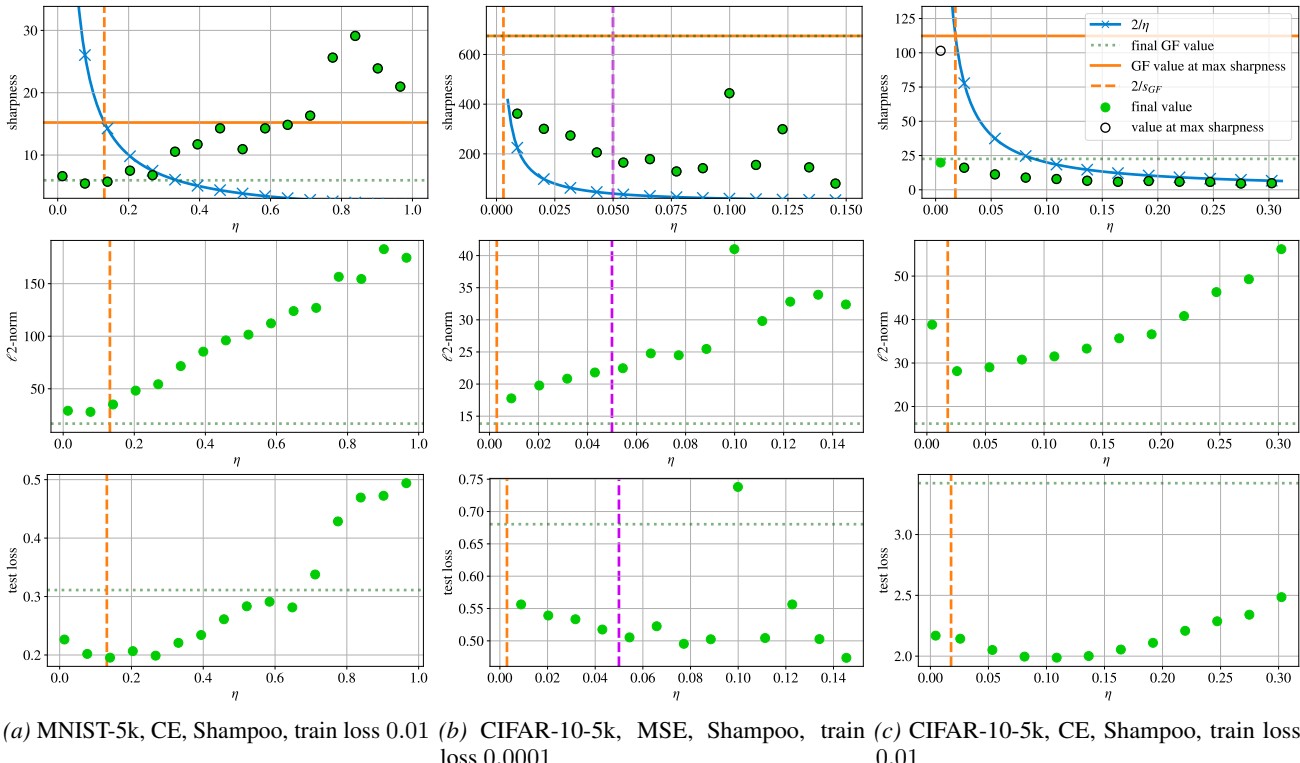

*(a)* MNIST-5k, CE, Shampoo, train loss 0.01    *(b)* CIFAR-10-5k, MSE, Shampoo, train loss 0.0001    *(c)* CIFAR-10-5k, CE, Shampoo, train loss 0.01

*Figure 19.* We show the sharpness, $\ell_2$-norm and test loss for our base set-ups, but trained with the optimizer Shampoo.

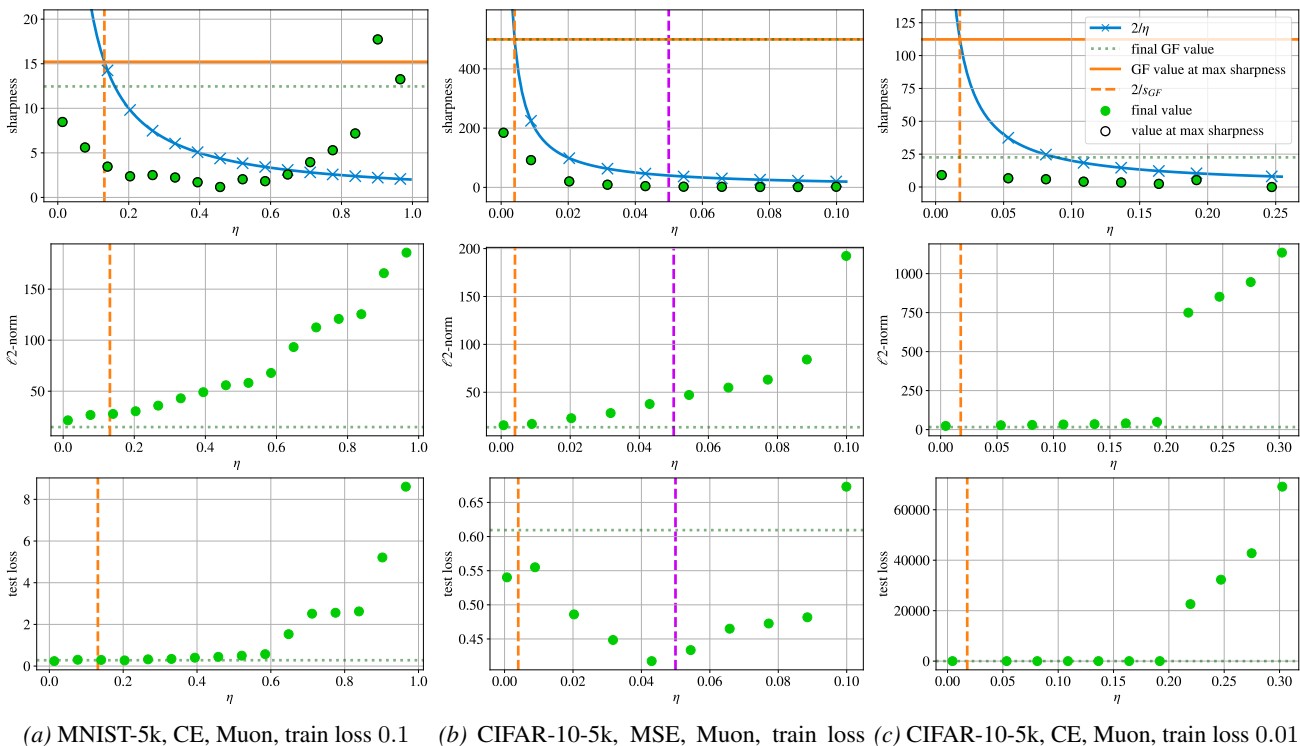

*(a)* MNIST-5k, CE, Muon, train loss 0.1 *(b)* CIFAR-10-5k, MSE, Muon, train loss *(c)* CIFAR-10-5k, CE, Muon, train loss 0.01
0.01

*Figure 20.* We show the sharpness, $\ell_2$-norm and test loss for our base set-ups, but trained with the optimizer Muon.

### I.15. Other Data Modalities

While the systematic evaluation presented in this paper focuses on the image domain, we also include examples suggesting that the observed trade-off is not limited to images. We consider a synthetic sequence-reversal task and two tabular tasks, one for binary classification and one for regression.

For the sequence domain, we use a synthetic sequence-reversal task with a fixed sequence length 10 and vocabulary size 9. Each input is a sequence of 10 tokens sampled uniformly from $\{1, \ldots, 9\}$. The target is its exact reversal. We train using teacher forcing. The model is a standard encoder-decoder transformer (Vaswani et al., 2017) with two encoder and two decoder layers, each using four attention heads, a model dimension of 64, and a feed-forward width of 128. The inputs pass through learned token embeddings and fixed sinusoidal positional encodings, and the decoder uses a causal mask for autoregressive prediction. A linear layer maps decoder outputs to vocabulary logits.

For the tabular tasks, we use the California Housing (regression) and Breast Cancer (classification) dataset by scikit-learn (Pedregosa et al., 2011). The California Housing dataset contains aggregated demographic and housing features (e.g., average number of rooms) and the target is the median house value. The Breast Cancer Wisconsin dataset contains 30 cell nuclei features such as radius or texture, and the goal is to identify whether a tumor sample is malignant or benign. For both datasets, we standardize all input features by subtracting the training-set mean and dividing by the training-set standard deviation for each feature dimension, and we apply the same transformation to the targets. The model is our standard feed-forward network with two-hidden layers and width 200.

For both data modalities, we observe a similar characteristic trade-off of sharpness and norm which we show in Figure 21. In contrast, for the reverse sequence, the sharpness value is not constant but increases also for small learning rates.

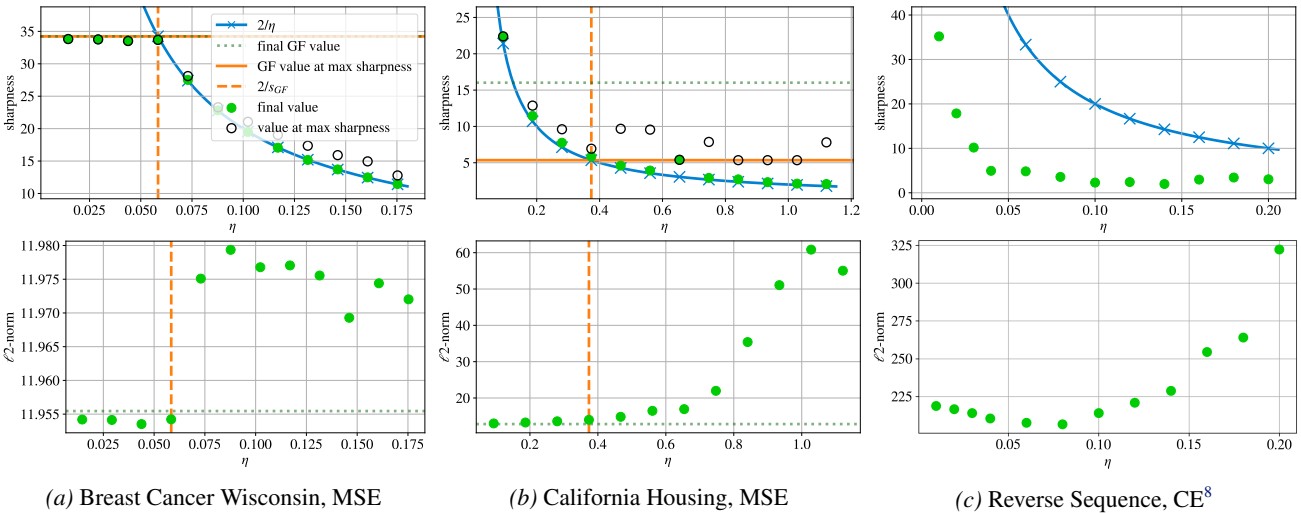

*(a)* Breast Cancer Wisconsin, MSE   *(b)* California Housing, MSE   *(c)* Reverse Sequence, CE[8]

*Figure 21.* We show the sharpness and $\ell_2$-norm for two tabular and a sequence-to-sequence data set. This indicates that our results extend beyond the image domain.

## J. Systematic Overview of Experiments

All performed experiments are summarized in Table 1. For most of these configurations, we present both coarse and fine-grained learning rate schedules to emphasize the transition region between flow-aligned and EoS regime around $\eta_c$, as well as the behavior at larger learning rates, demonstrating the trade-off between increasing $\ell_2$-norm and decreasing sharpness for varying the learning rate. Table 1 specifies for each setting the following attributes:

- **Model.** We state the model architecture (see Section I.2) and activation used. For the FCN models where we vary width and depth, we also indicate the size. When we do not specify a size, we refer to the standard architecture of $200 \times 2$.

- **Dataset.** MNIST or CIFAR-10, with the "-5k" suffix indicating that we train only on the first 5000 data points of the train set, while still testing on the full test set.

- **Loss.** Mean squared error (MSE) or cross-entropy (CE).

- **Seed.** The random seed used for generating weights at initialization. For experiments using a scaled initialization, the scaling factor is given.

- **Loss Goal.** We stop training gradient flow and gradient descent for each learning rate upon reaching this train loss value.

- **U-Shape.** For each setting we state whether optimal test loss aligns with either learning rate extreme, indicating a generalization advantage of either low-norm or low-sharpness bias. Settings where the optimum is attained for mid-range learning rates are marked by ✓, settings with an alignment towards either extreme by ✗, and somewhat inconclusive settings by either mark in brackets. In our experiments, in all cases with a clear optimum extreme alignment, the alignment is always towards high learning rates, that is, towards low sharpness solutions.

- **Figures.** List of figures throughout the paper where the respective setting appears.

In the main part of the systematic review, we present for each setting sharpness, $\ell_2$-norm and test loss plots, for both a fine-grained set of learning rate values focused around the critical threshold and a coarse set showing large-scale behaviors. In the plots we show

- the final respective value attained for each learning rate represented by green dots;

---

[8]We do not include the GF lines as we only run GD for this setup.

- a horizontal dotted green line indicating the final value reached by the gradient flow;

- a vertical dashed orange line showing the critical learning rate threshold of $2/\eta_{GF}$, for the transition from the flow-aligned to the EoS regime;

- for coarse-grained plots, a vertical dashed purple line, indicating the inverse value of sharpness at initialization, which has been proposed as a heuristic for learning rate initialization, if the line is missing this means that the GD did not converge for such learning rate;

- for sharpness plots, the $2/\eta$ curve, for $\eta$ being the learning rate variable, shown in blue with crosses at each used learning rate value;

- for sharpness plots, the maximum value reached throughout training, indicated by black circles;

- for sharpness plots, a horizontal orange line showing the maximal GF sharpness.

*Table 1.* Full list of experimental configurations.

| Model | Dataset | Loss | Seed | Loss Goal | U-Shape | Figures |
|---|---|---|---|---|---|---|
| FCN-ReLU | MNIST-5k | MSE | 43 | 0.0001 | ✓ | 4a,14,10,12b, 22,68,76 |
| FCN-ReLU | MNIST-5k | MSE | 43 | 0.001 | ✓ | 55 |
| FCN-ReLU | MNIST-5k | MSE | 43 | 0.01 | ✓ | 56 |
| FCN-ReLU | MNIST-5k | MSE | 43 | 0.1 | ✓ | 57 |
| FCN-ReLU | MNIST-5k | CE | 43 | 0.01 | ✓ | 23,69,77 |
| FCN-ReLU | MNIST-5k | CE | 43 | 0.1 | ✓ | 58 |
| FCN-ReLU | CIFAR-10-5k | MSE | 43 | 0.0001 | × | 2a,4c,10,12a, 24,72,80 |
| FCN-ReLU | CIFAR-10-5k | MSE | 43 | 0.001 | × | 59 |
| FCN-ReLU | CIFAR-10-5k | MSE | 43 | 0.01 | × | 7a,60 |
| FCN-ReLU | CIFAR-10-5k | MSE | 43 | 0.1 | (×) | 61,11 |
| FCN-ReLU | CIFAR-10-5k | MSE | 44 | 0.01 | × | 7b,63 |
| FCN-ReLU | CIFAR-10-5k | MSE | 45 | 0.01 | × | 64 |
| FCN-ReLU | CIFAR-10-5k | MSE | 43, ×5 | 0.1 | × | 7c,65,11 |
| FCN-ReLU | CIFAR-10-5k | CE | 43 | 0.01 | ✓ | 4b,25,73,81,11 |
| FCN-ReLU | CIFAR-10-5k | CE | 43 | 0.1 | ✓ | 62 |
| FCN-ReLU | CIFAR-10-5k | CE | 43, ×5 | 0.01 | × | 66 |
| FCN-ReLU | CIFAR-10-5k | CE | 43, ×10 | 0.01 | × | 67 |
| FCN-ReLU | MNIST | MSE | 43 | 0.01 | ✓ | 2b,26,70,78 |
| FCN-ReLU | MNIST | CE | 43 | 0.01 | (✓) | 27,71,79 |
| FCN-ReLU | CIFAR-10 | CE | 43 | 0.1 | × | 28 |
| FCN-ReLU 400 × 2 | MNIST-5k | MSE | 43 | 0.01 | × | 41 |
| FCN-ReLU 600 × 2 | MNIST-5k | MSE | 43 | 0.01 | (×) | 42 |
| FCN-ReLU 2000 × 2 | MNIST-5k | MSE | 43 | 0.01 | × | 43 |
| FCN-ReLU 200 × 4 | MNIST-5k | MSE | 43 | 0.01 | (×) | 44 |
| FCN-ReLU 200 × 6 | MNIST-5k | MSE | 43 | 0.01 | (✓) | 45 |
| FCN-ReLU 400 × 4 | MNIST-5k | MSE | 43 | 0.01 | × | 46 |
| FCN-ReLU 600 × 6 | MNIST-5k | MSE | 43 | 0.01 | (✓) | 47 |
| FCN-ReLU 400 × 2 | CIFAR-10-5k | MSE | 43 | 0.01 | × | 48 |
| FCN-ReLU 600 × 2 | CIFAR-10-5k | MSE | 43 | 0.01 | × | 49 |
| FCN-ReLU 2000 × 2 | CIFAR-10-5k | MSE | 43 | 0.01 | × | 50 |
| FCN-ReLU 200 × 4 | CIFAR-10-5k | MSE | 43 | 0.01 | ✓ | 51 |
| FCN-ReLU 200 × 6 | CIFAR-10-5k | MSE | 43 | 0.01 | ✓ | 52 |
| FCN-ReLU 400 × 4 | CIFAR-10-5k | MSE | 43 | 0.01 | ✓ | 53 |
| FCN-ReLU 600 × 6 | CIFAR-10-5k | MSE | 43 | 0.01 | ✓ | 54 |
| FCN-tanh | MNIST-5k | MSE | 43 | 0.1 | × | 29 |
| FCN-tanh | MNIST-5k | CE | 43 | 0.01 | (✓) | 30 |
| FCN-tanh | CIFAR-10-5k | MSE | 43 | 0.001 | × | 3c,31,74,82 |
| FCN-tanh | CIFAR-10-5k | MSE | 43 | 0.01 | × | 3b |
| FCN-tanh | CIFAR-10-5k | MSE | 43 | 0.1 | (×) | 3a |
| FCN-tanh | CIFAR-10-5k | CE | 43 | 0.01 | ✓ | 32,75,83 |
| CNN-ReLU | MNIST-5k | MSE | 43 | 0.1 | ✓ | 6a,33 |
| CNN-ReLU | MNIST-5k | CE | 43 | 0.01 | ✓ | 34 |
| CNN-ReLU | MNIST | MSE | 43 | 0.1 | (×) | 6b,35 |
| CNN-ReLU | MNIST | CE | 43 | 0.01 | ✓ | 36 |
| CNN-ReLU BN | CIFAR-10-5k | CE | 43 | 0.01 | ✓ | 6c,37 |
| ViT-ReLU | MNIST-5k | CE | 43 | 0.1 | (✓) | 2c,38 |
| ViT-ReLU | CIFAR-10-5k | CE | 43 | 1 | (✓) | 39 |
| ResNet20-ReLU | CIFAR-10-5k | CE | 43 | 0.1 | (×) | 40 |

## J.1. FCNs with ReLU Activation

### J.1.1. ON MNIST-5K

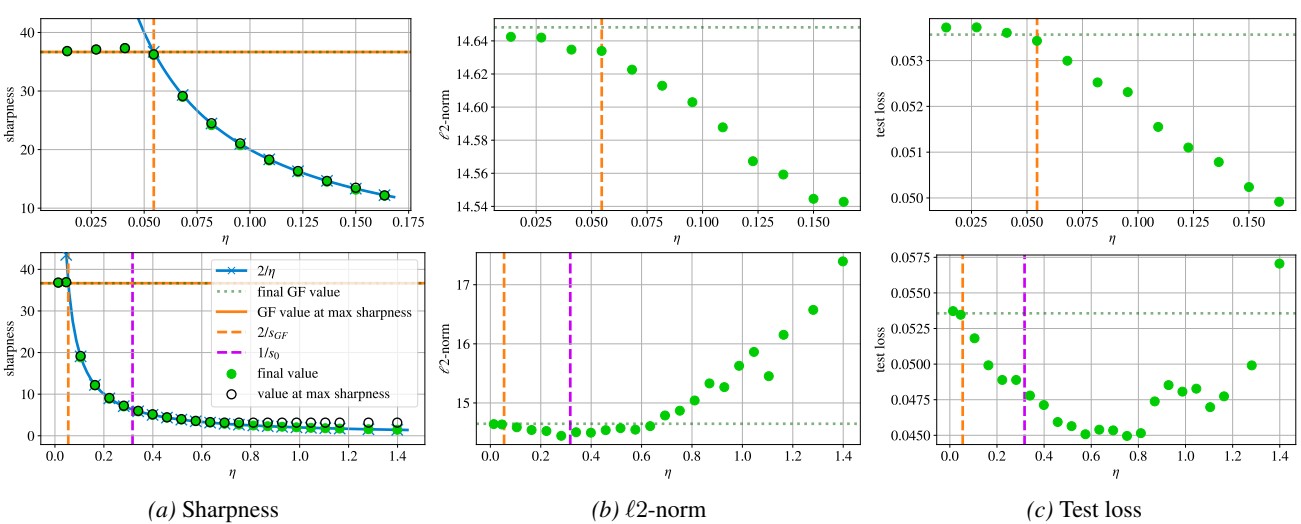

*(a)* Sharpness          *(b)* $\ell$2-norm          *(c)* Test loss

*Figure 22.* **MSE loss.** FCN-ReLU, MNIST-5k, train loss 0.0001. Both rows show the same setting, but different ranges of learning rate $\eta$ - the top row includes the fine grid, focused on the transition from the flow-aligned to the EoS regime, while the coarse grid in the bottom row displays more large-scale behavior, going typically up to diverging learning rates.

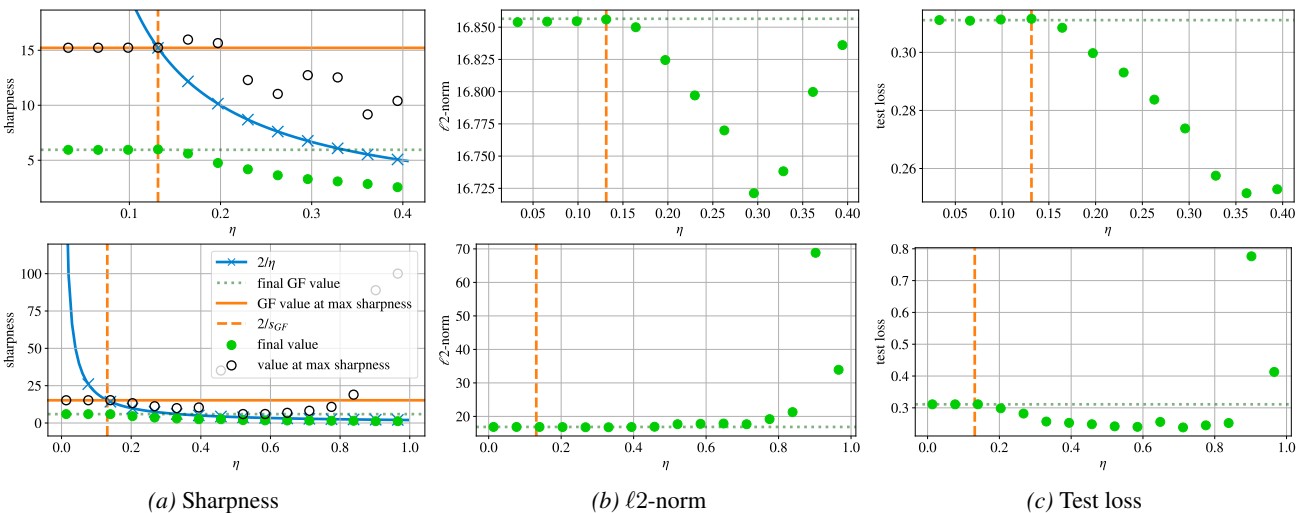

*(a)* Sharpness          *(b)* $\ell$2-norm          *(c)* Test loss

*Figure 23.* **CE loss.** FCN-ReLU, MNIST-5k, train loss 0.01

## J.1.2. ON CIFAR-10-5K

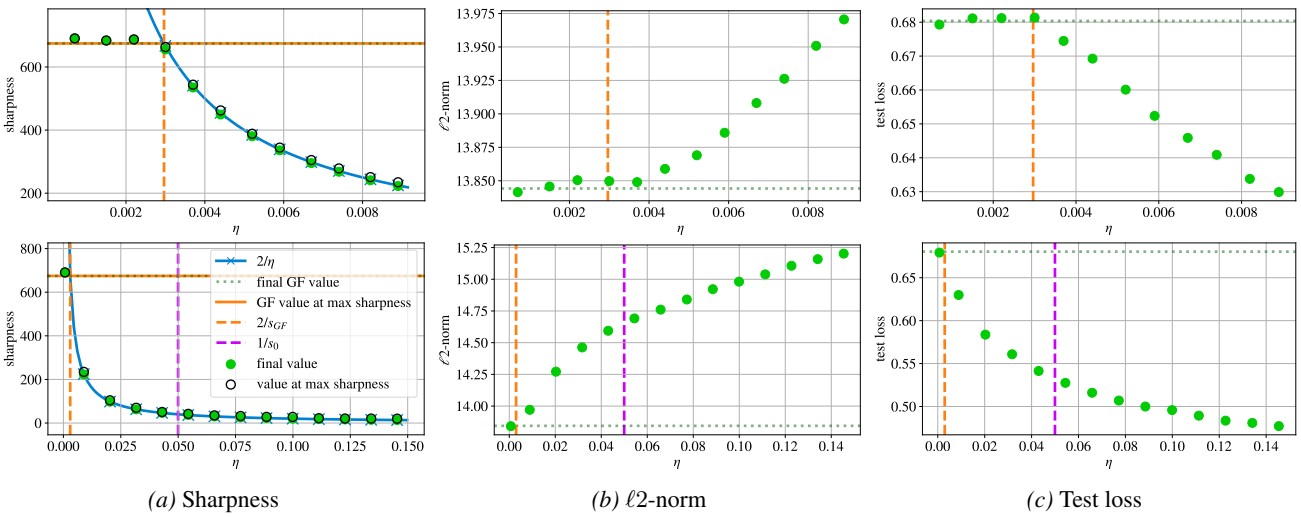

*(a)* Sharpness      *(b)* $\ell 2$-norm      *(c)* Test loss

*Figure 24.* **MSE loss.** FCN-ReLU, CIFAR-10-5k, train loss 0.0001

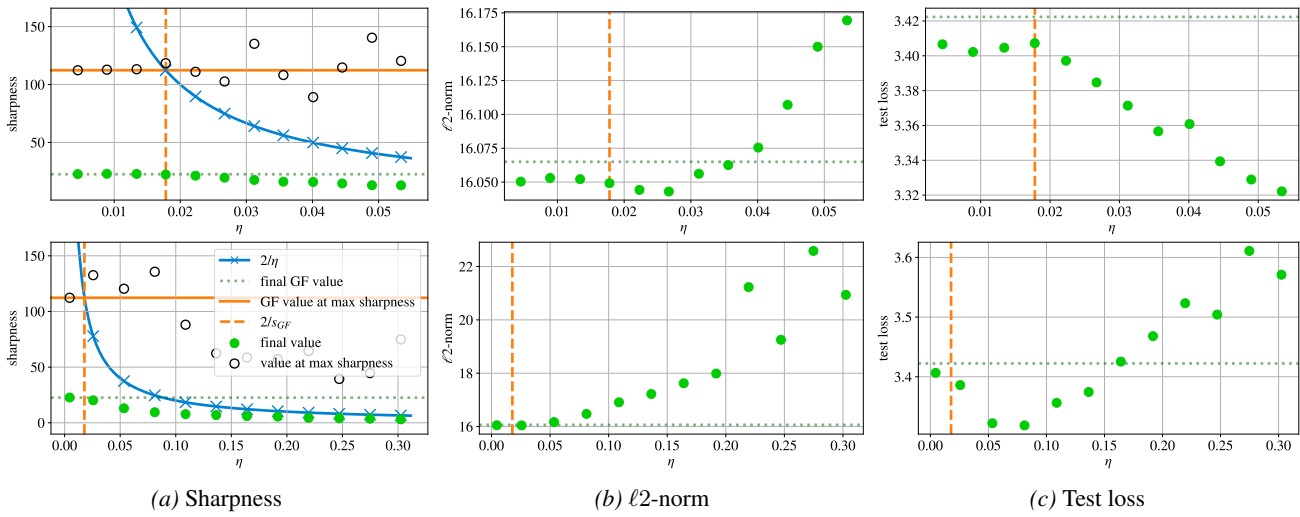

*(a)* Sharpness      *(b)* $\ell 2$-norm      *(c)* Test loss

*Figure 25.* **CE loss.** FCN-ReLU, CIFAR-10-5k, train loss 0.01

## J.1.3. ON FULL MNIST

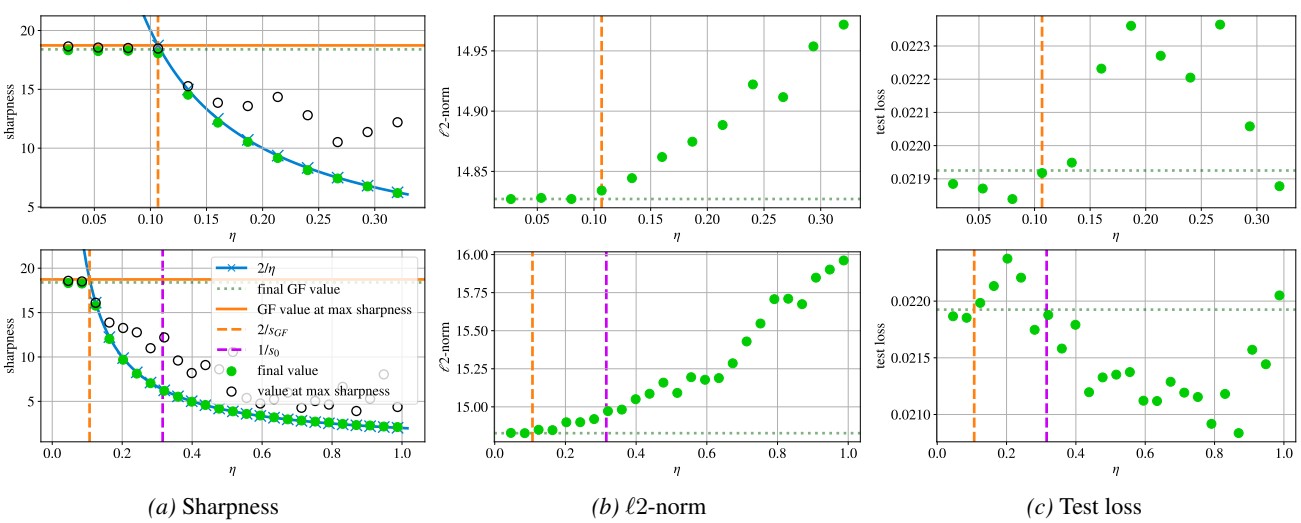

*Figure 26.* **MSE loss.** FCN-ReLU, MNIST, train loss 0.01

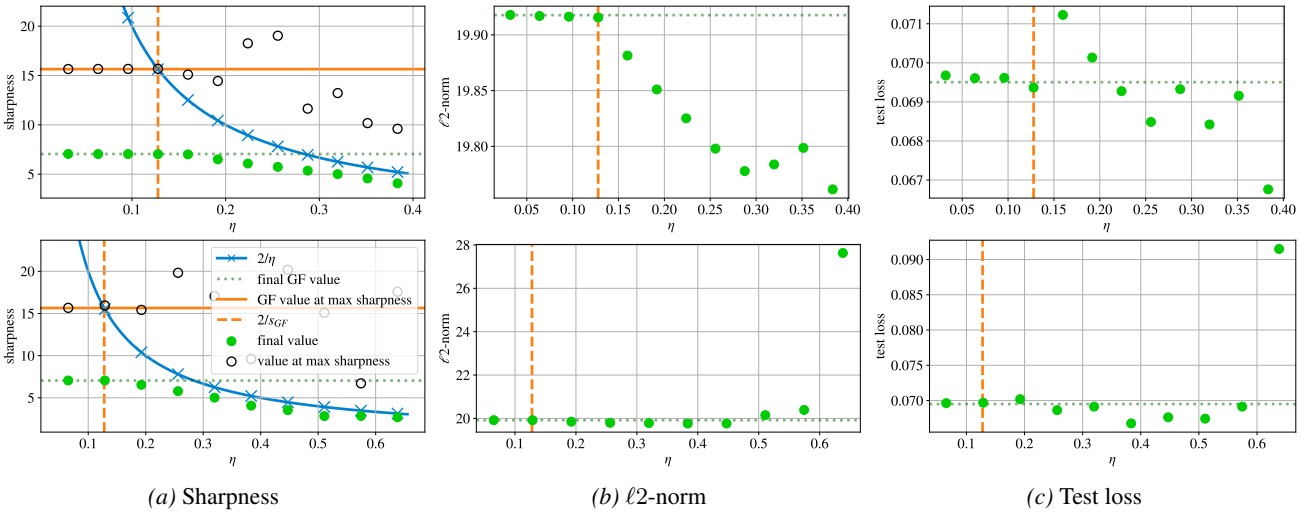

*Figure 27.* **CE loss.** FCN-ReLU, MNIST, train loss 0.01

### J.1.4. ON FULL CIFAR-10

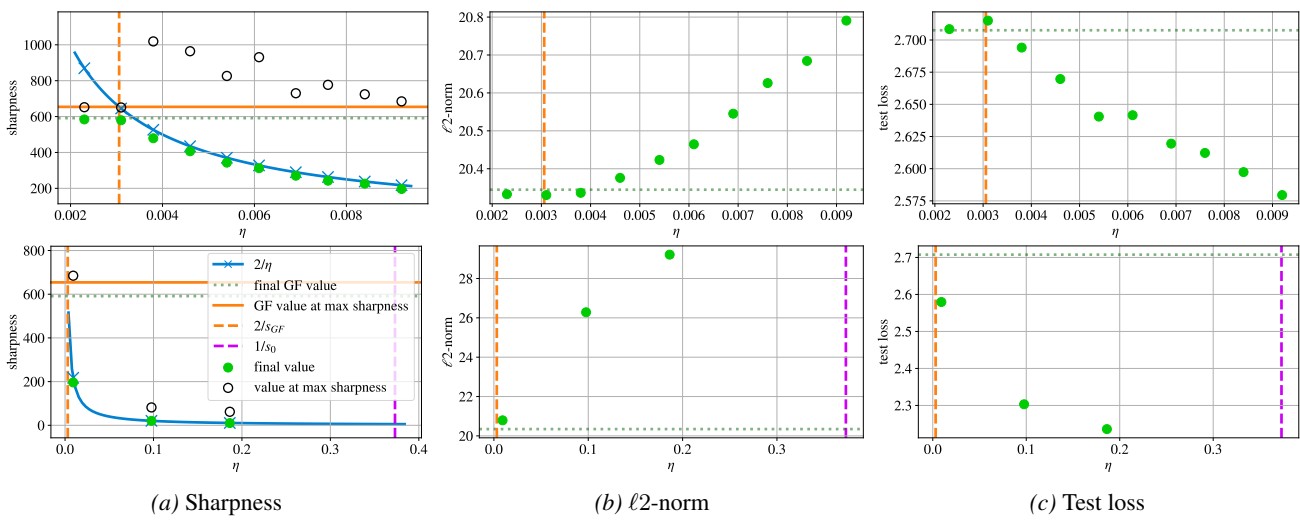

*(a)* Sharpness  *(b)* $\ell$2-norm  *(c)* Test loss

*Figure 28.* **CE loss.** FCN-ReLU, CIFAR-10, train loss 0.1

## J.2. FCNs with tanh Activation

### J.2.1. ON MNIST-5K

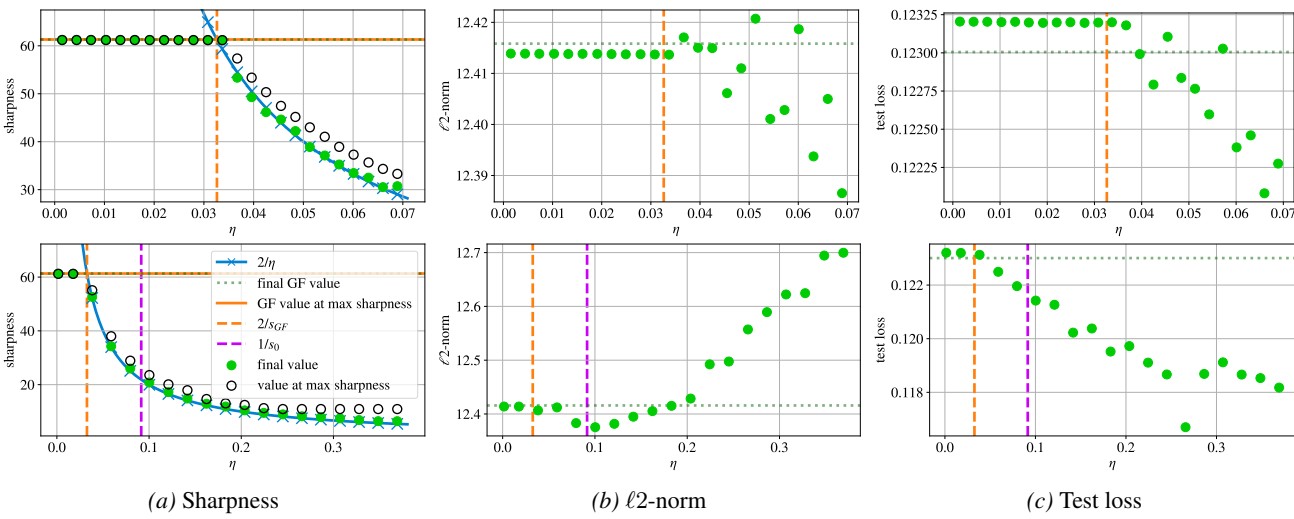

*(a)* Sharpness  *(b)* $\ell$2-norm  *(c)* Test loss

*Figure 29.* **MSE loss.** FCN-tanh, MNIST-5k, train loss 0.1

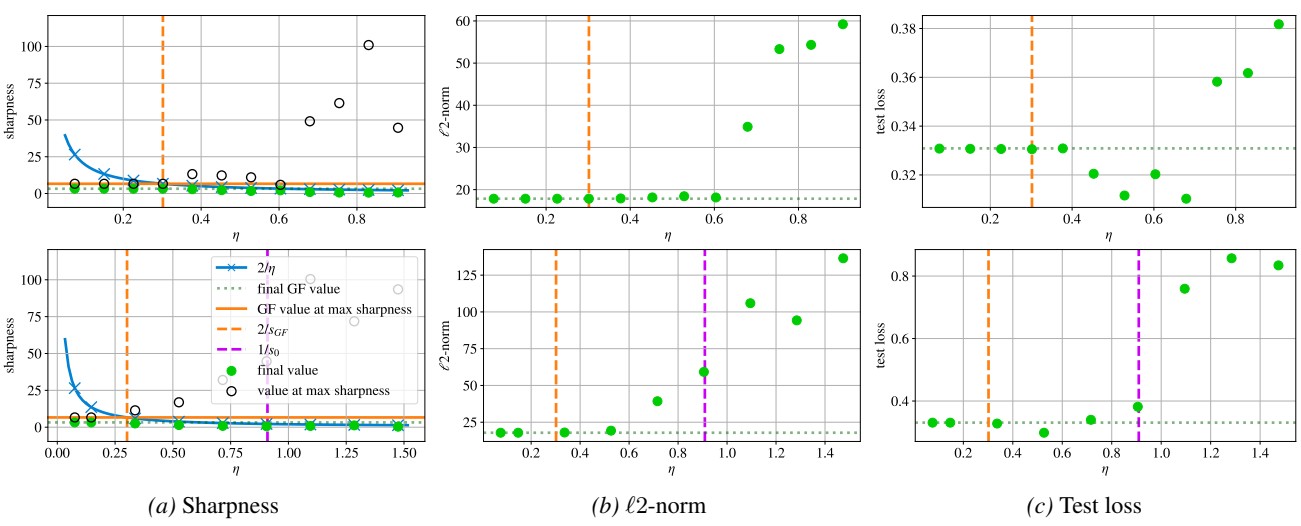

*(a)* Sharpness        *(b)* $\ell 2$-norm        *(c)* Test loss

*Figure 30.* **CE loss.** FCN-tanh, MNIST-5k, train loss 0.01

### J.2.2. ON CIFAR-10-5K

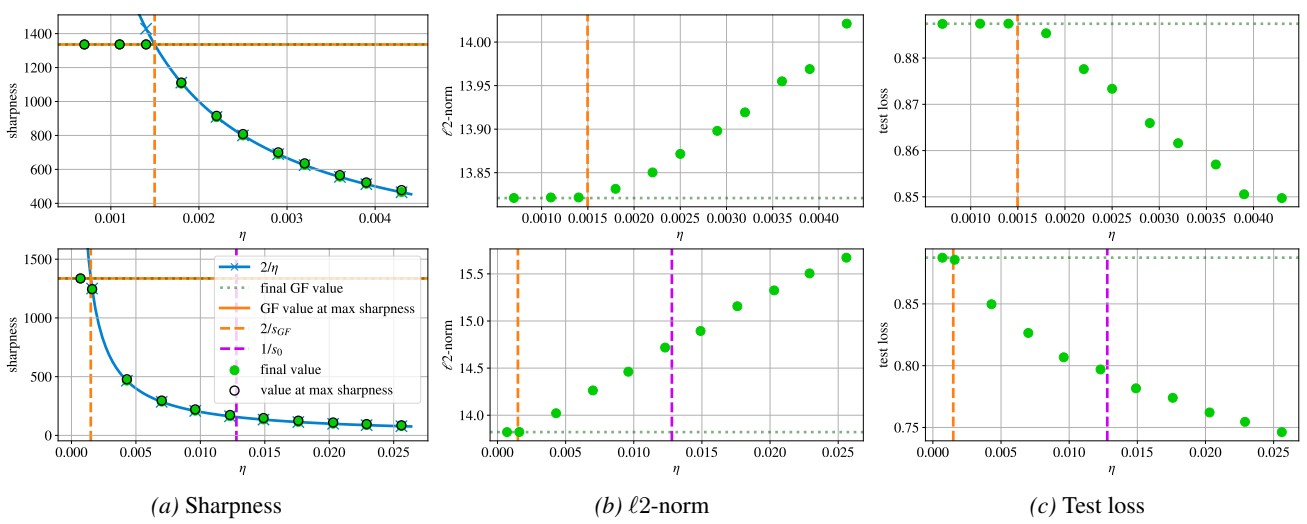

*(a)* Sharpness        *(b)* $\ell 2$-norm        *(c)* Test loss

*Figure 31.* **MSE loss.** FCN-tanh, CIFAR-10-5k, train loss 0.001

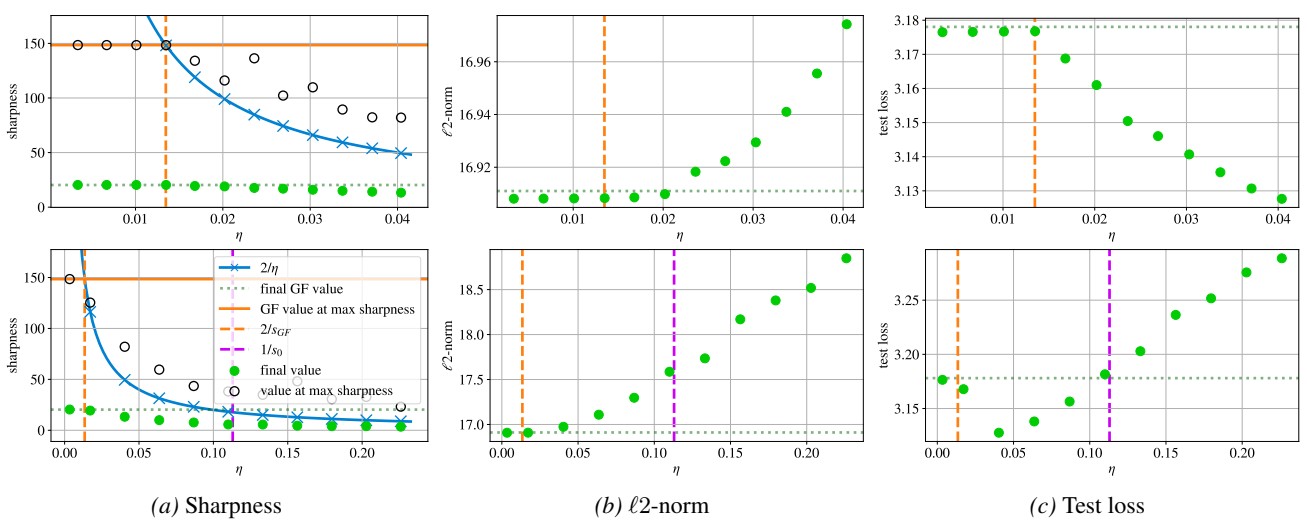

*(a)* Sharpness      *(b)* $\ell$2-norm      *(c)* Test loss

*Figure 32.* **CE loss.** FCN-tanh, CIFAR-10-5k, train loss 0.01

### J.3. CNNs with ReLU Activation

#### J.3.1. ON MNIST-5K

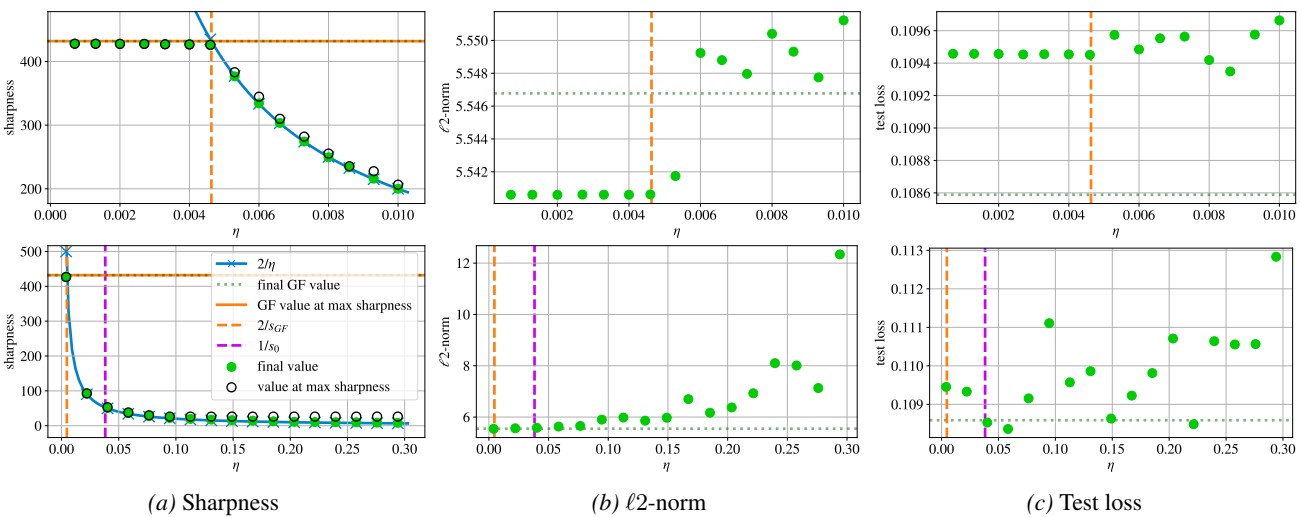

*(a)* Sharpness      *(b)* $\ell$2-norm      *(c)* Test loss

*Figure 33.* **MSE loss.** CNN-ReLU, MNIST-5k, train loss 0.1

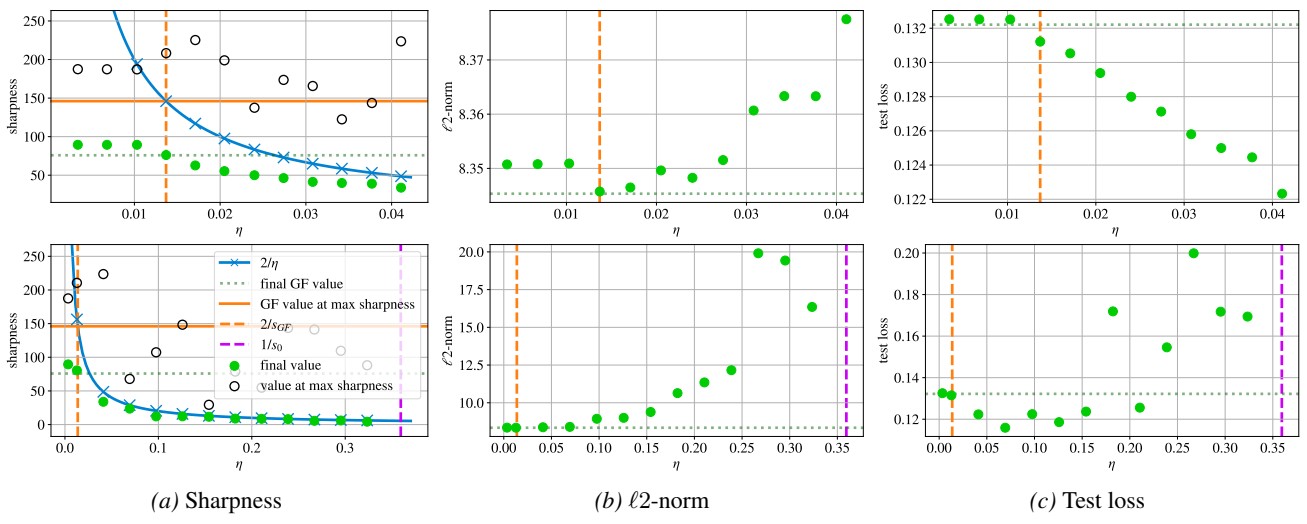

*(a)* Sharpness     *(b)* $\ell 2$-norm     *(c)* Test loss

*Figure 34.* **CE loss.** CNN-ReLU, MNIST-5k, train loss 0.01

### J.3.2. ON FULL MNIST

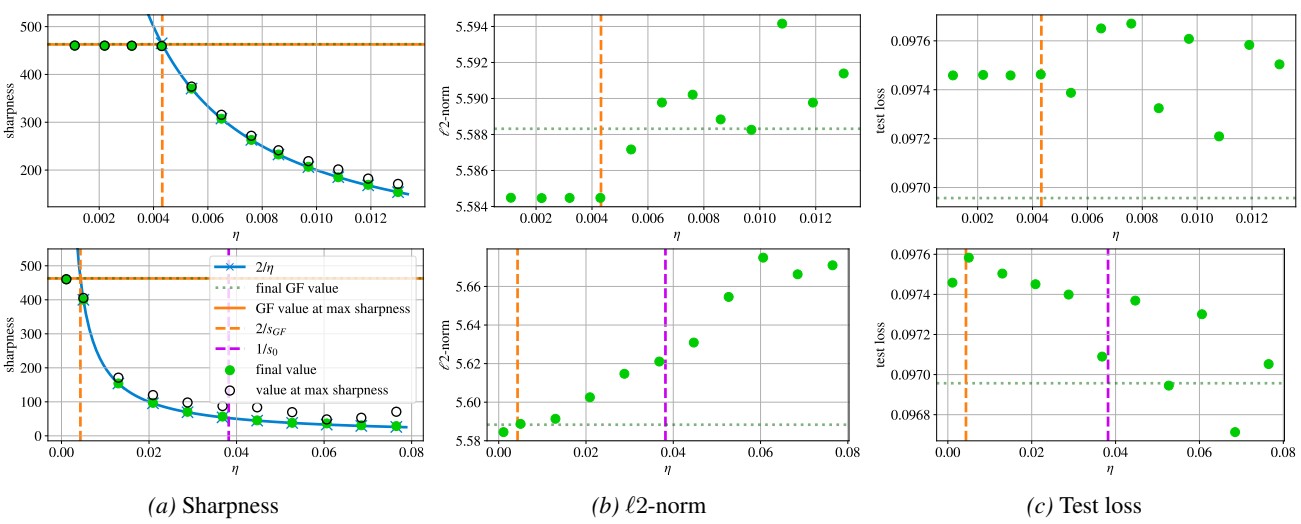

*(a)* Sharpness     *(b)* $\ell 2$-norm     *(c)* Test loss

*Figure 35.* **MSE loss.** CNN-ReLU, MNIST, train loss 0.1

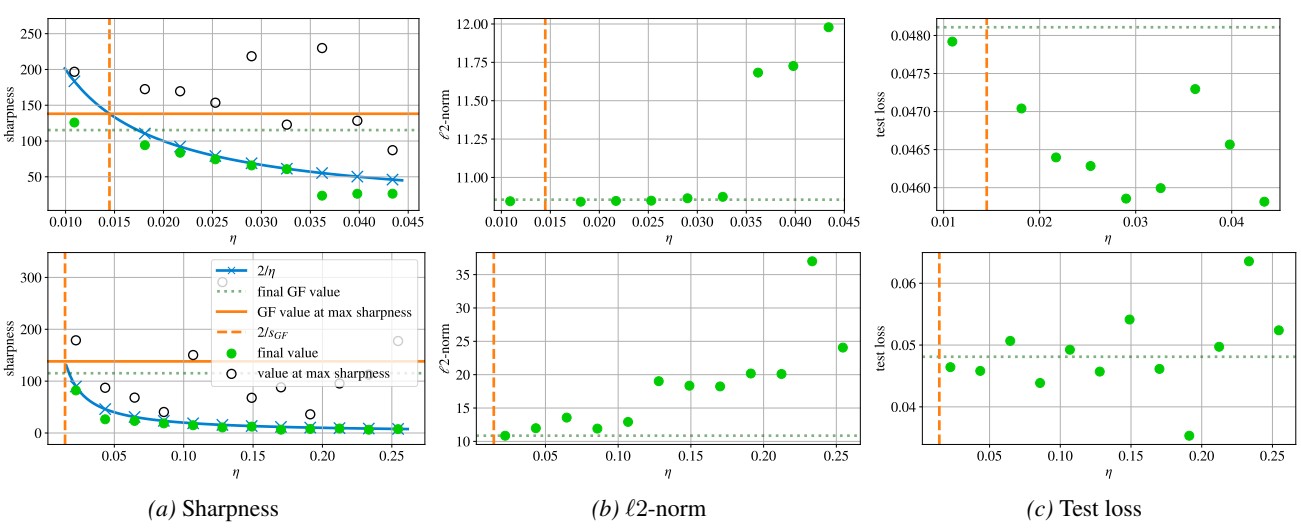

*(a)* Sharpness        *(b)* $\ell$2-norm        *(c)* Test loss

*Figure 36.* **CE loss.** CNN-ReLU, MNIST, train loss 0.01

### J.3.3. ON CIFAR-10-5K

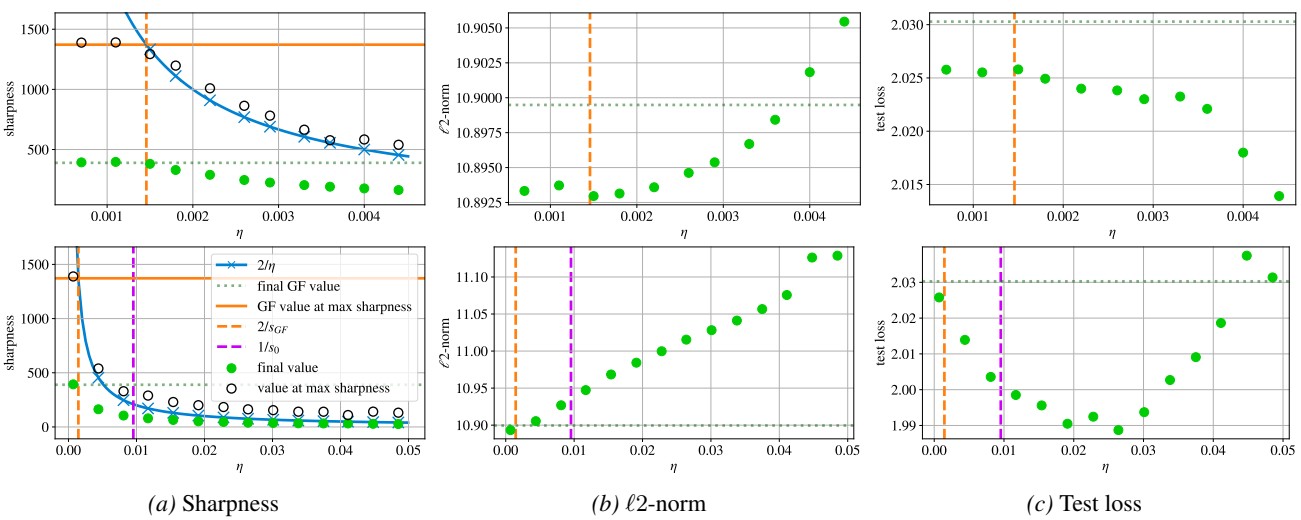

*(a)* Sharpness        *(b)* $\ell$2-norm        *(c)* Test loss

*Figure 37.* **CE loss.** CNN-ReLU with Batch Normalization, CIFAR-10-5k, train loss 0.01

## J.4. Vision Transformer

### J.4.1. ON MNIST-5K

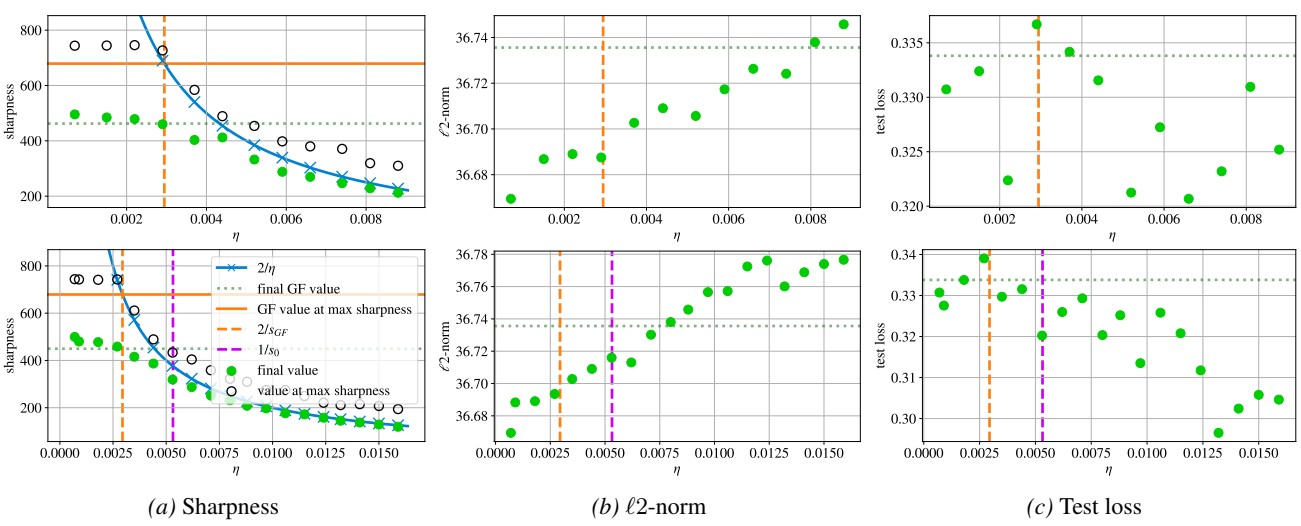

*(a)* Sharpness             *(b)* $\ell2$-norm             *(c)* Test loss

*Figure 38.* **CE loss.** ViT, MNIST-5k, train loss 0.1

### J.4.2. ON CIFAR-10-5K

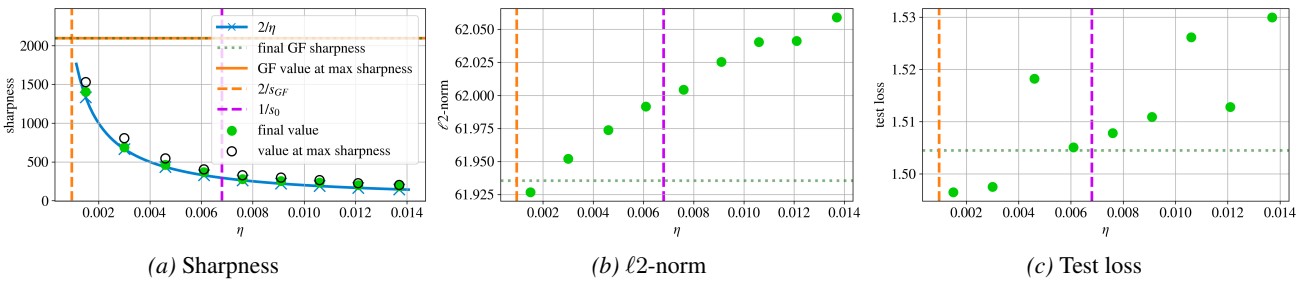

*(a)* Sharpness             *(b)* $\ell2$-norm             *(c)* Test loss

*Figure 39.* **CE loss.** ViT, CIFAR-10-5k, train loss 1

## J.5. ResNet20

### J.5.1. ON CIFAR-10-5K

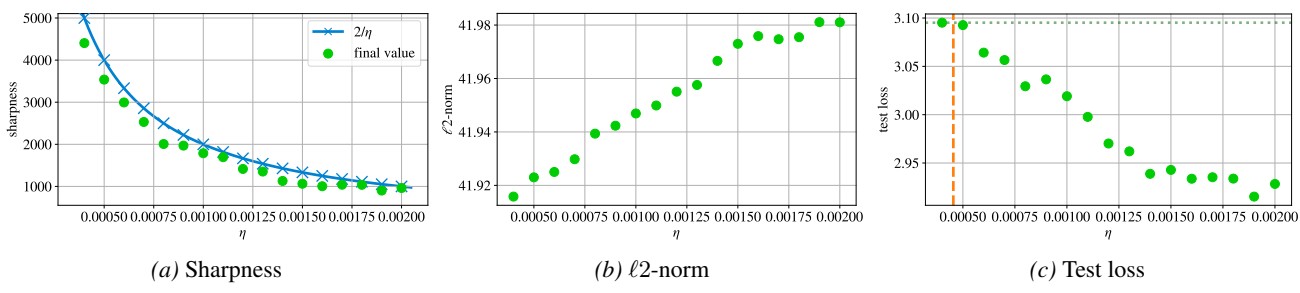

*(a)* Sharpness             *(b)* $\ell2$-norm             *(c)* Test loss

*Figure 40.* **CE loss.** ResNet20, CIFAR-10-5k, train loss 0.1

## J.6. Varying Width and Depth

### J.6.1. ON MNIST-5K

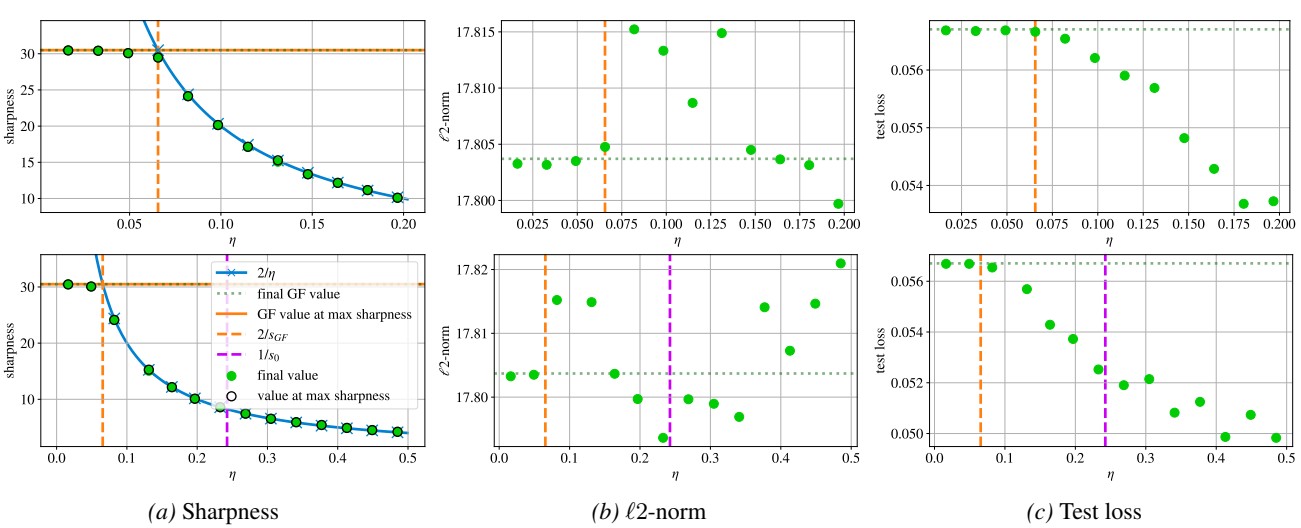

*(a)* Sharpness      *(b)* $\ell 2$-norm      *(c)* Test loss

*Figure 41.* **FCN-ReLU, $2\times$ width ($400 \times 2$).** Train loss 0.01, MNIST-5k, MSE loss

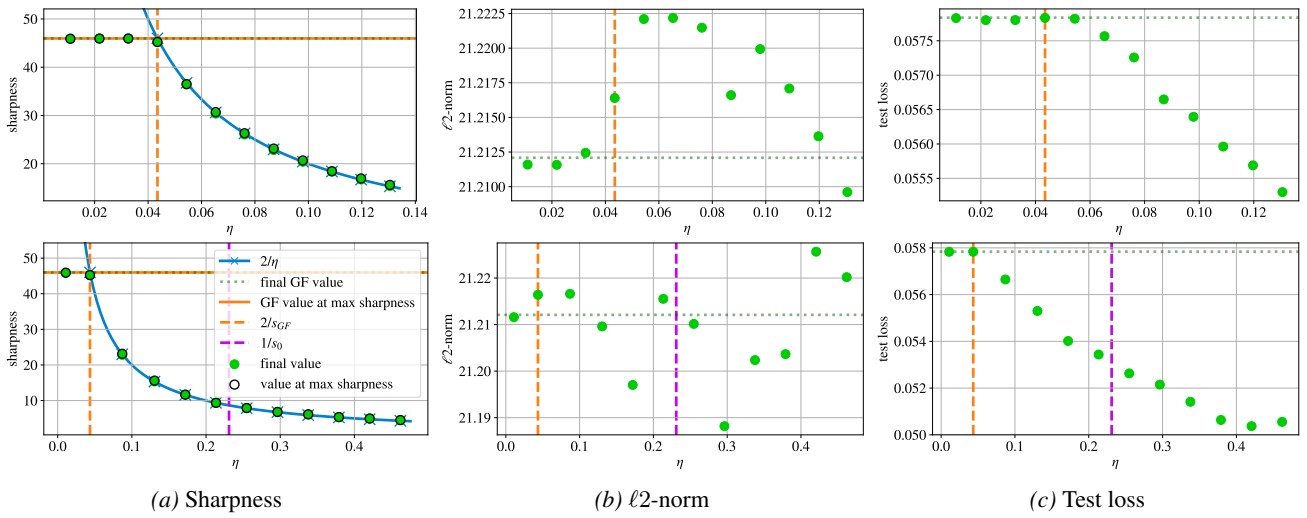

*(a)* Sharpness      *(b)* $\ell 2$-norm      *(c)* Test loss

*Figure 42.* **FCN-ReLU, $3\times$ width ($600 \times 2$).** Train loss 0.01, MNIST-5k, MSE loss

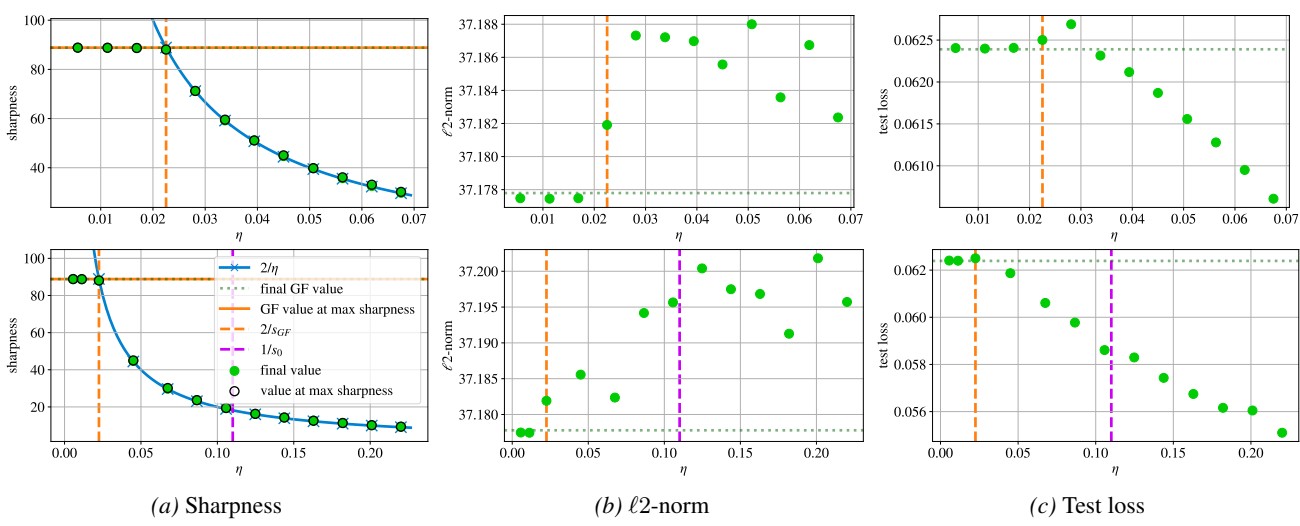

*(a)* Sharpness                                *(b)* $\ell 2$-norm                            *(c)* Test loss

*Figure 43.* **FCN-ReLU,** $10\times$ **width** ($2000 \times 2$)**.** Train loss 0.01, MNIST-5k, MSE loss

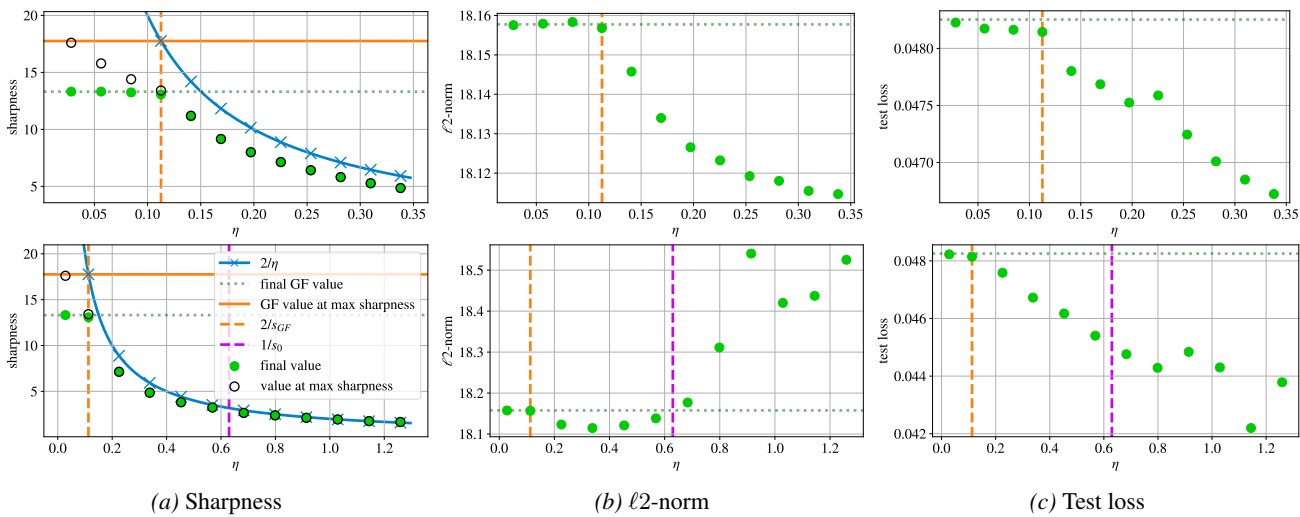

*(a)* Sharpness                                *(b)* $\ell 2$-norm                            *(c)* Test loss

*Figure 44.* **FCN-ReLU,** $2\times$ **depth** ($200 \times 4$)**.** Train loss 0.01, MNIST-5k, MSE loss

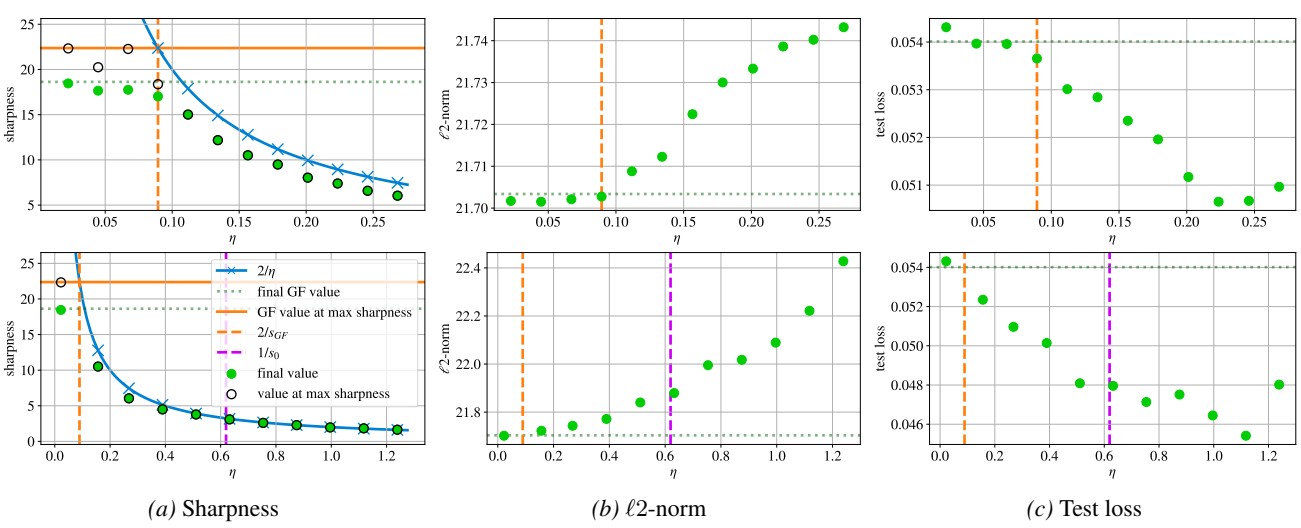

*(a)* Sharpness

*(b)* $\ell 2$-norm

*(c)* Test loss

*Figure 45.* **FCN-ReLU,** $3\times$ **depth** $(200 \times 6)$**.** Train loss 0.01, MNIST-5k, MSE loss

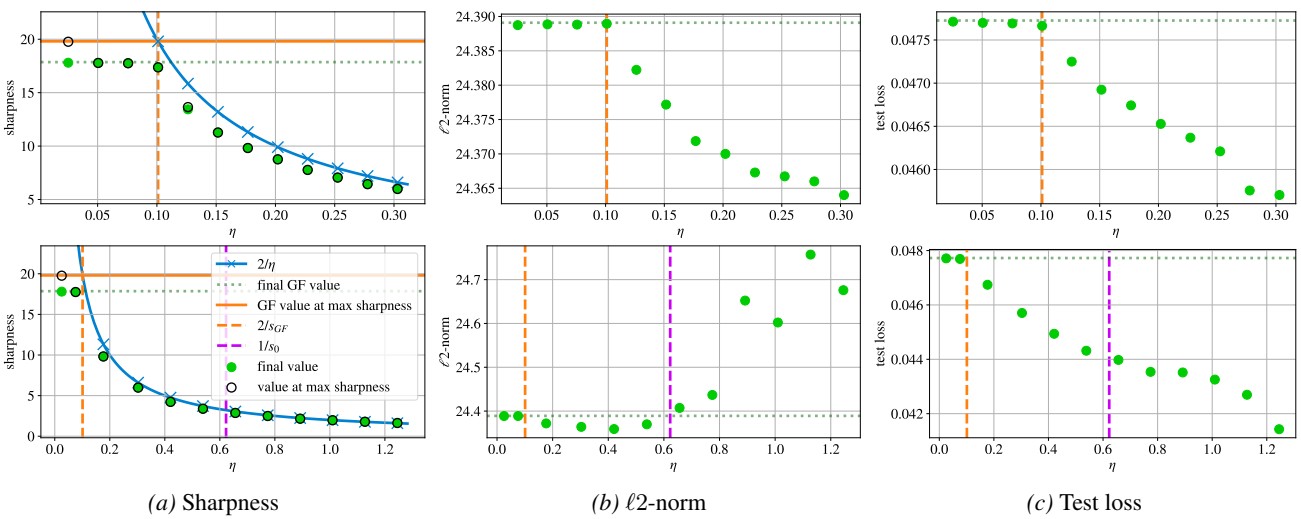

*(a)* Sharpness

*(b)* $\ell 2$-norm

*(c)* Test loss

*Figure 46.* **FCN-ReLU,** $2\times$ **width and depth** $(400 \times 4)$**.** Train loss 0.01, MNIST-5k, MSE loss

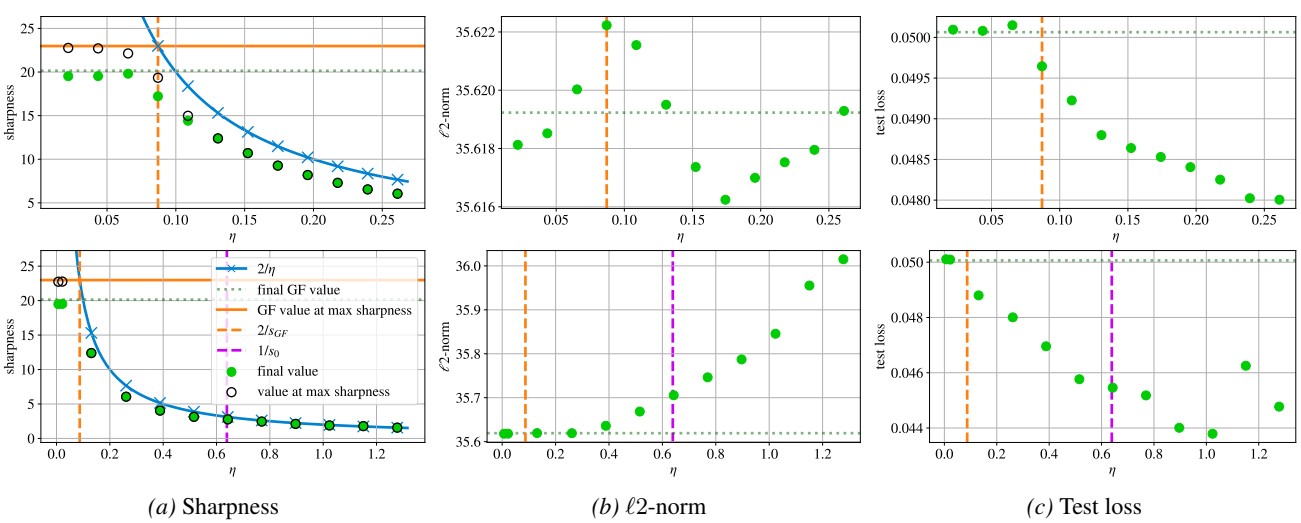

*(a)* Sharpness  *(b)* $\ell 2$-norm  *(c)* Test loss

*Figure 47.* **FCN-ReLU,** $3\times$ **width and depth** $(600 \times 6)$**.** Train loss 0.01, MNIST-5k, MSE loss

### J.6.2. ON CIFAR-10-5K

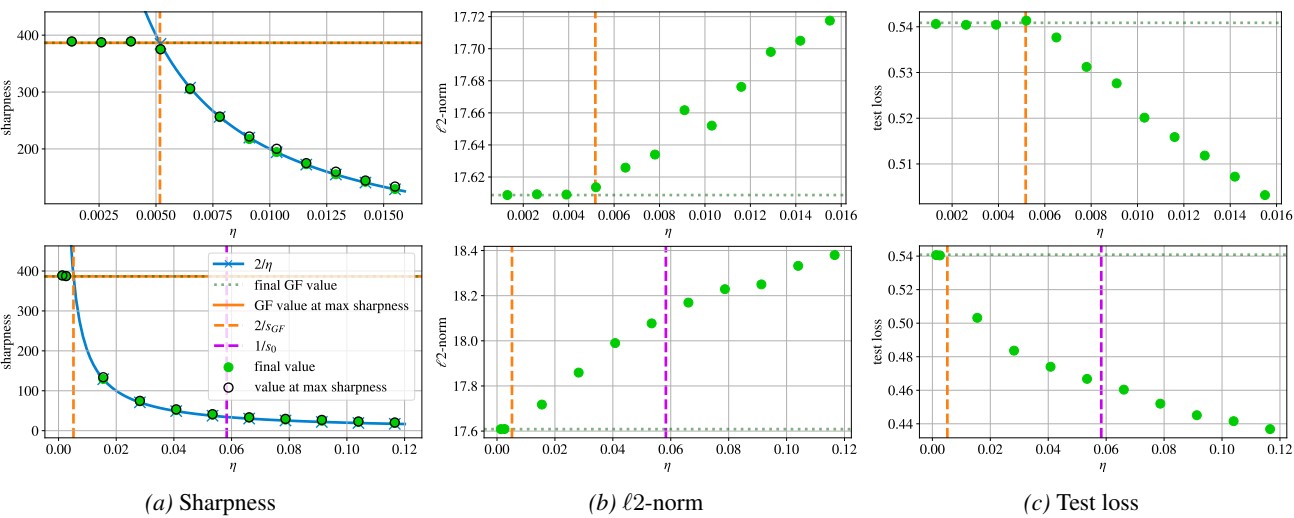

*(a)* Sharpness  *(b)* $\ell 2$-norm  *(c)* Test loss

*Figure 48.* **FCN-ReLU,** $2\times$ **width** $(400 \times 2)$**.** Train loss 0.01, CIFAR-10-5k, MSE loss

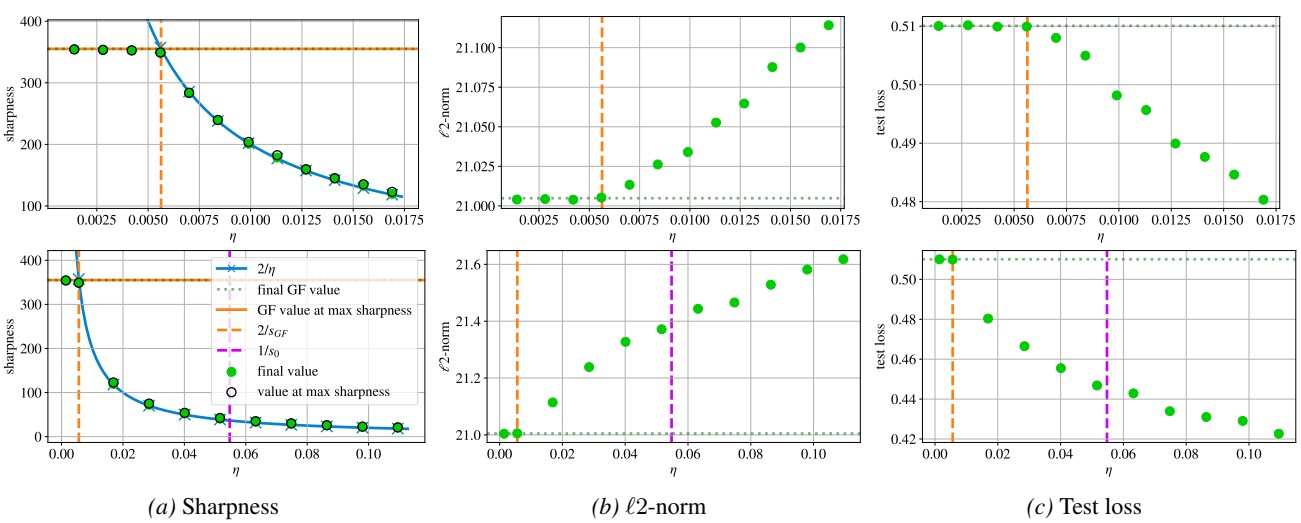

*(a)* Sharpness  *(b)* $\ell 2$-norm  *(c)* Test loss

*Figure 49.* **FCN-ReLU,** $3\times$ **width** $(600 \times 2)$**.** Train loss 0.01, CIFAR-10-5k, MSE loss

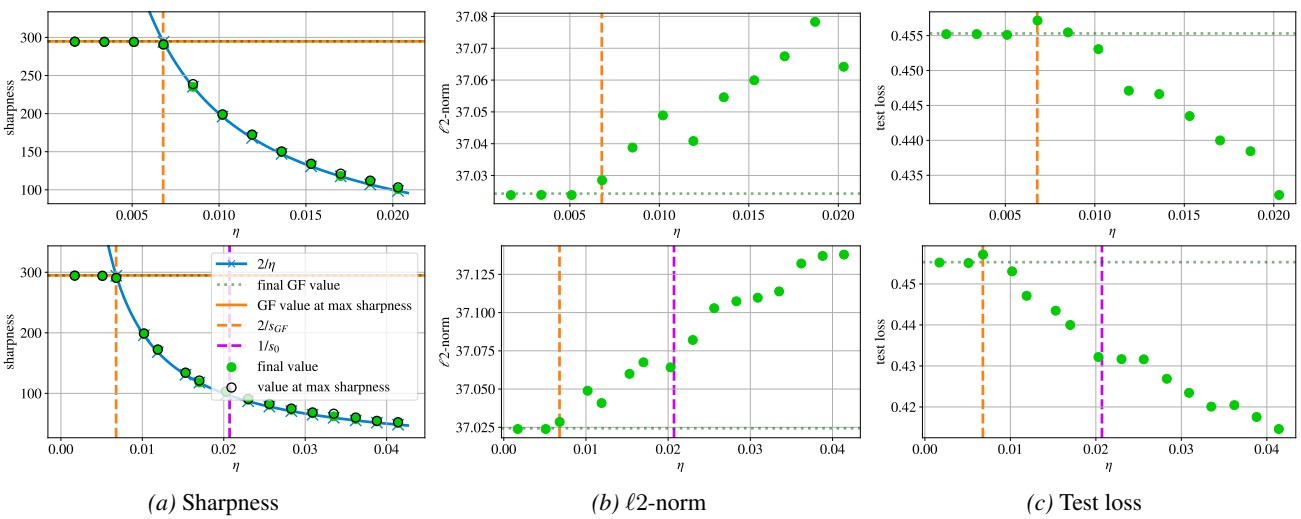

*(a)* Sharpness  *(b)* $\ell 2$-norm  *(c)* Test loss

*Figure 50.* **FCN-ReLU,** $10\times$ **width** $(2000 \times 2)$**.** Train loss 0.01, CIFAR-10-5k, MSE loss

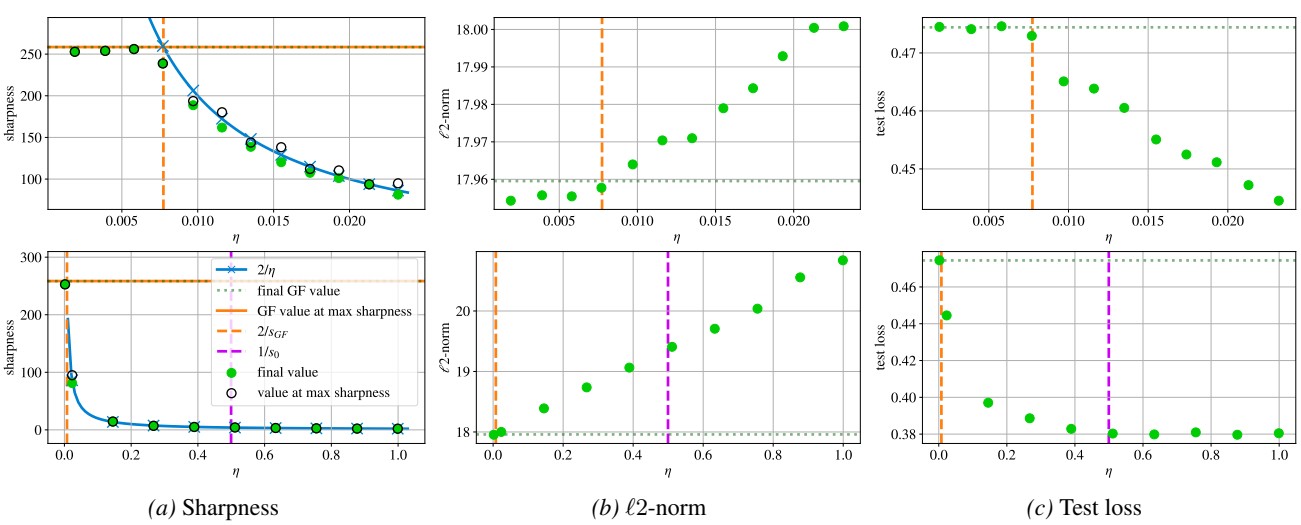

*(a)* Sharpness          *(b)* $\ell2$-norm          *(c)* Test loss

*Figure 51.* **FCN-ReLU,** $2\times$ **depth** ($200 \times 4$)**.** Train loss 0.01, CIFAR-10-5k, MSE loss

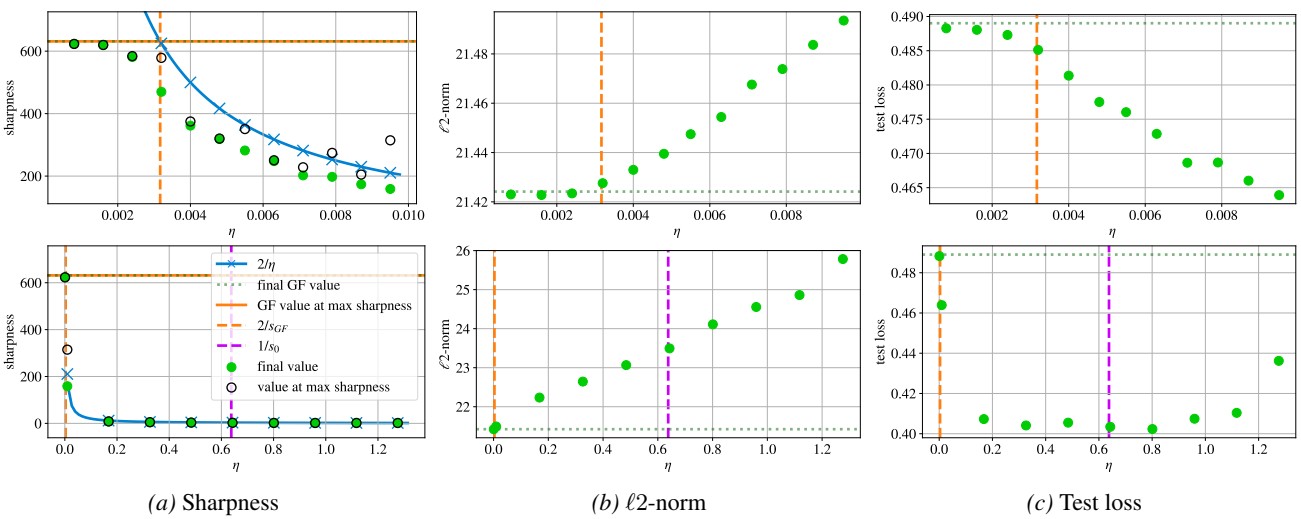

*(a)* Sharpness          *(b)* $\ell2$-norm          *(c)* Test loss

*Figure 52.* **FCN-ReLU,** $3\times$ **depth** ($200 \times 6$)**.** Train loss 0.01, CIFAR-10-5k, MSE loss

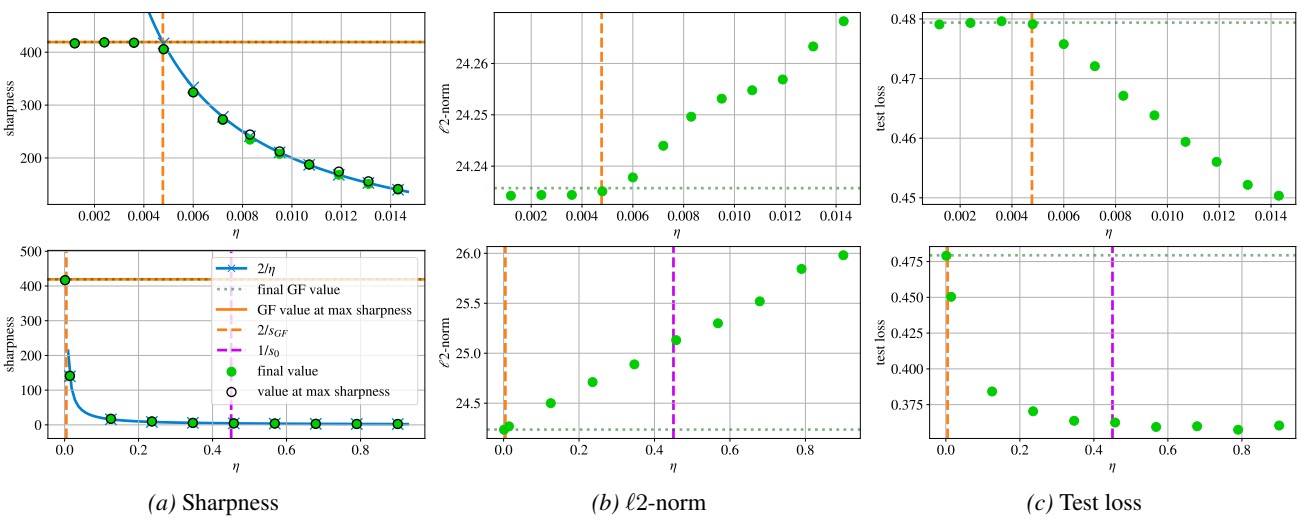

*(a)* Sharpness       *(b)* $\ell$2-norm       *(c)* Test loss

*Figure 53.* **FCN-ReLU,** $2\times$ **width and depth** $(400 \times 4)$**.** Train loss 0.01, CIFAR-10-5k, MSE loss

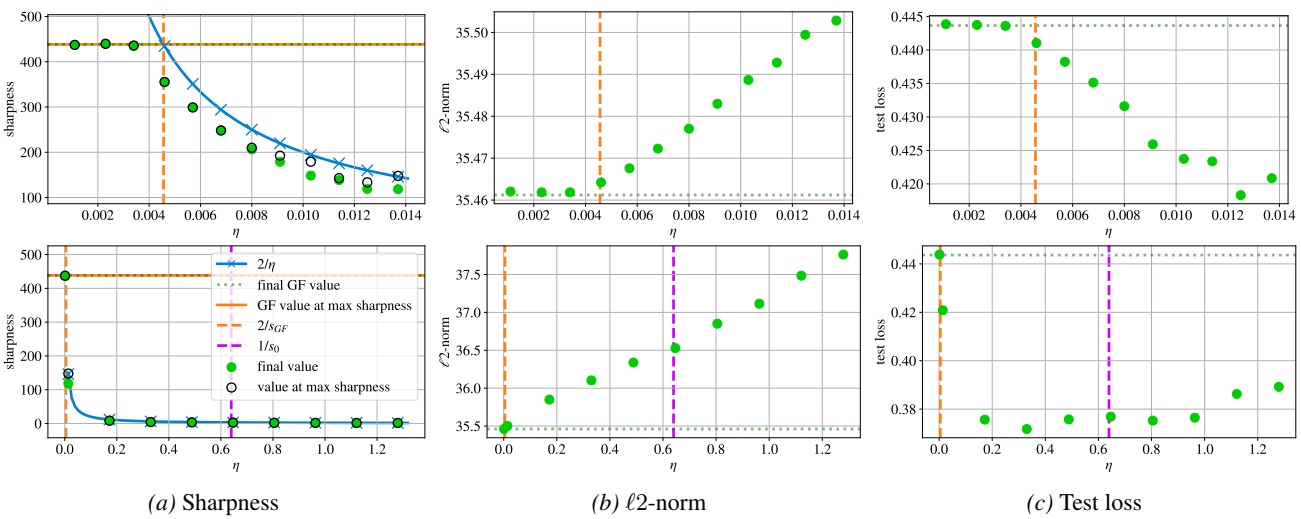

*(a)* Sharpness       *(b)* $\ell$2-norm       *(c)* Test loss

*Figure 54.* **FCN-ReLU,** $3\times$ **width and depth** $(600 \times 6)$**.** Train loss 0.01, CIFAR-10-5k, MSE loss

## J.7. Further Configurations

### J.7.1. DIFFERENT LOSS GOALS

FCN-ReLU ON MNIST-5k WITH THE MSE LOSS

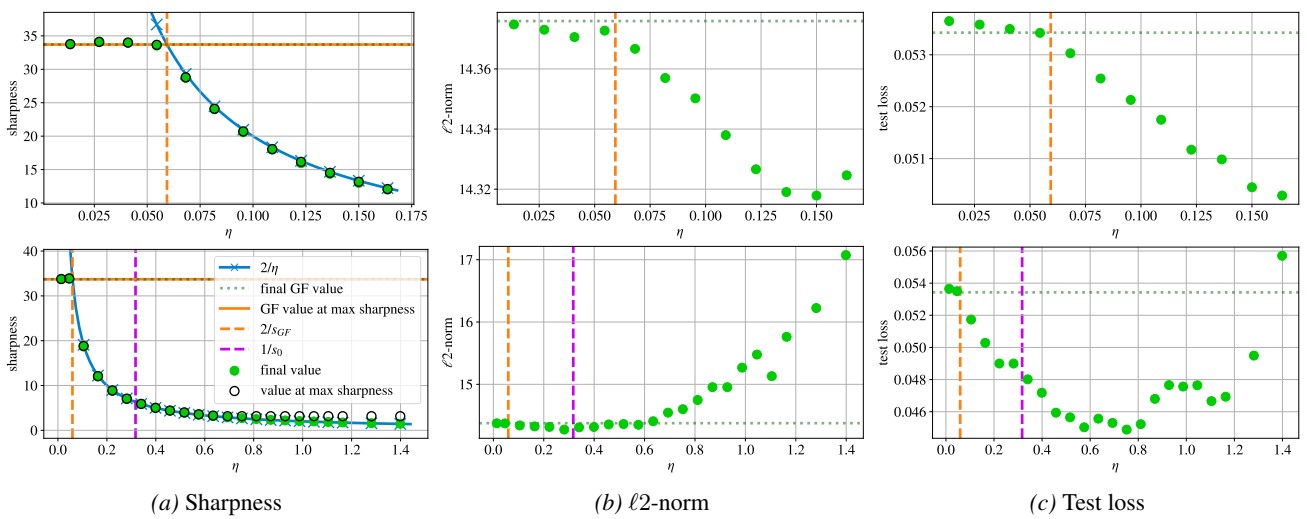

*(a)* Sharpness          *(b)* $\ell2$-norm          *(c)* Test loss

*Figure 55.* **Train loss** $0.001$. FCN-ReLU, MNIST-5k, MSE loss

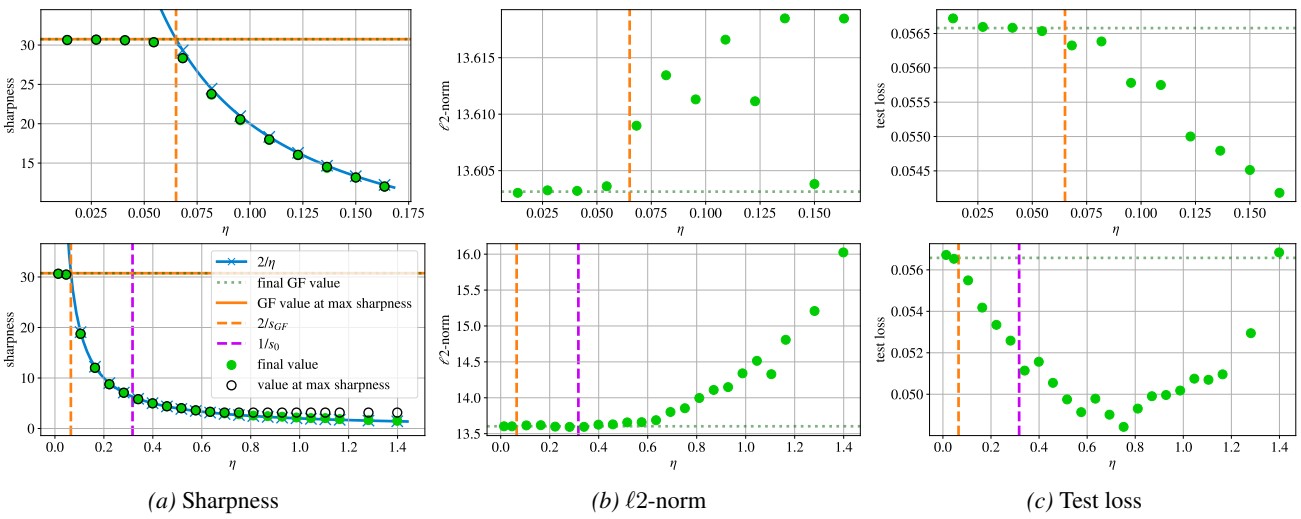

*(a)* Sharpness          *(b)* $\ell2$-norm          *(c)* Test loss

*Figure 56.* **Train loss** $0.01$. FCN-ReLU, MNIST-5k, MSE loss

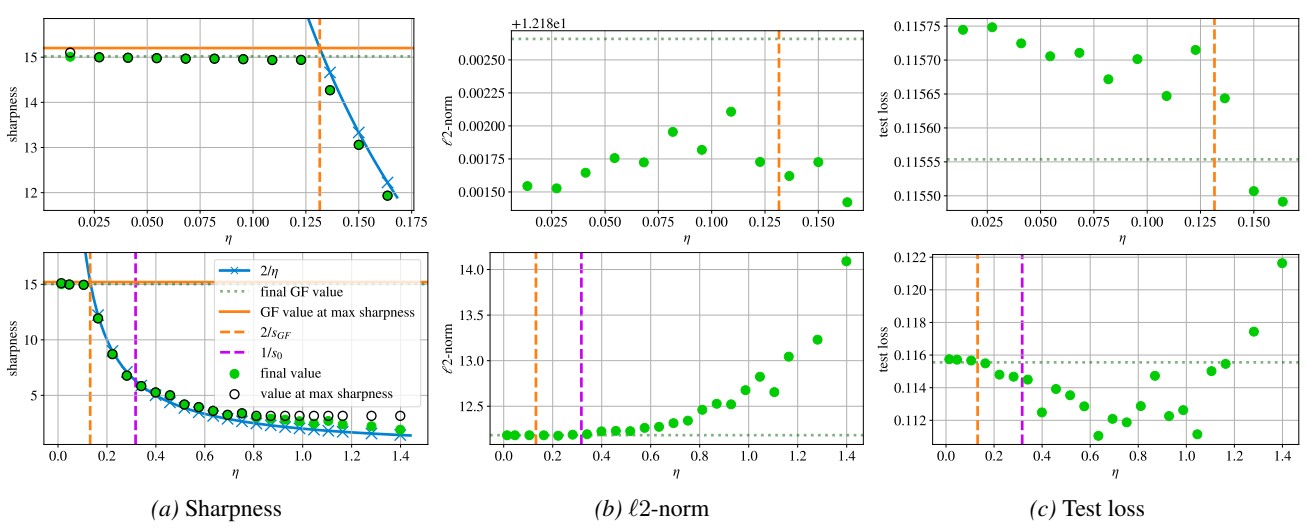

*(a)* Sharpness      *(b)* $\ell2$-norm      *(c)* Test loss

*Figure 57.* **Train loss** 0.1. FCN-ReLU, MNIST-5k, MSE loss

FCN-RELU ON MNIST-5K WITH THE CE LOSS

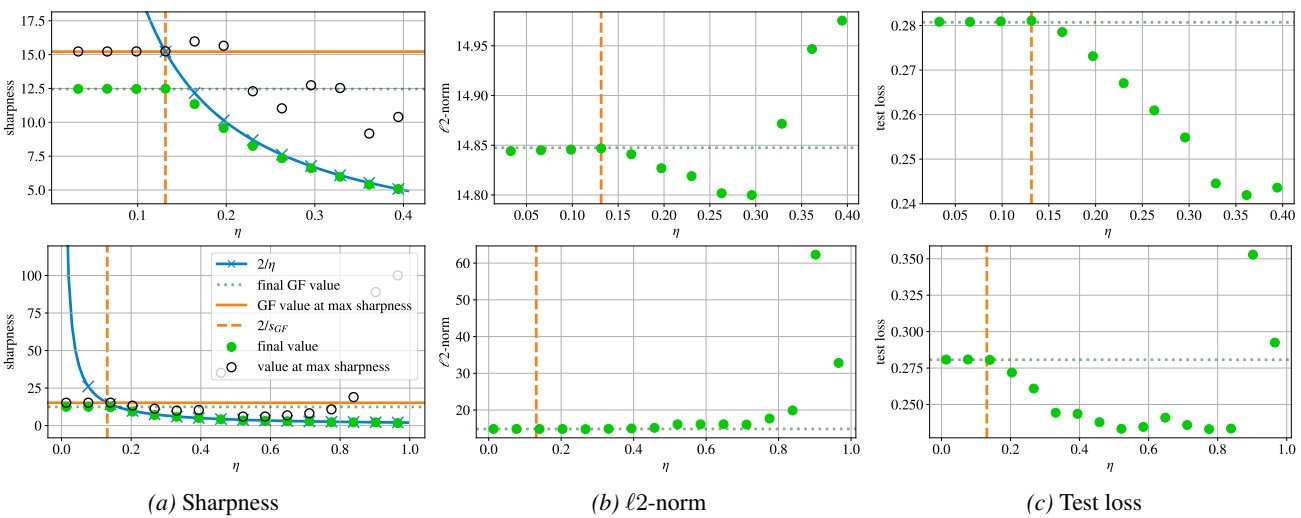

*(a)* Sharpness      *(b)* $\ell2$-norm      *(c)* Test loss

*Figure 58.* **Train loss** 0.1. FCN-ReLU, MNIST-5k, CE loss

FCN-RᴇLU ᴏɴ CIFAR-10-5ᴋ ᴡɪᴛʜ ᴛʜᴇ MSE Lᴏss

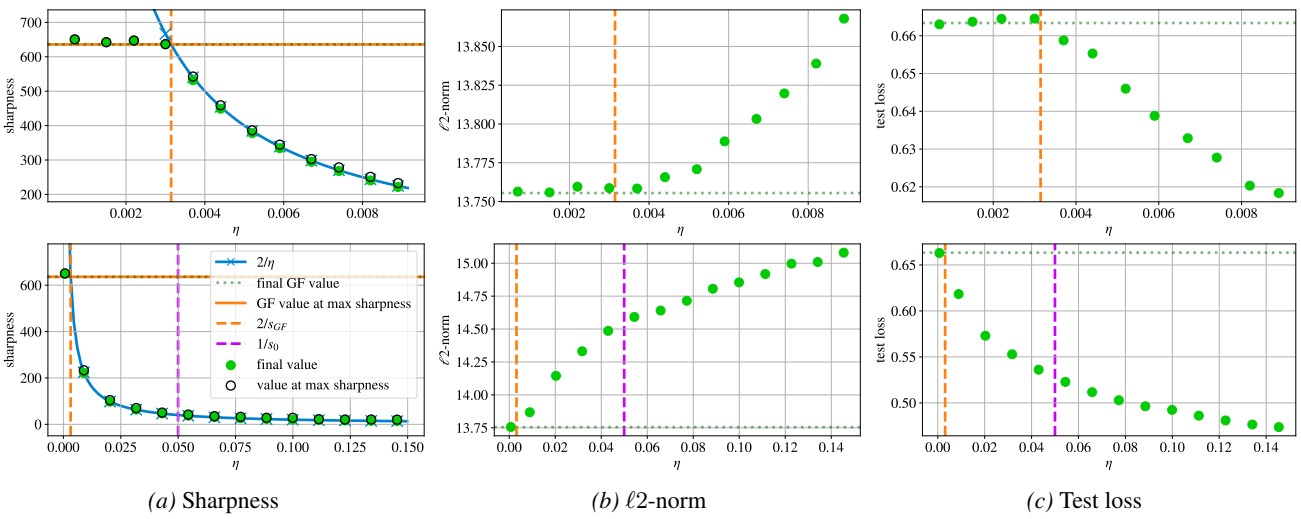

*(a)* Sharpness      *(b)* $\ell2$-norm      *(c)* Test loss

*Figure 59.* **Train loss** 0.001. FCN-ReLU, CIFAR-10-5k, MSE loss

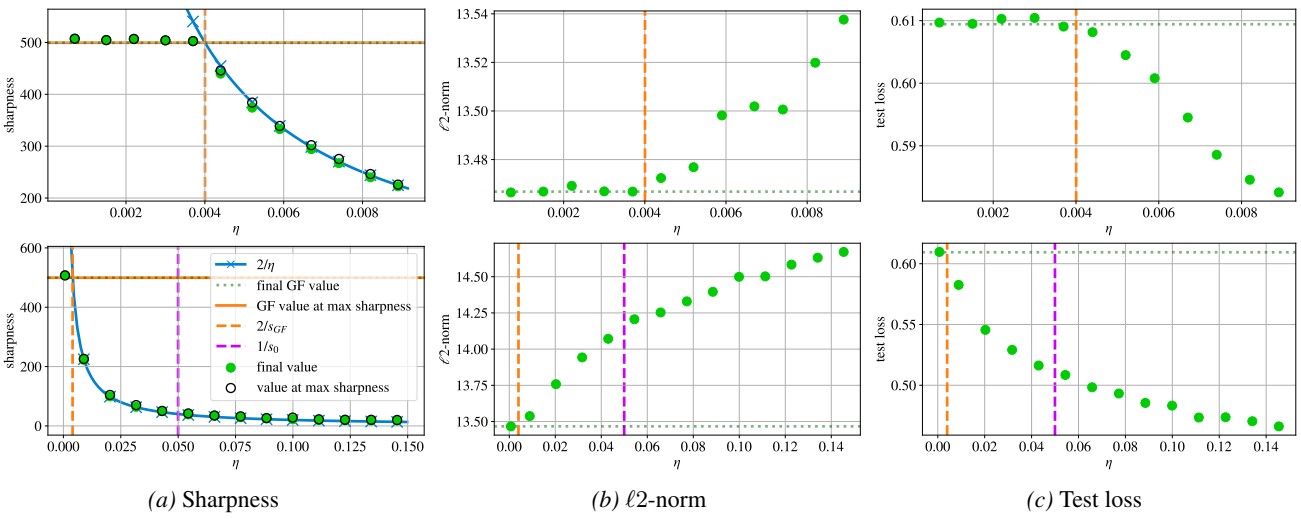

*(a)* Sharpness      *(b)* $\ell2$-norm      *(c)* Test loss

*Figure 60.* **Train loss** 0.01. FCN-ReLU, CIFAR-10-5k, MSE loss

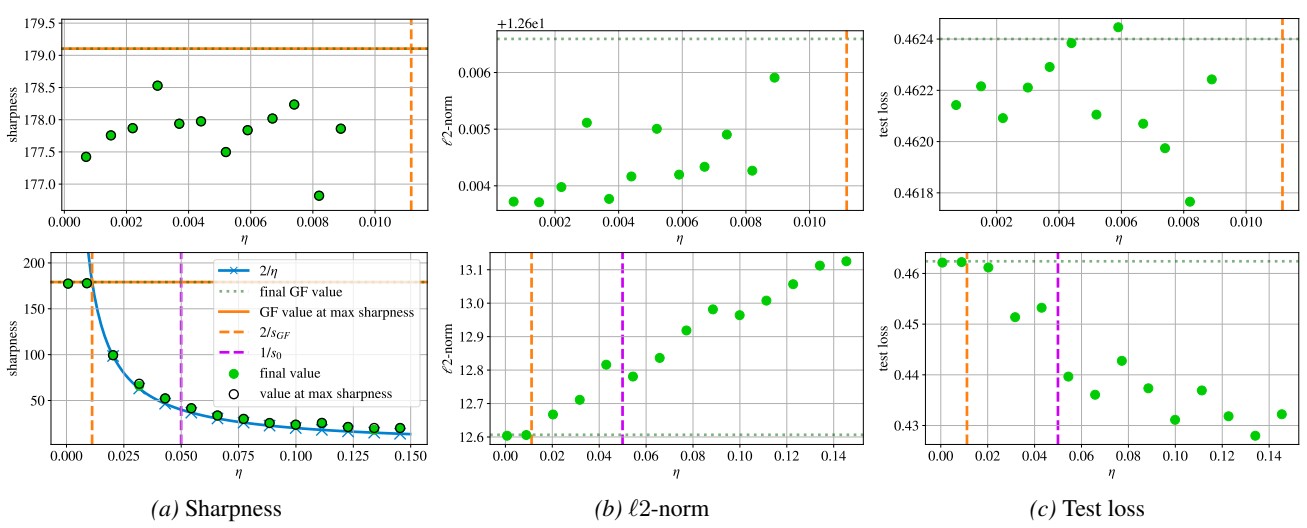

*Figure 61.* **Train loss** 0.1. FCN-ReLU, CIFAR-10-5k, MSE loss

FCN-RELU ON CIFAR-10-5K WITH THE CE LOSS

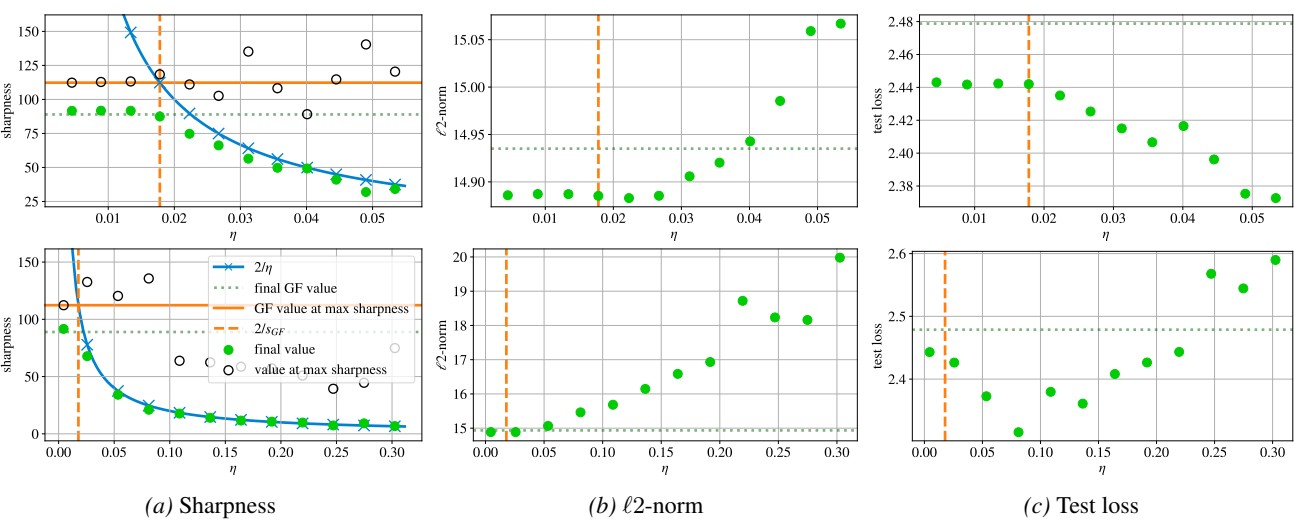

*Figure 62.* **Train loss** 0.1. FCN-ReLU, CIFAR-10-5k, CE loss

### J.7.2. OTHER INITIALIZATION SEEDS FOR FCN-RELU ON CIFAR-10-5K WITH THE MSE LOSS

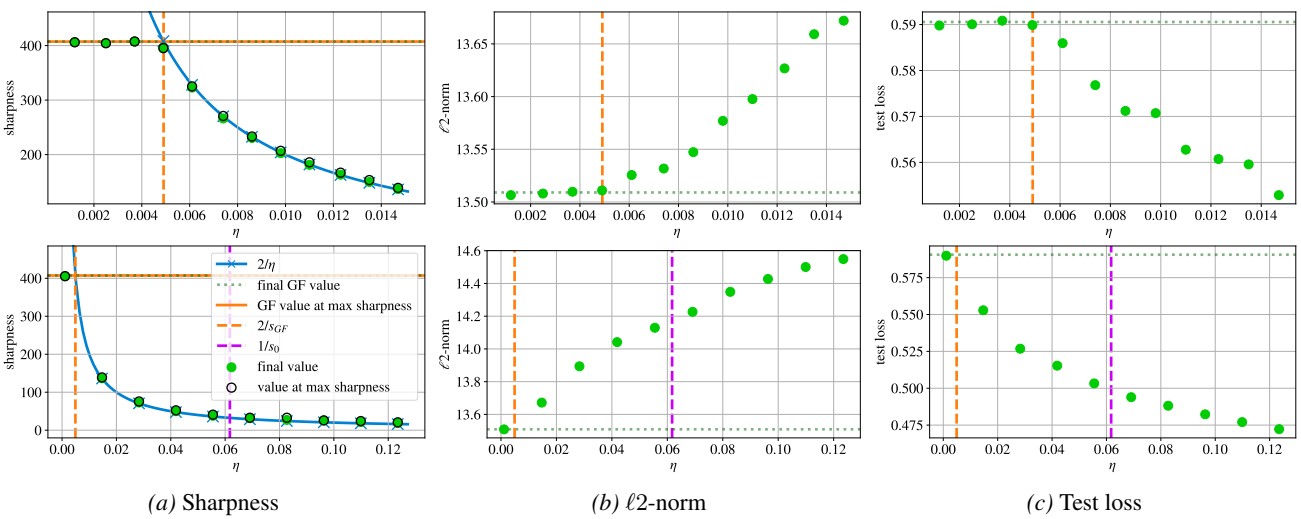

*(a)* Sharpness          *(b)* $\ell$2-norm          *(c)* Test loss

*Figure 63.* **Seed 44.** FCN-ReLU, CIFAR-10-5k, MSE loss, train loss 0.01

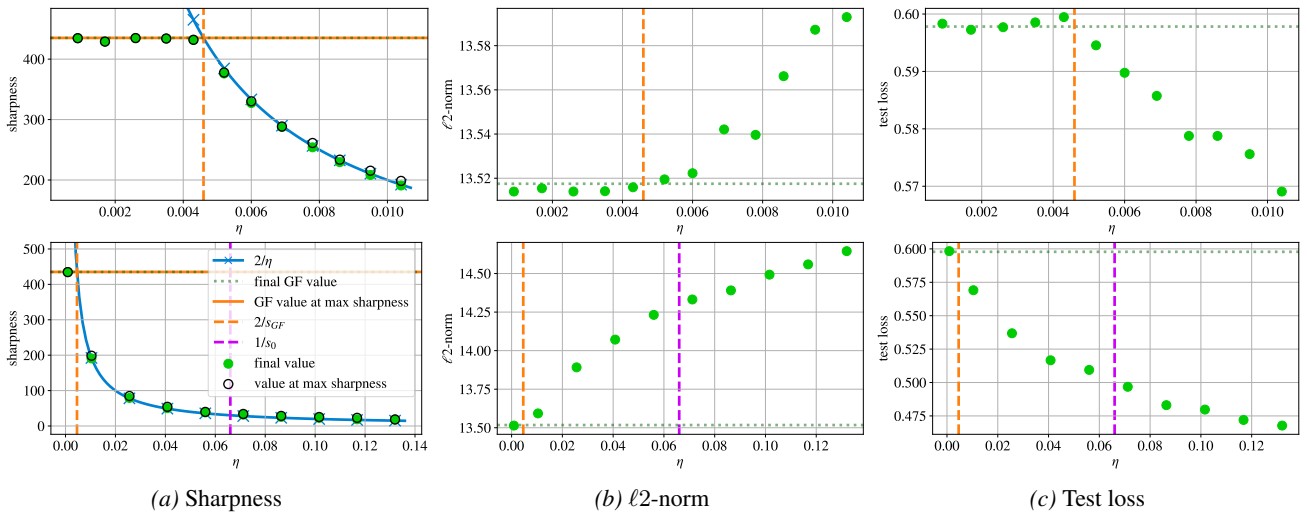

*(a)* Sharpness          *(b)* $\ell$2-norm          *(c)* Test loss

*Figure 64.* **Seed 45.** FCN-ReLU, CIFAR-10-5k, MSE loss, train loss 0.01

### J.7.3. SCALED INITIALIZATION FOR FCN-RELU ON CIFAR-10-5K WITH THE MSE LOSS

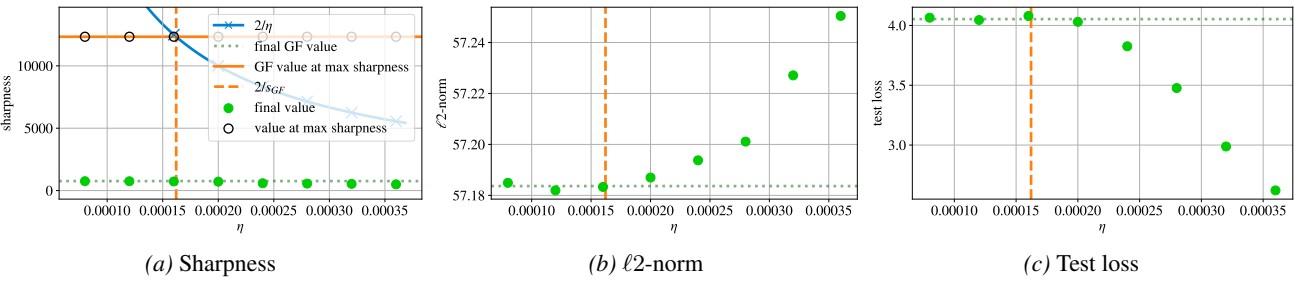

*(a)* Sharpness          *(b)* $\ell$2-norm          *(c)* Test loss

*Figure 65.* **Initialization from seed** 43 **scaled** $\times 5$. FCN-ReLU, CIFAR-10-5k, MSE loss, train loss 0.1

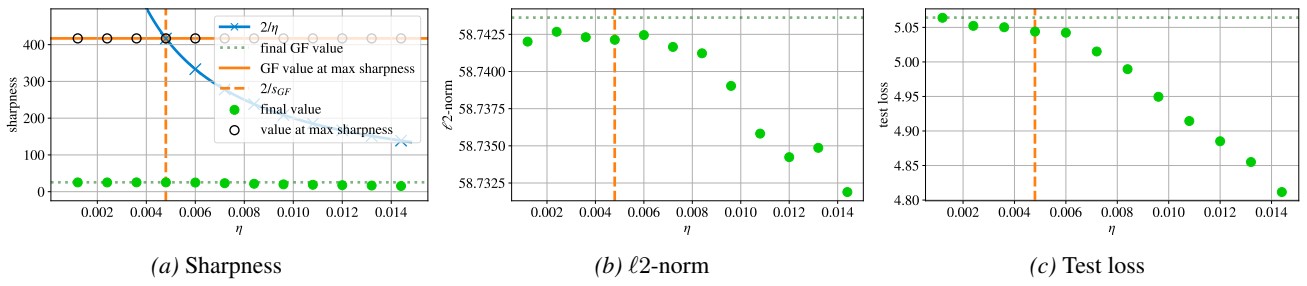

*(a)* Sharpness                  *(b)* $\ell$2-norm                  *(c)* Test loss

*Figure 66.* **Initialization from seed** $43$ **scaled** $\times 5$**.** FCN-ReLU, CIFAR-10-5k, CE loss, train loss 0.01

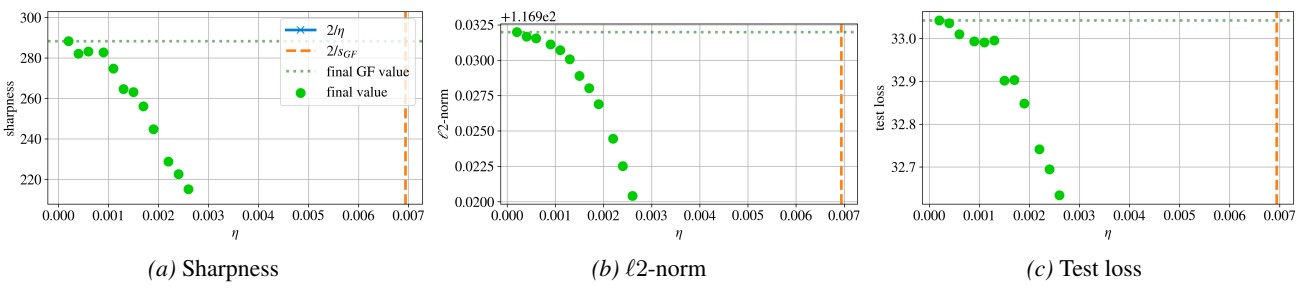

*(a)* Sharpness                  *(b)* $\ell$2-norm                  *(c)* Test loss

*Figure 67.* **Initialization from seed** $43$ **scaled** $\times 10$**.** FCN-ReLU, CIFAR-10-5k, CE loss, train loss 0.01

## J.8. Further Properties

### J.8.1. ALTERNATIVE NORMS AND DISTANCE FROM GF SOLUTION

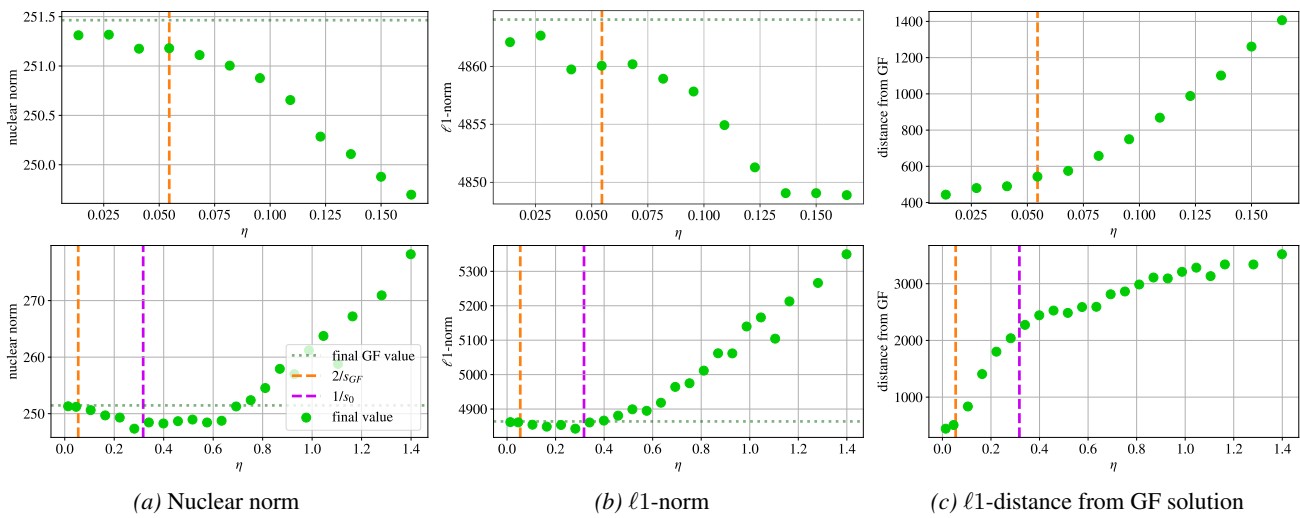

*(a)* Nuclear norm                  *(b)* $\ell$1-norm                  *(c)* $\ell$1-distance from GF solution

*Figure 68.* **FCN-ReLU on MNIST-5k with the MSE loss.** Train loss 0.0001

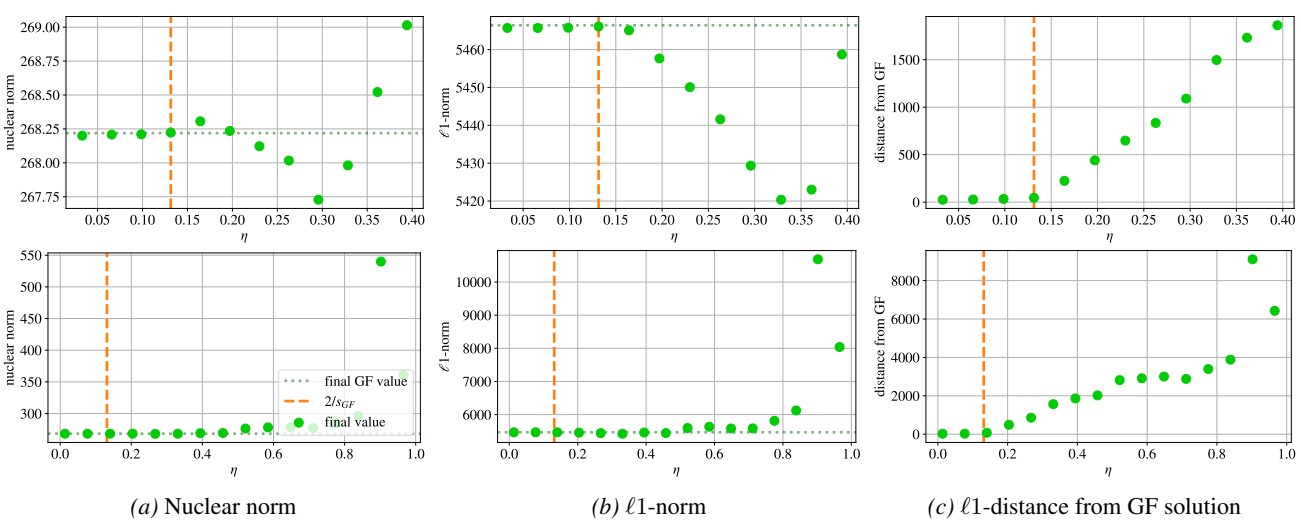

*(a)* Nuclear norm    *(b)* $\ell 1$-norm    *(c)* $\ell 1$-distance from GF solution

*Figure 69.* **FCN-ReLU on MNIST-5k with the CE loss.** Train loss 0.01

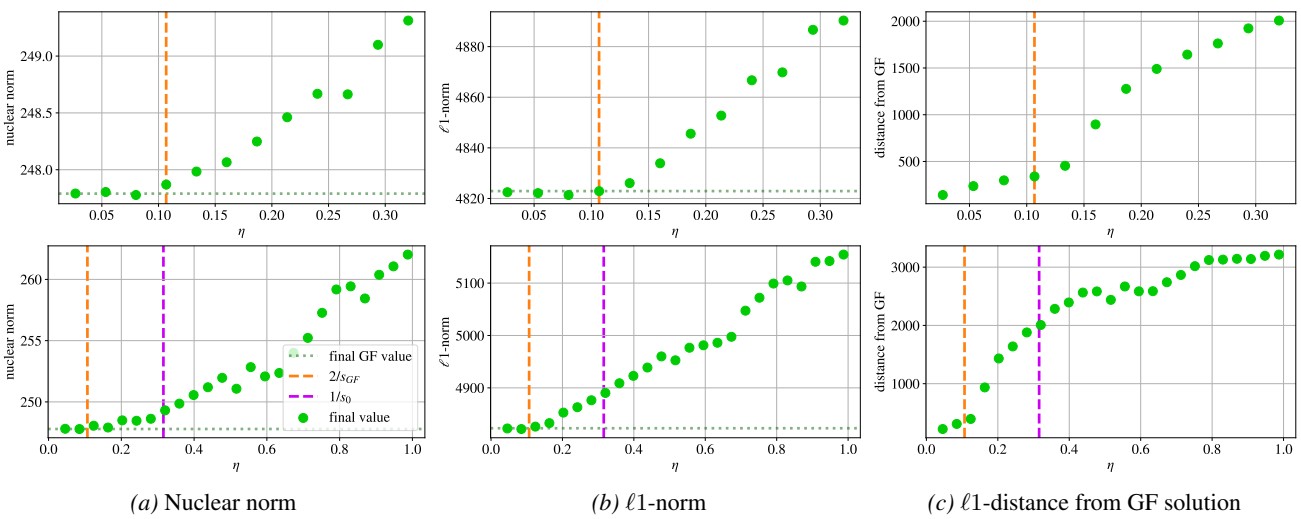

*(a)* Nuclear norm    *(b)* $\ell 1$-norm    *(c)* $\ell 1$-distance from GF solution

*Figure 70.* **FCN-ReLU on full MNIST with the MSE loss.** Train loss 0.01

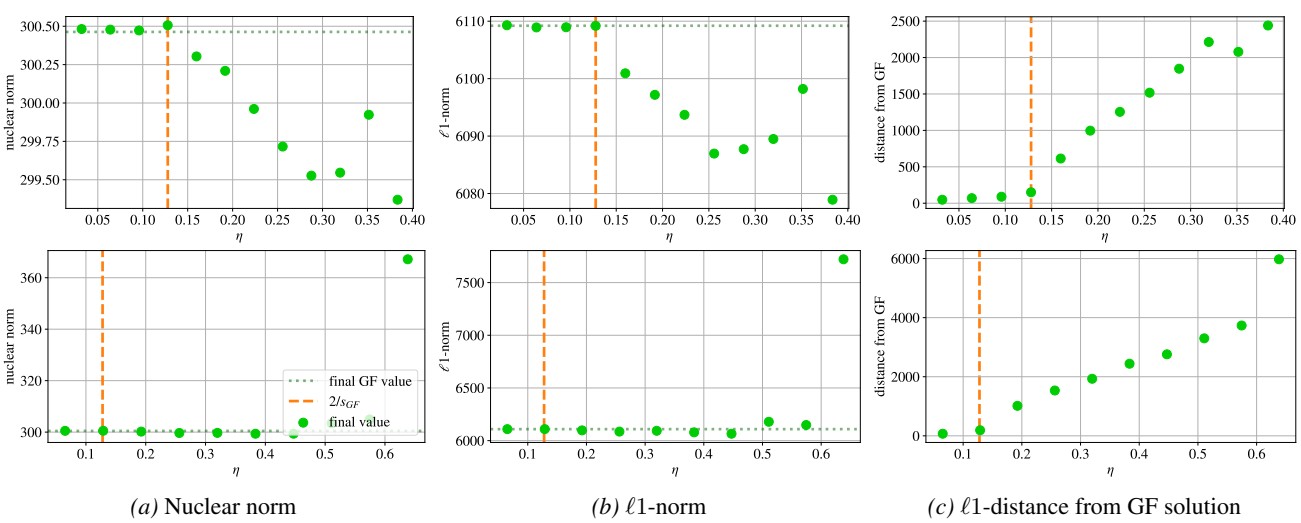

*(a)* Nuclear norm        *(b)* $\ell 1$-norm        *(c)* $\ell 1$-distance from GF solution

*Figure 71.* **FCN-ReLU on full MNIST with the CE loss.** Train loss 0.01

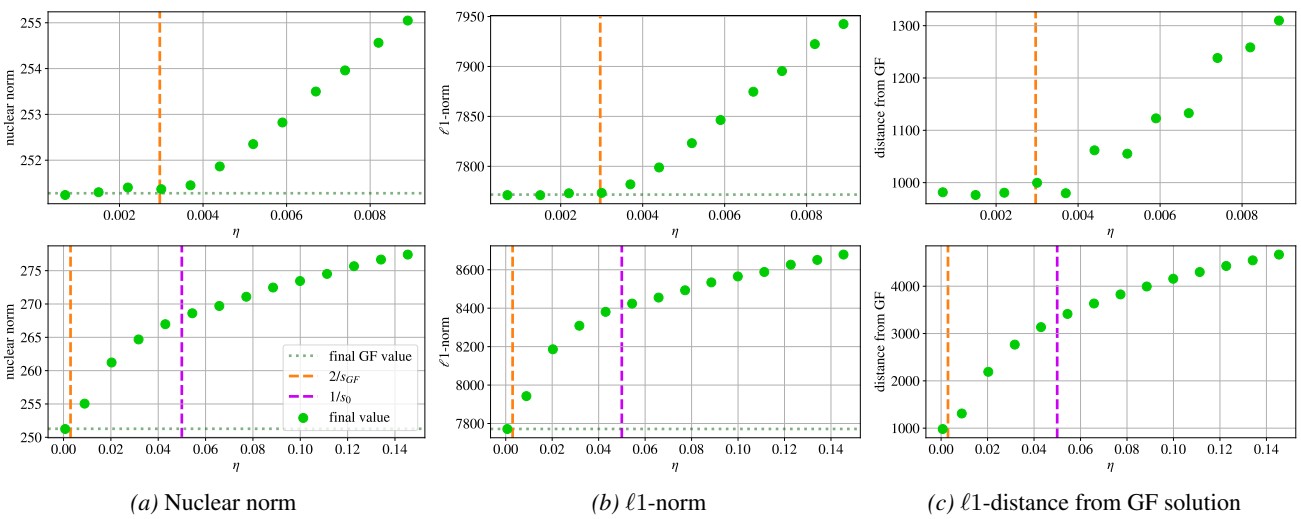

*(a)* Nuclear norm        *(b)* $\ell 1$-norm        *(c)* $\ell 1$-distance from GF solution

*Figure 72.* **FCN-ReLU on CIFAR-10-5k with the MSE loss.** Train loss 0.0001

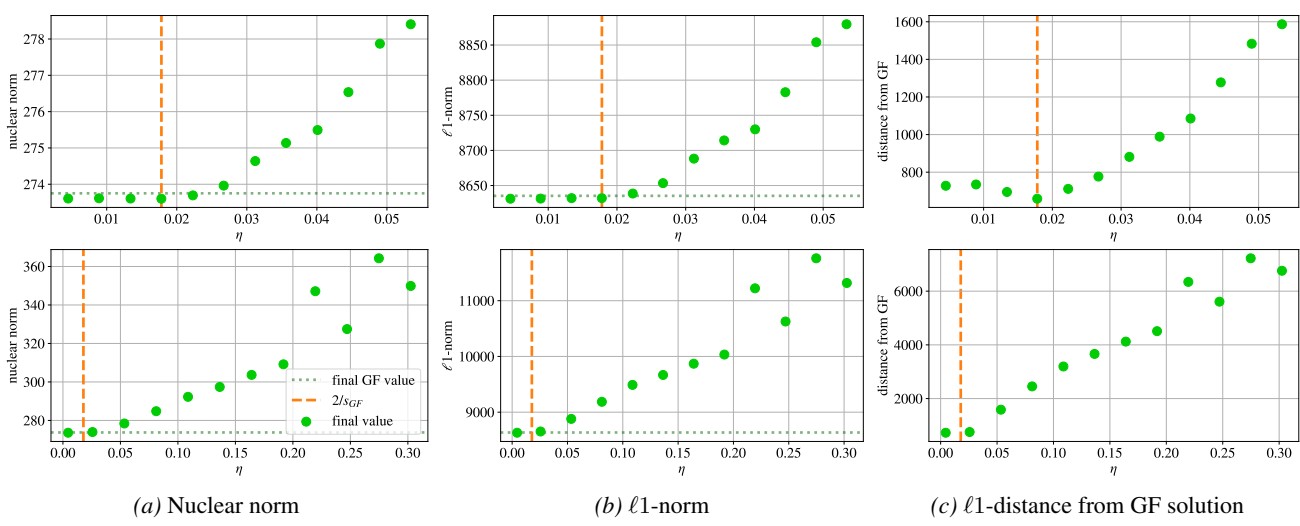

*(a)* Nuclear norm       *(b)* $\ell 1$-norm       *(c)* $\ell 1$-distance from GF solution

*Figure 73.* **FCN-ReLU on CIFAR-10-5k with the CE loss.** Train loss 0.01

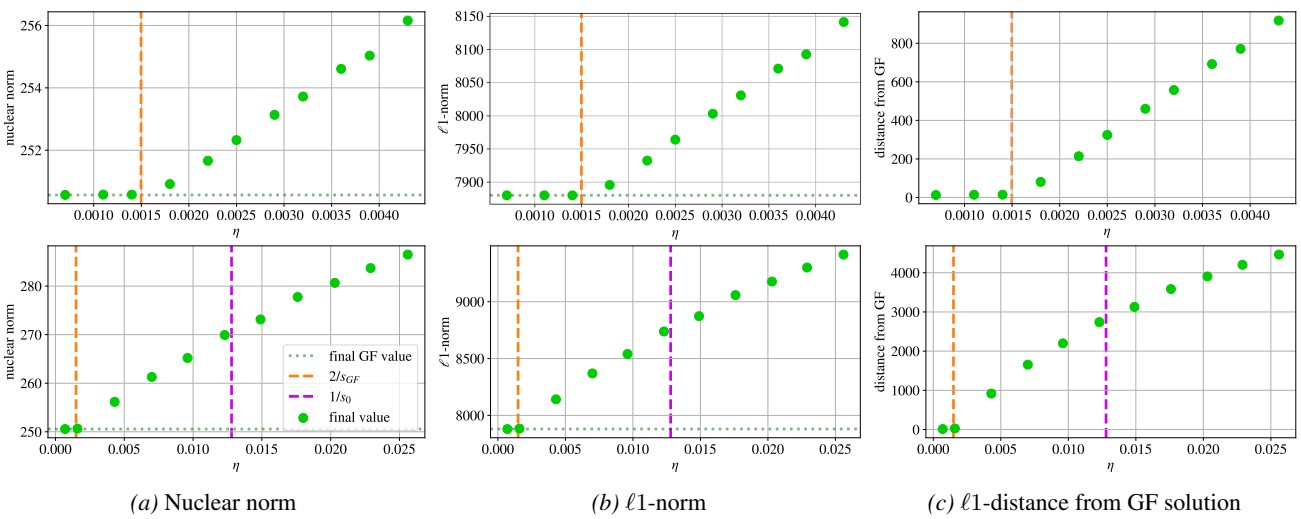

*(a)* Nuclear norm       *(b)* $\ell 1$-norm       *(c)* $\ell 1$-distance from GF solution

*Figure 74.* **FCN-tanh on CIFAR-10-5k with the MSE loss.** Train loss 0.001

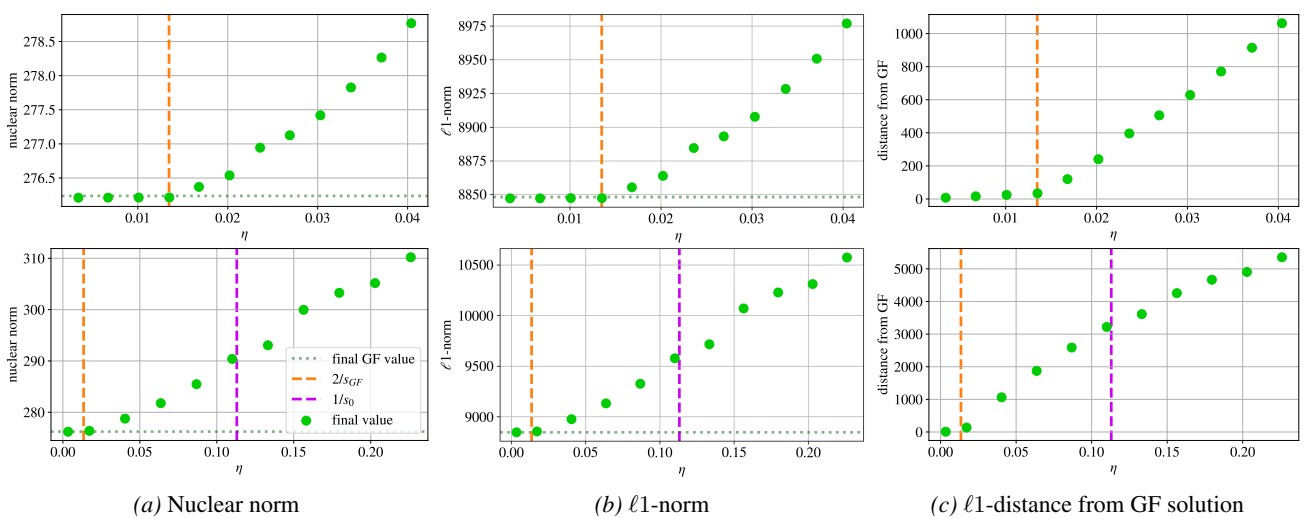

*(a)* Nuclear norm       *(b)* $\ell 1$-norm       *(c)* $\ell 1$-distance from GF solution

*Figure 75.* **FCN-tanh on CIFAR-10-5k with the CE loss.** Train loss 0.01

## J.8.2. CONVERGENCE SPEED AND TEST ACCURACY

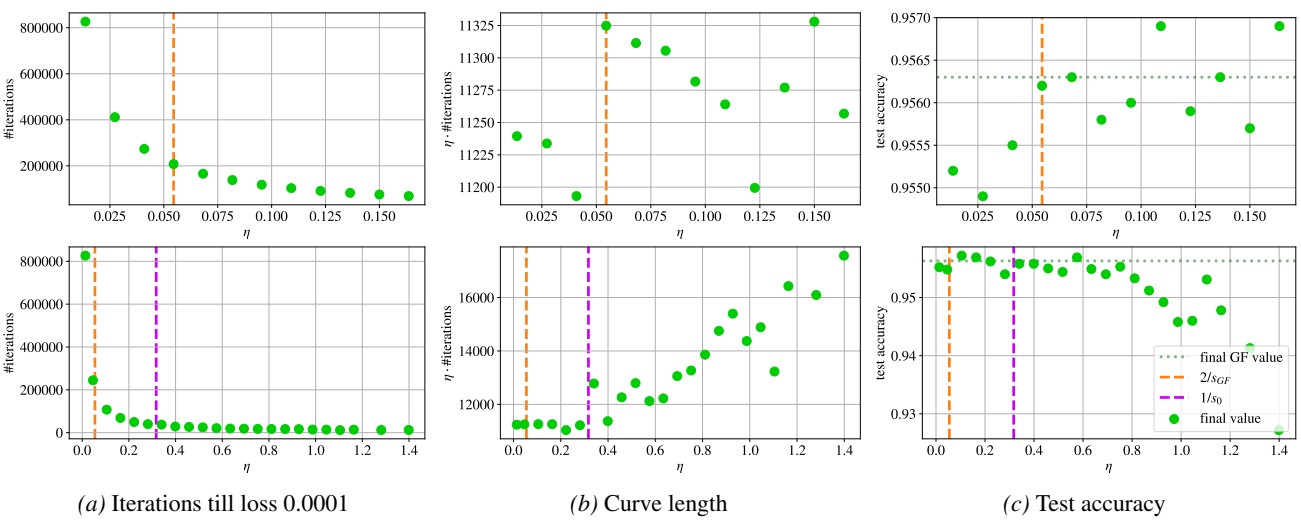

*(a)* Iterations till loss 0.0001       *(b)* Curve length       *(c)* Test accuracy

*Figure 76.* **FCN-ReLU on MNIST-5k with the MSE loss.** Train loss 0.0001

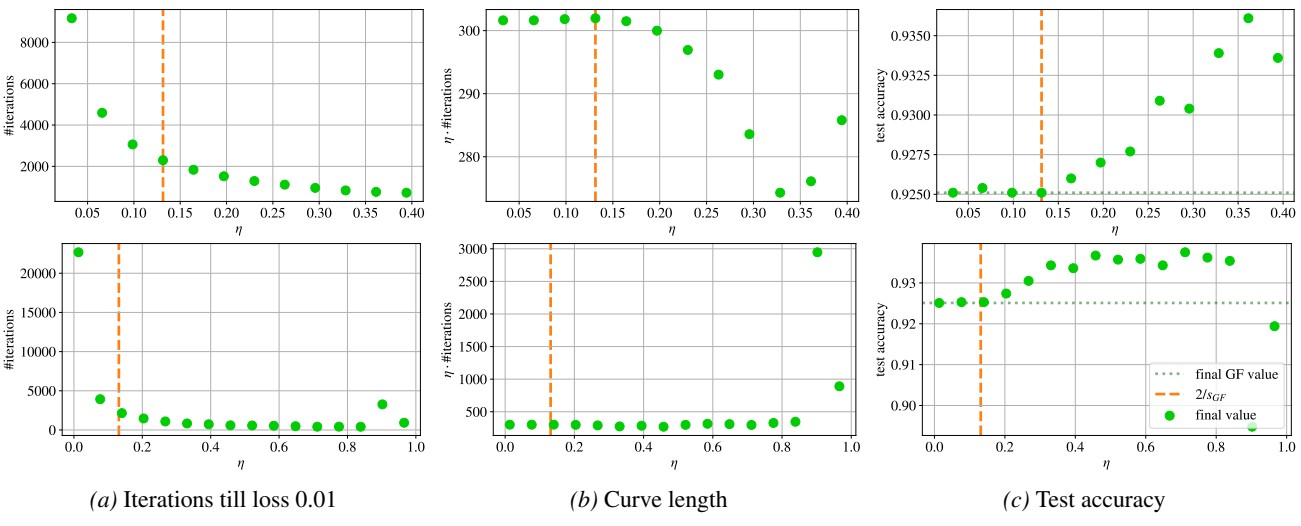

*(a)* Iterations till loss 0.01    *(b)* Curve length    *(c)* Test accuracy

*Figure 77.* **FCN-ReLU on MNIST-5k with the CE loss.** Train loss 0.01

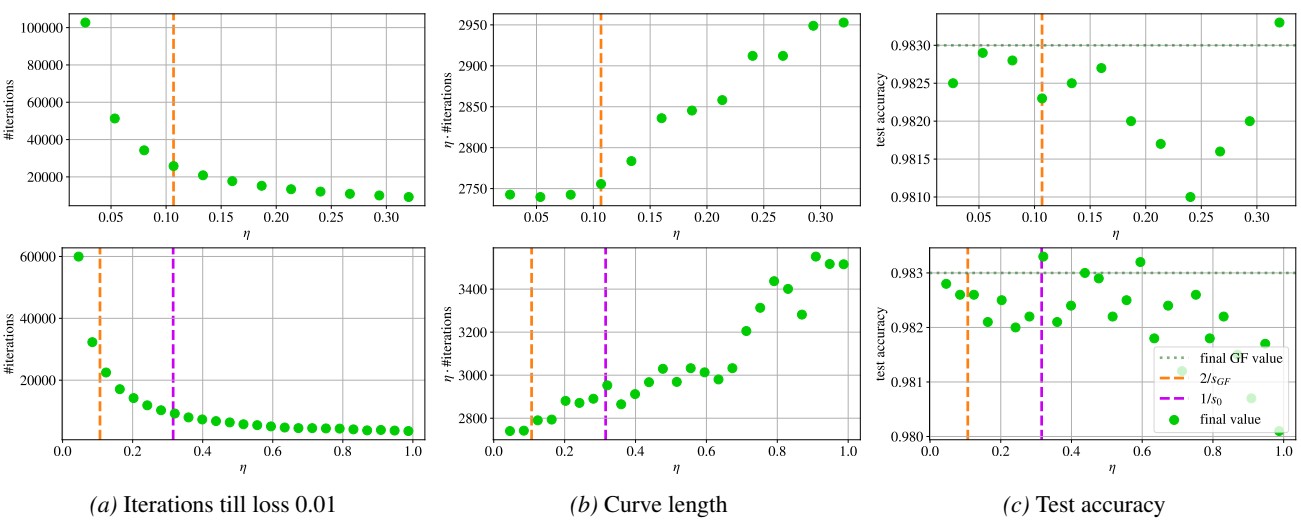

*(a)* Iterations till loss 0.01    *(b)* Curve length    *(c)* Test accuracy

*Figure 78.* **FCN-ReLU on full MNIST with the MSE loss.** Train loss 0.01

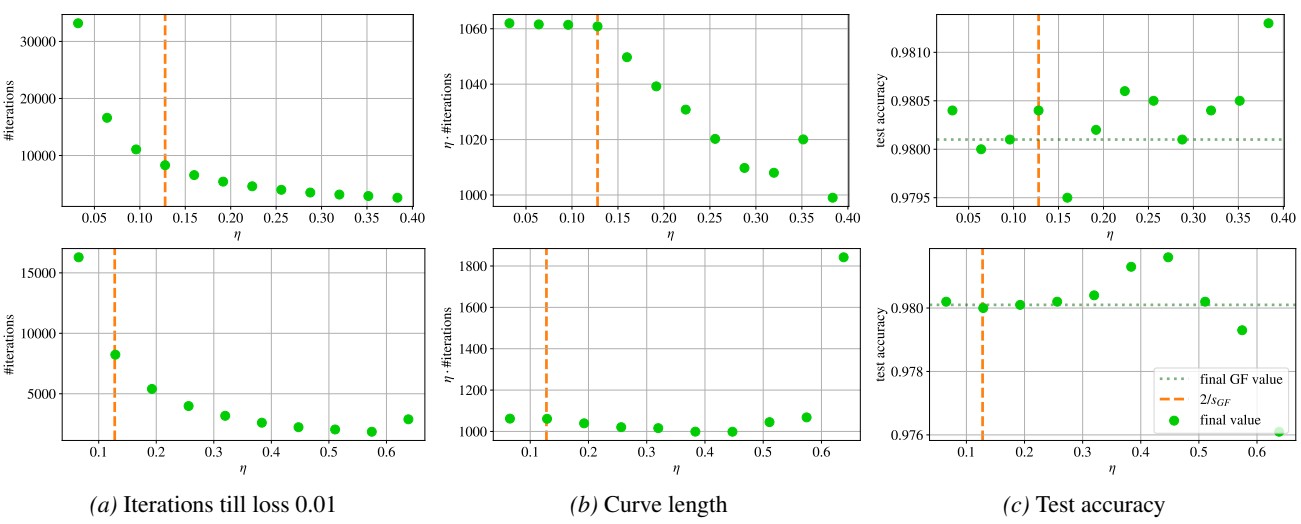

*(a)* Iterations till loss 0.01      *(b)* Curve length      *(c)* Test accuracy

*Figure 79.* **FCN-ReLU on full MNIST with the CE loss.** Train loss 0.01

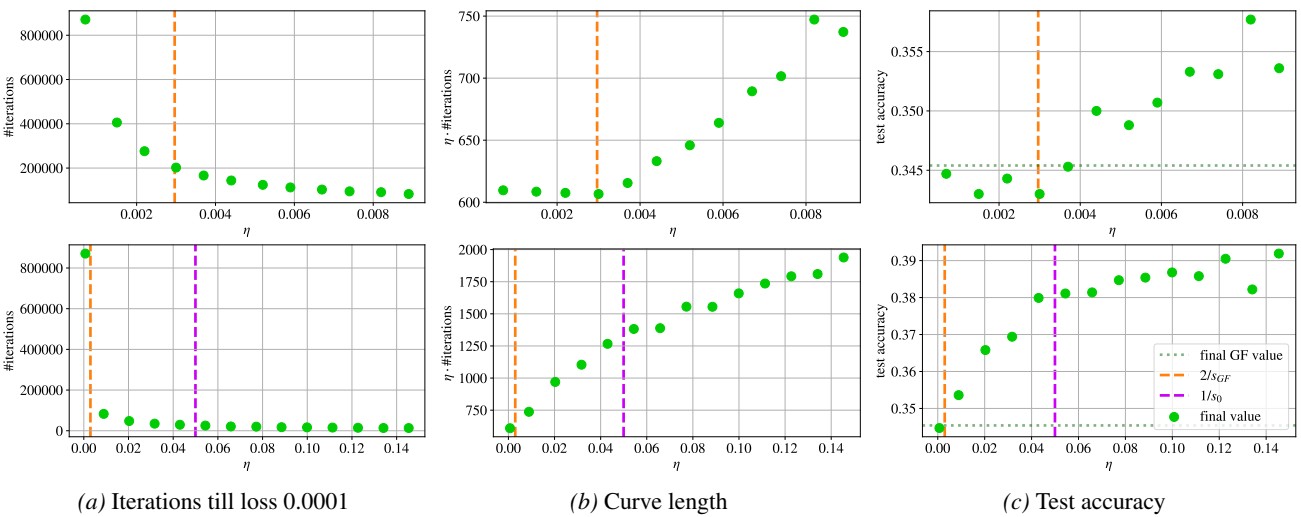

*(a)* Iterations till loss 0.0001      *(b)* Curve length      *(c)* Test accuracy

*Figure 80.* **FCN-ReLU on CIFAR-10-5k with the MSE loss.** Train loss 0.0001

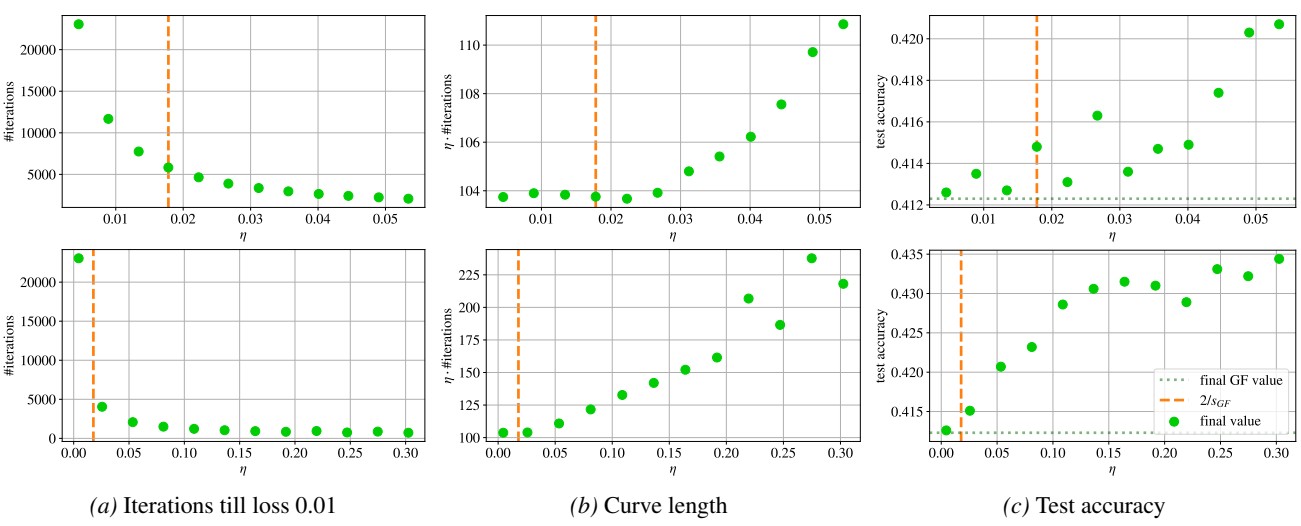

*(a)* Iterations till loss 0.01      *(b)* Curve length      *(c)* Test accuracy

*Figure 81.* **FCN-ReLU on CIFAR-10-5k with the CE loss.** Train loss 0.01

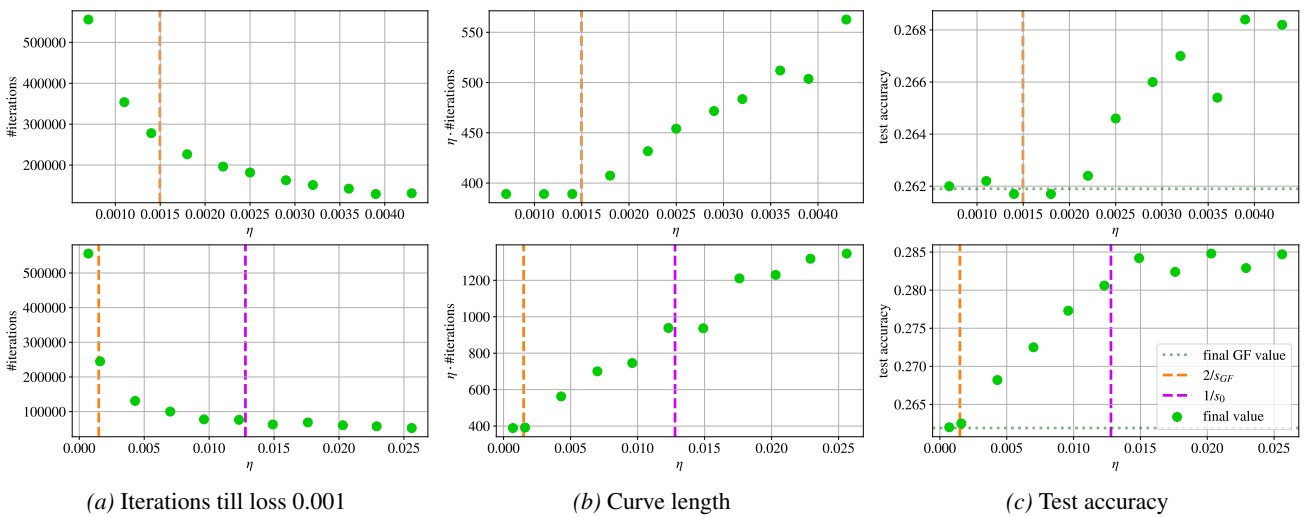

*(a)* Iterations till loss 0.001      *(b)* Curve length      *(c)* Test accuracy

*Figure 82.* **FCN-tanh on CIFAR-10-5k with the MSE loss.** Train loss 0.001

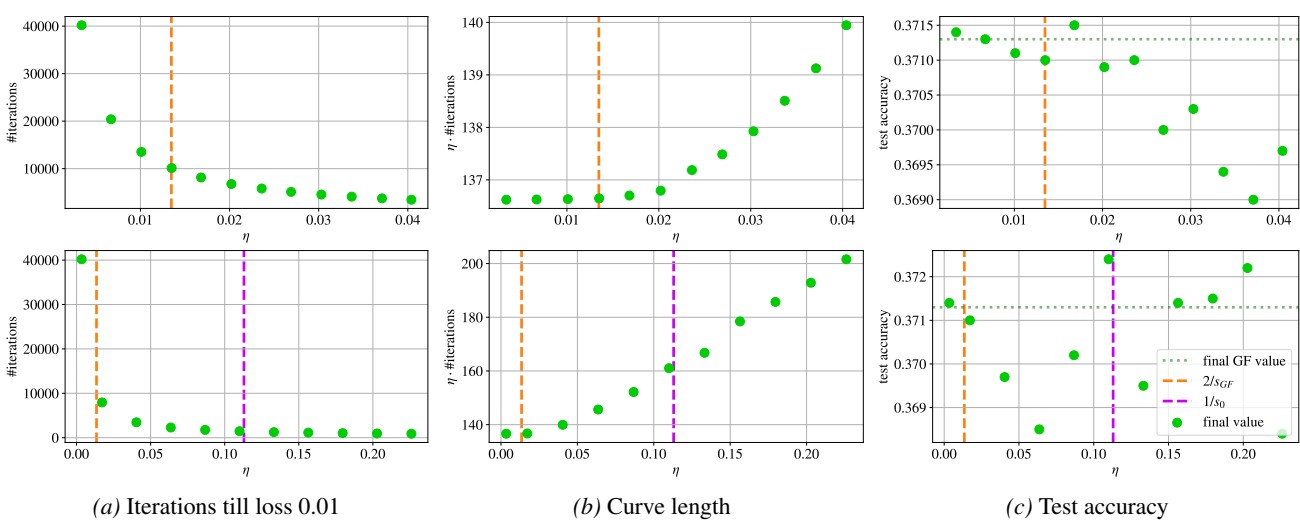

*(a)* Iterations till loss 0.01      *(b)* Curve length      *(c)* Test accuracy

*Figure 83.* **FCN-tanh on CIFAR-10-5k with the CE loss.** Train loss 0.01

