# OpenReview forum: "Conflicting Biases at the Edge of Stability: Norm versus Sharpness Regularization"
_ICML.cc/2026/Conference — ICML 2026 regular_

### Official Review · Reviewer_JJ2s · 2026-03-03

**Soundness:** 3
**Presentation:** 4
**Significance:** 3
**Originality:** 3
**Overall Recommendation:** 4
**Confidence:** 4

**Summary:**

This paper studies the generalization behavior under different learning rates covering gradient flow regime and EoS regime. It demonstrates that the optimal generalization performance appear in EoS regime with a trade-off between minimum parameter norm and minimum sharpness. The authors provide thorough empirical evidence about the comparison between parameter norm, sharpness, and generalization. They also show a simple theoretical example (linear diagonal network with one data point that explicitly computes the empirical and population minimizer) to illustrate the difference between minimum parameter norm, minimum sharpness, and minimum generalization error.

**Compliance With Llm Reviewing Policy:**

Affirmed.

**Final Justification:**

This paper studies the EoS through a trade-off between minimum parameter norm and minimum sharpness, which is novel. The results shown in the paper are rather complete.

The authors addressed my concern regarding the illustrative example of generalization by extending the theory to multiple data point setting; the optimal learning rate is not explored in detail and is left for future work.

I will maintain my evaluation.

**Key Questions For Authors:**

- Figure 3: for (c), I wonder what happen if $\eta$ further increases, will it just blow up (even in smaller learning rate interval?) or it can also show some u-shaped curve. I would expect this u-shape might be very sharp on larger learning rate side, and therefore overall story is more consistent.
- Figure 4: The background color is a bit confusing. It represents the sharpness of the loss which should not change with learning rate, but the colors are different in each plot. Also, not sure whether the trajectory in (b) is hidden behind the label box. It might be better to make it clearer.
- The theoretical setting when illustrating generalization is not very convincing, and may need more discussions. There is only one data point in the empirical loss; thus there are many data structures that can fit this model. The linear regression model used in the paper is reasonable; however, there should also be discussions about other data structures, which can complete the story. For example, when the data is noiseless, the whole manifold is optimal for generalization. There could also be cases where any two of the three min (sharpness, parameter norm, generalization) could coincide. This also echoes with my first point, which I feel the whole story can be made richer instead of just discussing specific theoretical setting or empirical outlier.
- Since the paper is mainly about the optimal generalization performance across different learning rates, it might be better if there were more quantitative discussions about optimal learning rates.

**Limitations:**

yes

**Strengths And Weaknesses:**

**Soundness**
The empirical and theoretical analysis themselves are sound but the whole story could be made more complete. See questions.

**Presentation**
The paper is well written.

**Significance**/**Originality**
This paper studies the generalization behavior in the EoS regime, which is important and novel. The demonstration of optimal generalization seems new and intuitive.

---

> ### Author Rebuttal · Authors · 2026-03-31
>
> Dear reviewer JJ2s,
>
> Thank you very much for your helpful review. We are glad you found our analyses sound and our work well-written. We answer your questions below.
>
> **Figure 3c**
>
> In most of our settings, the considered range of learning rates already extends close to divergence. Near this boundary, there is typically a transition region in which nearby values of $\eta$ may either converge or diverge.\
> For the setup of Fig. 3(c), divergent rates indeed follow soon after the largest value shown in the figure. To test whether, on a finer scale, the curve develops a very steep right-hand side of a U-shape, we conducted a more detailed experiment: Starting from the largest convergent learning rate in our original grid, we iteratively narrowed down the convergence boundary and tested 30+ additional learning rates in this region. In the transition region (granularity of 0.0001) $\eta$ values 0.1619-0.1621, and 0.1623-0.1626 still converge, whereas 0.1618, 0.1622 and 0.1627-0.1630 blow up. We then examined the convergent runs within this interval and do **not** observe a sharp right-hand U-shaped curve. At the same time, we cannot determine whether this is because such solutions do not exist, or if an even finer search would reveal such unstable trajectories.
>
> However, in some settings the right branch does become very steep; see, e.g., Fig. 19(c), bottom row. Fig. 73 for the same setup shows that the curve length becomes especially large there, indicating strong instability.
>
> **Figure 4**
>
> Thank you for pointing this out. The sharpness of the loss is indeed independent of the learning rate and is the same _smooth_ blue background in all four figures. ​​In each panel, we additionally overlay in _solid_ light blue the region where the sharpness is close to $2/\eta$, which does change with the learning rate. This overlay is intended to highlight the learning-rate-dependent stability threshold.\
> The trajectory colors are used to match each trajectory to its corresponding point in Fig. 4(d). For example, the second point in Fig. 4(d) corresponds to the purple trajectory in Fig. 4(b) which passes close to the positive y-axis, in the first quadrant, and is not hidden by the legend. That said, we agree that the current presentation can be made clearer. In the revision, we will update the figure to improve its clarity.
>
> **On the generalization model**
>
> You are right that for only one data point there are many data structures that can fit this model. Still, despite its simplicity, our toy model reproduces core features of our general experiments such as norm and sharpness curves exhibiting increasing and decreasing behavior with sharp phase transitions, and a U-shaped generalization curve.
>
> To make the story more complete, **we have now extended the generalization analysis to the multiple-data-point setting** using the data model introduced in Appendix E. There, we show that the sharpness $S_\mathcal{L}$ concentrates around its expectation, which is proportional to $|\mathbf{w}|_\infty^2$. Using this quantity as the sharpness constraint in the KKT analysis (eq. (31) in Lemma F.1), we characterize the KKT points of risk minimization under these approximate sharpness constraints. This gives a direct analogue of the single-sample picture in Section 3: changing $\eta$ interpolates between a regime where GD behaves like an unconstrained risk minimizer on the interpolation manifold and a regime where GD is effectively restricted by sharpness constraints. Thus, the qualitative algorithmic conclusions of Section 3 extend beyond the single-sample setting to the multi-data-point case. We will include this new lemma and discussion as Appendix F.2 in the revision.
>
> We emphasize that the goal of the theory is to provide an **explicit counterexample** to the claim that either the minimum-norm or the minimum-sharpness solution generalizes best. This provides a formal argument for the trade-off we observe systematically.
>
> **On optimal learning rate choice**
>
> Thank you for this suggestion. Given the foundational nature of our work, we do not yet provide quantitative recommendations for the optimal learning rate, although we agree that this is an important direction for future work. Qualitatively, when the generalization curve improves only to one side, this improvement is toward larger learning rates, in line with previous work that shows EoS _can_ benefit generalization. At the same time, excessive instability appears to be harmful; for example, the high test-loss outliers often coincide with highly unstable trajectories (see response to Q1). Our experiments further suggest that norm and sharpness alone are not sufficient to determine the optimal learning rate.  We view a better understanding of the competing implicit biases induced by different learning-rate regimes as a necessary step toward eventual quantitative guidance.
>
> Thank you again for your valuable feedback. If any further questions arise, we would be happy to clarify them.

---

> > ### Author Rebuttal · Reviewer_JJ2s · 2026-04-01
> >
> > I would like to thank the authors for the reply.
> >
> > I don't have additional questions. All my concerns have been addressed. Please consider revising the corresponding parts of the paper. I will keep my score

---

> > > ### Author Response · Authors · 2026-04-07
> > >
> > > Thank you for the positive feedback and for confirming that all concerns have been addressed.
> > >
> > > We will incorporate these clarifications into the revision, and also add the new multi-data-point generalization lemma. Given that your technical concerns are resolved and the paper is now clearer, we hope you might consider reflecting this in your final assessment.
> > >
> > > We appreciate your time and thoughtful review!

---

### Official Review · Reviewer_PELk · 2026-03-08

**Soundness:** 3
**Presentation:** 3
**Significance:** 3
**Originality:** 3
**Overall Recommendation:** 4
**Confidence:** 4

**Summary:**

This paper studies two implicit biases of SGD together: norm and sharpness regularization, which are often studied separately. The authors argue that the two norms coexist, and the learning rate can be used to dial them. In the small learning rate regime (Gradient flow regime), SGD prefers small norm solutions, whereas at large learning rates, while the norm grows with the learning rate, sharpness decreases. The main result of this paper is that neither minimum sharpness nor minimum norm solutions yield the best generalization performance. Rather, a balance of the two performs the best. Furthermore, they analyze a toy model that shows that neither min norm or min sharpness solutions are optimal.

**Compliance With Llm Reviewing Policy:**

Affirmed.

**Final Justification:**

The paper studies the norm and sharpness together, which is novel and timely. While the authors' rebuttal resolved my concerns, I am maintaining my score because of the limited scope of the paper and the weak link to generalization.

**Key Questions For Authors:**

- Do you think that the results generalize to modern optimization regimes, such as pre-training and finetuning?
- Similarly, how specific are the results to SGD? Is it possible for this tradeoff to disappear when using adaptive optimizers?
- Is it there a causal relationship between norm/sharpness and generalization, or just a correlation?
- I would request the authors to help me understand the cross-entropy results. They appear almost the same with minor differences. But what are the implications for practical training?

**Limitations:**

The results are restricted to small-scale image classification settings. Its unclear if these results have implications for modern training regimes involving language modeling and adaptive optimizers. Nevertheless, I believe future work can build on these ideas.

**Strengths And Weaknesses:**

Strengths:
- The paper is well motivated. Norm and sharpness regularization are studied independently, and analyzing these two phenomena together is timely.
- The main result of the paper, that norm and sharpness regularization are at odds with each other, is quite intriguing.
- The diagonal network analysis acts a good counter example.

Weaknesses:
- The results are restricted to small-scale image classification settings, and its unclear to me if these have any implications on modern training regimes such as language pre-training or finetuning
- While its important to understand these implicit biases in the simpler setting of SGD, it would be interesting to if the results hold for modern optimizers like Adam or Muon.
- I think the causal relationship between norm/sharpness regularization and generalization is missing. Consider the following hypothesis: We expect optimal performance to be between the gradient flow (GF) regime and divergence. Is it there a causal relationship between norm/sharpness and generalization, or just a correlation?

---

> ### Author Rebuttal · Authors · 2026-03-31
>
> Dear reviewer PELk,
>
> Thank you very much for your thoughtful feedback! We address your questions and concerns below.
>
> **On different optimizers**
>
> We would first like to clarify that our paper studies **full-batch gradient descent**, not SGD. Our current results are therefore specific to the deterministic full-batch GD setting. This was a deliberate choice as our goal was to isolate as cleanly as possible the trade-off between norm bias and sharpness bias induced purely by the learning rate $\eta$, without the additional effects introduced by mini-batch noise or momentum.
>
> We agree that it is interesting to understand whether the same trade-off persists for modern optimizers. There is prior work suggesting that stochasticity is associated with a bias toward flatter/wider minima, so there we would assume that a similar trade-off exists but is modulated by the optimizer choice.
> To address this point explicitly in our setup, we will add a new subsection to Appendix I with exploratory experiments using different optimizers. A full systematic study is beyond the scope of the present paper, given its already substantial length, and we leave it to future work.
>
>
> **On modern optimization regimes**
>
> The applicability of our findings to modern large-scale optimization regimes, such as pre-training and fine-tuning, remains an open question beyond the scope of this work. In Appendix I.14, we already include exploratory experiments beyond the image domain, specifically sequence modelling and tabular tasks. They suggest that the observed trade-off is not limited to images, although clear qualitative differences remain and require further study beyond this work.
>
> As with the previous question on different optimizers, this is precisely why we focus here on a simpler setting that isolates the learning-rate-driven trade-off between two particularly common and practically relevant implicit biases, namely norm bias and sharpness bias, which are closely connected to widely used techniques such as weight decay and SAM. Modern training pipelines introduce several additional sources of implicit bias beyond these mechanisms. As examples, LoRA restricts updates to a low-rank subspace, and the attention architecture also enforces low rankness (key/query matrix). We therefore view the broader message of our work as showing that multiple seemingly beneficial biases may conflict, and often a balance between them is needed for best generalization. We will update the discussion to make this clearer.
>
>
> **On causal relationship**
>
> Thank you for raising this point. If the question is whether higher norm directly causes lower sharpness, or vice versa, our answer is no: we do not claim a direct causal relationship between norm and sharpness in general. Likewise, we do not establish this relationship to generalization. For the empirical results, the evidence is therefore correlational: as the learning rate varies, GD selects solutions with different norm/sharpness profiles, having different generalization performance.
>
> In the diagonal model, however, Proposition 3.2 shows that low-norm and low-sharpness minima do not coincide, and the optimal generalization is attained at neither. Thus, as $\eta$ changes, GD selects different parts of the solution set. Because the low-norm and low-sharpness minima are geometrically distinct, moving toward lower-sharpness solutions in this model is accompanied by a larger norm, which for carefully tuned $\eta$ yields the risk minimizer. So in the toy model the guiding mechanism is the (approximative) learning-rate-dependent solution selection.
>
> **On CE loss**
>
> CE shows an important difference in how the trade-off manifests over the course of training (Fig. 13 show the temporal evolution of  MSE and CE side by side, Appendix I.4 and I.11 discuss it). In particular, while the loss sharpness’ iterates still rise toward and oscillate around $2/\eta$, the final sharpness under CE subsequently falls below this value, whereas for MSE it remains close to $2/\eta$ until convergence. Correspondingly, the CE training loss decreases again monotonically later in training, while under MSE it stays in this oscillatory regime until convergence. This phenomenon is less visible in the original EoS paper of Cohen et al., since training there is stopped earlier, during the oscillatory phase. Still, this oscillatory phase already steers GD toward different parts of the loss landscape and thereby induces a similar norm/sharpness trade-off. \
> Since CE is a highly important loss function used in many classification tasks, we can expect this stabilization phenomenon for extended training in practice. These insights can also inform learning rate choice, which has to account for the higher instability under CE.
>
> Thank you again for your helpful review. If you have any further questions, we would be glad to answer them.

---

> > ### Author Rebuttal · Reviewer_PELk · 2026-04-02
> >
> > I thank the authors for their rebuttal. Most of my concerns are resolved. I am currently going through other reviews and the responses. I will update my score before the discussion period ends.
> >
> > > our paper studies full-batch gradient descent, not SGD
> >
> > I meant GD and not SGD. I apologize for the typo.

---

> > > ### Author Response · Authors · 2026-04-07
> > >
> > > Thank you! We appreciate your engagement with our rebuttal and your positive note regarding the score update.
> > >
> > > We are glad to hear the concerns are resolved. We will ensure the final version reflects these clarifications, and will add our exploratory experiments using different optimizers to Appendix I. Based on early results, we can confirm that for SGD and shampoo a similar qualitative trade-off emerges across various settings. We are continuing to run experiments for Muon and Adam and will include all of these in the final manuscript.
> > >
> > > Thank you for your time in evaluating our work.

---

### Official Review · Reviewer_2LrE · 2026-03-13

**Soundness:** 2
**Presentation:** 3
**Significance:** 1
**Originality:** 2
**Overall Recommendation:** 2
**Confidence:** 4

**Summary:**

This paper argues that the implicit bias of full-batch gradient descent should not be understood through a single regularization principle. Instead, the authors propose that the learning rate controls a trade-off between two competing biases: a gradient-flow bias toward low parameter norm (active at small learning rates) and an Edge-of-Stability-related bias toward low sharpness (active at larger learning rates). Empirically, the paper reports a sharp phase transition in the *final* trained solution as a function of learning rate: below a critical $\eta_c$, the final norm and sharpness are nearly constant, whereas above $\eta_c$ the sharpness decreases and the norm increases. In several configurations, intermediate learning rates also achieve the best test performance, producing a U-shaped generalization curve. To interpret these observations, the paper analyzes a shallow diagonal linear network trained on a single data point and proves that the norm-minimizer and sharpness-minimizer on the interpolation manifold are generically distinct, and that neither extreme alone necessarily minimizes the generalization error.

**Compliance With Llm Reviewing Policy:**

Affirmed.

**Final Justification:**

My low score is not because I believe the paper is sloppy or technically incorrect. The issue is a mismatch between framing and support. The rebuttal to my review narrows the intended claims substantially: no classical EoS-dynamics theorem, no causal mediation claim, no theorem-backed statement beyond the toy model, and no explanation for the sharp transition. Under that narrower reading, the remaining contribution could be interesting but, in my view, too modest for acceptance at ICML.
Furthermore, the narrowing of the authors further decreases significance in my opinion.
Under the broader reading suggested by the title and narrative, the evidence is insufficient. This is why I remain below threshold.

**Key Questions For Authors:**

1. In what precise sense do you claim that the toy model exhibits Edge-of-Stability, rather than merely a finite-step stability constraint on convergent limits? Can you provide a dynamical statement in the toy model that parallels the classical EoS literature (progressive sharpening, non-monotone loss, curvature hovering near $2/\eta$)?

2. What is the strongest theorem-backed claim you believe the paper establishes about *realistic* neural networks, as opposed to the stylized diagonal model?

3. What evidence justifies treating parameter norm as an operative implicit bias in modern architectures, especially in the presence of normalization or reparameterization effects?

4. How sensitive is the toy-model generalization picture to the specific sampled training datum? In the one-sample setting, the solution-set geometry can vary substantially across draws. Do the qualitative conclusions survive systematic averaging over draws of $(x_0, y_0)$?

5. What happens if runs are compared under alternative stopping rules—fixed compute, fixed number of epochs, or validation-based stopping—rather than a fixed train-loss threshold $\varepsilon$?

6. Can you separate more explicitly which claims are theorem-level, which are empirical observations, and which are broader conjectural interpretations? The current prose sometimes blurs these boundaries.

7. How should readers think about the beyond-EoS regime relative to your main message? Does the norm–sharpness trade-off persist there, or is your claim intended only for convergent trajectories between the flow-aligned and convergent EoS regimes?

8. Can you add multi-seed results with confidence intervals for the core norm / sharpness / test-error claims in the main text?

9. Do you have a mechanistic explanation for why the transition in final norm and final sharpness is so sharp near $\eta_c$, rather than a gradual crossover?

**Limitations:**

The paper's limitations are not merely technical footnotes; they materially affect the scope of the conclusions. The theory is restricted to a very stylized model with a single training sample, and the associated generalization analysis relies on idealized distributional assumptions that the appendix itself acknowledges. The empirical study is limited to full-batch gradient descent with fixed learning rates, so it does not directly address the optimizers, stochasticity, regularizers, and training recipes that dominate practical deep learning. The stopping protocol—training to a fixed loss threshold—is entangled with early-stopping effects, and the paper itself notes this connection without resolving it. The U-shaped generalization curve is not universal across settings, and scaled-initialization experiments show that the mechanism can change qualitatively outside the small-initialization regime.

Moreover, although alternative sharpness notions are checked empirically, the interpretation of sharpness remains delicate because worst-case Hessian sharpness is not invariant under reparameterization. The beyond-EoS regime is acknowledged but excluded from the analysis, leaving unclear how far the proposed trade-off story extends in learning rate. Most importantly, the paper does not causally isolate norm and sharpness as separate mediators of the generalization effect: it shows a structured co-variation with learning rate, but several plausible parallel explanations—early-stopping artifacts, trajectory displacement, compute-path effects, initialization-regime dependence, and coincident biases—remain open. In my view, the core bridge from the toy theory to the broad generalization message remains too weakly established.

**Strengths And Weaknesses:**

## Strengths

1. **Important and timely question.** Asking whether the implicit bias of GD is learning-rate-dependent rather than fixed is the right thing to study, and the paper frames this clearly.

2. **Broad empirical sweep.** The experiments cover multiple architectures (FCN, CNN, ResNet, ViT), datasets, activation functions (ReLU, tanh), losses (MSE, CE), parameterizations ($\mu$P, kernel), initialization scales, norm choices, and several sharpness notions including scale-invariant variants. For this style of paper, the breadth is commendable and well above average.

3. **The oscillation-artifact concern is addressed well.** The temporal plots showing that sharpness oscillates around $2/\eta$ while the norm rises monotonically without oscillation are effective. The per-layer norm analysis confirming an increasing trend across all layers further strengthens this point. This directly addresses a natural concern that the norm increase might be a trivial byproduct of oscillatory dynamics.

4. **Clean toy geometry.** Proposition 3.2 makes the separation between the norm-minimizer and the sharpness-minimizer on the interpolation manifold very explicit. The added KKT analysis of the sharpness-constrained risk problem (Lemma E.1) and the comparison of GD endpoints to three idealized algorithms strengthen the toy-model section. More broadly, the theory does support a genuine counterexample-level message: single-bias explanations need not be universally sufficient.

5. **Careful self-awareness in places.** The paper explicitly states that its main phase transition is a comparison of *final solutions across learning rates* and "does not correspond" to the classical progressive-sharpening-to-EoS transition for a fixed $\eta$. I appreciate this honesty; it also informs my main concern.

---

## Weaknesses

### Conceptual / Framing

1. **The paper is stronger as a finite-learning-rate endpoint-selection study than as an EoS paper.** The title and much of the narrative invoke the Edge of Stability, but the theoretical analysis does not really analyze EoS *dynamics*. What is actually proved is (a) a geometric separation between norm-minimizers and sharpness-minimizers on a stylized manifold, and (b) a stability-based restriction $S_L(\theta^*) < 2/\eta$ on convergent GD limit points. Classical EoS is a *dynamical* claim about a fixed learning rate: progressive sharpening, entry into a regime where curvature hovers near $2/\eta$, non-monotone loss, and often faster optimization than stable schedules. The theory here is a cross-$\eta$ endpoint comparison combined with a sharpness constraint. These are related but not the same thing. In particular, the current analysis does not establish bona fide EoS behavior in the strong dynamical sense.

2. **The headline generalization claim is not causally established.** The paper shows that intermediate learning rates *often* coincide with better test loss and with intermediate norm–sharpness trade-offs. But this remains an observational association across learning-rate sweeps. The paper does not perform the kind of intervention needed to isolate the mechanism: holding norm approximately fixed while varying sharpness, holding sharpness approximately fixed while varying norm, or running a mediation-style analysis testing whether the learning-rate effect on test loss is actually carried through these two variables. So the evidence supports "learning rate, norm, sharpness, and test loss move together in a structured way," but not the stronger causal statement that the norm–sharpness balance is *the* operative cause.

3. **The U-shaped generalization curve is not universal.** The paper's own Table 1 and Figure 3 show that the U-shape appears in some configurations but not others. CIFAR-10-5k with MSE does not follow it; large-initialization experiments show that the mechanism can qualitatively change or even reverse. The most defensible conclusion is "sometimes balancing these biases helps generalization," which is materially weaker than the paper's broader phrasing.

4. **The sharp transition is documented more clearly than it is explained.** A critical learning rate at which norm and sharpness change abruptly is an interesting empirical fact, but the manuscript does not yet provide a convincing mechanism for why the transition should be so sharp, what determines its form, or in which settings it should disappear.

### Theoretical

5. **The theory remains too narrow for the headline message.** The toy model is a shallow diagonal linear network trained on a *single* data point with an idealized Gaussian generalization model that the appendix itself calls "idealized." In that regime, the empirical risk $\tilde{L}$ is a very weak proxy for the population risk $L$, and the discrepancy between the norm-minimizer and the risk-minimizer may largely reflect the one-sample nature of the setup rather than a general failure of norm-based explanations in realistic settings. The authors are entitled to present this as a counterexample, but then the paper should be much more disciplined about not extrapolating from that counterexample to broad claims about deep-network generalization.

6. **The beyond-EoS regime is still not integrated.** The paper now cites the relevant literature and explicitly excludes stable oscillatory trajectories from its analysis. But this leaves open whether the proposed norm–sharpness trade-off is truly global in learning rate or mainly a phenomenon between the flow-aligned regime and the convergent EoS regime. At even larger learning rates, the behavior could be qualitatively different, which would complicate the monotone trade-off story.

### Empirical / Causal

7. **Learning rate is not cleanly isolated from stopping time.** The comparison is made at a fixed training-loss threshold $\varepsilon$, and the paper itself notes that varying $\varepsilon$ is naturally related to early stopping and changes the critical learning rate. This means the reported endpoint geometry is partly entangled with when each run is stopped, not purely with how the learning rate biases the trajectory. No ablation under alternative stopping rules (fixed compute, fixed epochs, validation-based stopping) is provided.

8. **The central latent variables are not directly manipulated or isolated.** A genuinely causal paper would need to show that norm and sharpness are separately mediating the generalization effect. The current evidence is a structured co-variation with learning rate, which is consistent with many parallel explanations.

9. **Statistical support is limited.** The key figures are mostly deterministic single-seed curves without uncertainty quantification. For a paper claiming sharp transitions and U-shaped generalization optima, multi-seed confidence intervals should be in the main text, not only a few appendix checks.

10. **Parameter norm as the operative bias in realistic networks is still unjustified.** The paper de-emphasizes a specific norm and shows robustness across norm choices, which is good. But it still does not explain why the *parameter norm of modern architectures* should be viewed as the relevant implicit bias for generalization rather than a loosely correlated proxy. This concern is especially acute in architectures with normalization or strong reparameterization invariances, where parameter norm is not obviously the relevant complexity quantity for generalization. The suggestion that norm-minimization may instead regulate "relations between weights" is interesting, but it is not yet backed by theory or experiment.

---

## Plausible Parallel Explanations Not Ruled Out

Several alternative explanations remain live and are not experimentally discriminated against:

- **Early-stopping / threshold artifact.** Since all comparisons use a fixed loss threshold, the observed test-loss differences may reflect where each learning rate lands along the train/test trajectory rather than a norm–sharpness balancing law. This is, in my view, the strongest competing explanation.

- **Trajectory displacement beyond norm and sharpness.** Appendix I.10 shows that the GD solution can move far from the GF solution even when coarse summary statistics remain similar. Learning rate may matter because it redirects the solution in parameter space, with norm and sharpness being incomplete shadows of that displacement.

- **Compute / optimization-path effects.** The number of iterations to reach the threshold changes significantly with $\eta$. Generalization differences could partly come from different amounts of implicit iterative regularization along the path.

- **Initialization-regime dependence.** The scaled-initialization experiments show that for large enough initialization the mechanism changes qualitatively—even reversing some patterns. This is evidence that the story is regime-dependent, not universal.

- **Reparameterization structure.** With normalization layers, sharpness can scale roughly like $1/\|w\|^2$, so part of the observed anti-correlation between sharpness and norm may be structural to the parameterization rather than causally linked to generalization.

- **Cross-entropy-specific loss geometry / margin growth.** In cross-entropy settings, increasing norm can itself flatten parts of the loss landscape and modify convergence behavior. Some of the reported sharpness changes may therefore reflect loss-geometry effects specific to CE rather than a general norm–sharpness balancing mechanism.

- **Multiple coincident biases.** Flatness, low norm, low rank, and other implicit biases may align only in some regimes. In that case, the apparent success of an intermediate learning rate could reflect several coincident regularization effects rather than a single identifiable two-bias balance.

- **Survivorship over convergent runs.** If only convergent trajectories are compared, large parts of learning-rate space are silently excluded. The observed test-loss pattern may partly reflect which runs survive the convergence filter.



## Summary

I find the empirical phenomenon interesting and the experimental sweep commendably broad for a curvature-heavy paper. Within the authors' controlled protocol—full-batch GD, identical initialization within each sweep, and comparison at a fixed train-loss threshold—the narrower claim that changing $\eta$ changes the geometry of the final endpoint is reasonably well supported. However, I remain unconvinced by two aspects of the current manuscript. First, the Edge-of-Stability framing is not matched by what the theory actually establishes: the toy model provides a *stability-constrained endpoint selection* picture, not a theory of EoS dynamics. Second, the stronger generalization claim—that good test performance arises *because* GD balances norm and sharpness—is supported only correlationally, not causally. The theory backing this claim is a single-sample counterexample in a highly stylized model, which is too narrow to support the headline message about realistic neural networks.

---

> ### Author Rebuttal · Authors · 2026-03-31
>
> Dear Reviewer 2LrE,
>
> Thank you for your detailed review. We believe several of your concerns stem from a broader reading of our claims than intended. Since several points rely on this understanding, we first clarify the scope of the paper. If there are passages in the manuscript that you believe make such claims, we would be grateful if you could point us to them so we can revise them. For brevity, we refer to a Strength/Weakness/Alternative Explanation/Question as S/W/E/Q.
>
> 1. **This is not a classical EoS-dynamics paper.** As you already note in S5, our main phase transition is a comparison of final solutions across learning rates, not a temporal one. Accordingly, W1 and the corresponding statement in the summary appear to evaluate a stronger dynamical EoS claim than we make, whereas E2 corresponds to the claim we make (role of learning rate on sharpness/norm bias).
> 2. **We do not make a causal claim.** In particular, we do not establish norm and sharpness are individual causal mediators of generalization, nor that one causally determines the other. Our claim is narrower: varying the learning rate changes the implicit bias of GD and thereby changes the norm/sharpness profile of the selected solution. In the toy model, we obtained that the minimizers differ and how they relate to generalization; empirically, we document the same trade-off systematically across learning rate. This addresses W2/W8/E5/E6.
> 3. **We do not claim that the U-shaped generalization curve is universal**. In fact, W2 already describes this to hold “often” which aligns with our empirical findings. This addresses W3.
> 4. **We restrict the analysis to convergent trajectories.** The beyond-EoS regime is therefore explicitly outside the scope of our main claim. Once GD no longer converges, there is generally no final solution whose norm and sharpness can be compared in the manner studied here. This addresses W6/E8.
> 5. **Our claims concern the small-initialization regime**. The large-initialization figures are included to show that the mechanism can qualitatively change outside this regime, not to claim universality or robustness beyond it. This addresses E4.
> 6. **We do not claim theoretical guarantees for realistic deep networks.** Rather, the aim of the theory is to formalize the trade-off observed empirically in a toy setting. This addresses Q2.
> 7. **We do not claim that norm and sharpness are the only relevant implicit biases.** On the contrary, part of our point is that the picture is more complex, and that even two common and practically relevant biases can already conflict. This addresses E7.
>
> To summarize, E2, E4-E8 and W1-W3, W6, W8 appear to evaluate broader claims than those made in the paper. We hope the scope clarifications above resolve these concerns.
>
> Let us next answer your questions:
>
> **Q1:** For large $\eta$, the toy model exhibits EoS along convergent trajectories (Fig. 16). We do not claim a full classical EoS dynamics theorem.
>
> **Q2:** None beyond the toy model. For realistic networks, our claims are empirical.
>
> **Q3/W10:** We study parameter norm because it is both practically relevant (via weight decay) and theoretically motivated by the GF literature, where it is associated with parsimoniously structured solutions; see Appendix A for details. ​​Empirically, across our setups, including normalization layers, the GF regime consistently selects lower-norm solutions.
>
> **Q4/W5:** We extended our generalization analysis to multiple data points, see our reply to reviewer JJ2s.
>
> **Q5/W7/E1/E3:** For our scope, a fixed training-loss threshold is the natural stopping rule, since we compare convergent endpoints across learning rates. We compare different loss thresholds in Fig. 2. Fixed compute/epochs would compare runs at different optimization stages (cf. Fig. 13), and validation stopping would introduce an explicit generalization-based rule.
>
> **Q6:** Section 2 is experiments, Section 3 theory. Could you point us to any phrasing in the paper which feels unclear in this respect?
>
> **Q7:** Yes. Our claim is only for convergent trajectories.
>
> **Q8/W9:** We believe confidence intervals don't make sense in our set-up, since the gradient flow solutions differ slightly with initialization, and our claims are relative to the learning rate which is influenced by this.
>
> **Q9/W4:** We don’t have a good explanation why the phase transition is so sharp. In particular, as we show with the distance from the GF solution (see Appendix I.10), the trajectories deviate from each other already for learning rates in the flow-aligned regime, despite their limits still matching in sharpness and norm values. We also mention that the norm increase is not always occurring at but sometimes at a larger step size (cf. Fig. 18).
>
> This addresses all items in E, except E6, which we discuss in Appendix I.4 and our reply to reviewer PELk.
>
> We hope our response clarifies the intended scope of the paper and addresses the concerns raised in your review.

---

> > ### Author Rebuttal · Reviewer_2LrE · 2026-04-02
> >
> > I thank the authors for the detailed rebuttal. The response usefully clarifies that the intended scope is narrower than I had inferred in some places: the paper is not meant as a theory of classical Edge-of-Stability dynamics, does not claim a causal mediation result for generalization, does not claim universality of the U-shaped curve, and does not claim theorem-backed statements beyond the toy model. *These clarifications are helpful, however, further extensively narrow the usefulness of this paper and make it not buildable upon.* Which is a requirement for 4:
> > > "[...] with a contribution that others are likely to build on [...]"
> >
> > I want to warmly suggest to the authors to spend a bit more energy on the paper which could have potentials, but not in the current form which needs an extensive rewrite and even a different title. Such extensive rewrite is not what ICML guidelines allow for the camera ready version.
> >
> > ### **Further comments for the Authors, etc.**
> >
> > I will spend some more words for the authors, other reviewers, and AC for why this rebuttal does not change my overall recommendation.
> > Under this narrower reading, several of my main concerns remain and some are effectively confirmed by the rebuttal. The contribution then becomes: (i) an empirical observation that, under a controlled full-batch, fixed-threshold protocol, varying the learning rate changes the norm/sharpness profile of the final convergent endpoint; and (ii) a stylized *(shallow diagonal linear)* network *(trained on a single data point)* counterexample showing that norm- and sharpness-based biases can conflict. I agree that this could be somewhat interesting **if** properly argued to be novel and surprising.
> > But, for sure, I do not find it sufficient for acceptance at ICML in its current form.
> >
> > First, the EoS framing remains stronger than the actual support. The rebuttal explicitly says that this is not a classical EoS-dynamics paper and does not provide a full EoS theorem in the toy model. This leaves the current **title** and narrative stronger than what is formally established.
> >
> > Second, the main generalization interpretation remains too weakly supported. The rebuttal states that the paper does not make a causal claim about norm and sharpness as individual mediators of generalization. I appreciate that clarification, but it means the paper should be read as *documenting a structured association* rather than identifying the mechanism behind improved test performance. For me, this materially weakens the headline message.
> >
> > Third, the theory-to-practice bridge remains too thin. The authors state that they do not claim theorem-backed guarantees for realistic deep networks. Again, this is a fair clarification, but it leaves the theory primarily at the level of a stylized counterexample rather than a supporting explanation of the empirical behavior in modern models. Relatedly, motivating parameter norm via weight decay or gradient-flow literature is reasonable, but it still does not resolve the concern that parameter norm is not obviously the relevant complexity quantity in normalized or reparameterization-invariant architectures.
> >
> > Fourth, several empirical issues remain unresolved. Defending the fixed training-loss threshold as the natural stopping rule for the intended scope does not remove the fact that stopping time is entangled with the reported endpoint geometry. Likewise, I do not find the response on uncertainty quantification persuasive: even if the exact gradient-flow solution shifts with initialization, one can still report seed variability of the central curves, transition locations, or relative ordering of norm, sharpness, and test loss across learning rates.
> >
> > Finally, the rebuttal acknowledges that there is still no good mechanistic explanation for why the transition is so sharp. Since this phase transition is one of the most striking empirical findings in the paper, the absence of such an explanation remains a substantial weakness. The mention of a multi-sample extension in another response may be promising, but without details here or integration into the manuscript, it does not materially change my assessment.
> >
> > For these reasons, my evaluation remains heavily below threshold. The rebuttal narrows the scope of the paper, but it does not materially strengthen the evidence for the main framing or the broader interpretive message. I therefore keep my original recommendation.

---

> > > ### Author Response · Authors · 2026-04-04
> > >
> > > Thank you for the detailed follow-up. We appreciate the clarification that the remaining disagreement is mainly about the strength of the contribution under the intended scope.
> > > We agree that the paper does not provide a full theory of EoS dynamics or a causal account of generalization. We highlight, however, that this scope is not newly introduced in the rebuttal, but already intended in the manuscript (as also reflected in the strength section of the initial review). We are happy to further sharpen the framing to make this even clearer.
> > >
> > > Under this scope, we would like to emphasize that the contribution remains meaningful: the paper demonstrates, both empirically and via a constructive counterexample, that commonly used single-bias explanations (e.g., norm-only or sharpness-only) are insufficient to explain the behavior of gradient descent across learning rates. We have also completed a multiple-data-point extension showing that the same qualitative picture for generalization is not tied only to the single-sample setting.\
> > > We view this as a diagnostic contribution. Rather than proposing a complete mechanism, it identifies a limitation in prevailing explanations and shows that learning rate can systematically shift between competing implicit biases. Such results help refine the space of plausible theories and motivate more nuanced accounts of implicit bias beyond single-quantity descriptions.\
> > > While the empirical results are observational, the phenomenon is consistently reproduced across a broad range of settings, and the toy model provides a minimal setting in which we explore this  insufficiency rigorously.\
> > > We hope this clarifies why we believe the contribution is relevant and buildable upon, even under the clarified scope.
> > >
> > >
> > > Let us comment on your remaining critiques as following:
> > > 1. Our observed trade-off happens for models that exhibit EoS. While we do not study EoS dynamics itself in the classical temporal sense, we believe that analyzing the consequences of EoS on the final convergent solution is a natural and important component of EoS theory. In this sense, we respectfully maintain that the EoS framing is appropriate.
> > > 2. We believe documenting such a structured relationship to generalization is itself a valuable and nontrivial finding. As we emphasize above, this identifies a limitation in prevailing explanations and shows that learning rate can systematically shift between competing implicit biases in a way that impacts generalization.
> > > 3. We agree that the model is stylized. However, such minimal settings are standard and often necessary for rigorous proofs. The goal of the theory, as stated in our contribution (iii) in the submission, is to provide a counterexample in which individual biases do not generalize optimally. This supports our conjecture that the generalization behavior of neural networks can not be explained by a single implicit bias of GD.\
> > > Empirically, the phase transition we study remains with batch normalization and under $\mu$P/kernel parameterization, and the GF regime consistently selects low-norm solutions in the parameterization used for training. We agree that parameter norm is not obviously the fundamental complexity quantity in normalized/reparameterization-invariant architectures, nor do we assume there is a single universally correct one.
> > > 4. Concerning the experimental protocol, we maintain that using a fixed training-loss threshold is the most appropriate way to compare convergent endpoints across learning rates. Since different learning rates reach the solution manifold at different speeds, alternative stopping rules (e.g., fixed epochs) would confound optimization stage with endpoint geometry.\
> > > We will add a seed-variability visualization to the revision, complementing the qualitative discussion and side-by-side seed plots already present in Appendix I.6. Our point was not that seed dependence should be omitted, but that standard confidence intervals are not the most natural summary here, since initialization affects both the learning-rate location of transitions (x-axis) and the corresponding values (y-axis). We will therefore present this variability in a way that makes both aspects visible.
> > > 5. We agree that a deeper mechanistic understanding of the sharp phase transition, beyond the toy model we present, would be valuable future work. However, we believe that identifying this clear, reproducible transition aligned with the onset of EoS behavior is already a significant empirical contribution. The fact that this transition appears sharply across a broad range of experiments further supports the distinction between the flow-aligned and EoS regimes.
> > >
> > > In summary, we believe the paper offers a novel, well-supported, and practically relevant contribution that opens a new direction for studying interacting implicit biases. As an initial step in this direction, we hope it will serve as a foundation that future work can build upon.\
> > > We thank you again for your review.

---

### Decision · Program_Chairs · 2026-04-30

**Decision:**

Accept (regular)

**Comment:**

We acknowledge the concerns raised by the authors and we have carefully taken the comment into account in the decision-making process. While some reviews may be influenced by prior exposure to earlier manuscript versions, we were careful not to rely on it in isolation; instead, we considered the substance of its technical points, some of which remain valid and should be reflected in the final assessment.

During the discussion phase, the authors clarified the intended scope of the paper, while some reviewers raised concerns about the overall significance of the contributions.

After carefully reading the paper and considering the reviews together with the authors’ responses, we recommend acceptance. The paper addresses an interesting question with sound analyses and offers focused and clear contributions. That said, the paper would benefit from further discussion and we encourage the authors to incorporate additional clarifications to refine the presentation and better highlihgt the significance of their contributions.